

# Alluvial channel response to environmental perturbations: Fill-terrace formation and sediment-signal disruption

Stefanie Tofelde[1], Sara Savi[1], Andrew D. Wickert[2], Aaron Bufe[3], Taylor F. Schildgen[1,3]

[1]Institut für Erd- und Umweltwissenschaften, Universität Potsdam, 14476 Potsdam, Germany
[2]Department of Earth Sciences and Saint Anthony Falls Laboratory, University of Minnesota, Minneapolis, MN 55455, USA
[3]Helmholtz Zentrum Potsdam, GeoForschungsZentrum (GFZ) Potsdam, 14473 Potsdam, Germany

*Correspondence to*: Stefanie Tofelde (tofelde@uni-potsdam.de)

**Abstract.** The sensitivity of fluvial fill terraces to tectonic and climatic boundary conditions make them potentially useful archives of past climatic and tectonic conditions. However, we currently lack a systematic understanding of the impacts of
base-level, water discharge, and sediment discharge changes on terrace formation and associated sediment storage and release. This knowledge gap precludes a quantitative inversion of past environmental changes from terraces. Here we use a set of seven physical experiments to explore terrace formation and sediment export from a braided channel system that is perturbed by changes in upstream water discharge and sediment supply, or downstream base-level fall. Each perturbation differently affects (1) the geometry of terraces and channels, (2) the timing of terrace formation, and (3) the transient response of sediment
discharge. In general, an increase in water discharge leads to near-instantaneous channel incision across the entire fluvial system and consequent local terrace cutting, preservation of the initial channel profile on terrace surfaces, and a transient increase in sediment export from the system that eventually returns to its pre-perturbation rate. In contrast, changes in the upstream sediment supply rate may result in longer lag-times before terrace cutting, leading to a less well-preserved pre-perturbation channel profile, and may also produce a gradual change in sediment output towards a new steady-state value.
Finally, downstream base-level fall triggers the upstream migration of a knickzone, forming terraces with upstream-decreasing ages. The gradient of terraces triggered by base-level fall mimicks that of the newly-adjusted active channel, whereas gradients of terraces triggered by variability in upstream sediment or water discharge are steeper compared to the new equilibrium channel. Our findings provide guidelines for distinguishing between different types of perturbations when interpreting fill terraces and sediment export from fluvial systems.

# 1 Introduction

Sediment is moved across the Earth's surface from the production zone (mountainous regions), through the transfer zone (fluvial channels and floodplains), to the final depositional zone (continental and oceanic sedimentary basins) (Allen, 2017; Castelltort and Van Den Driessche, 2003). Because sediment production in mountainous regions is thought to vary with climatic and tectonic conditions, any changes in those conditions may be reflected in the sedimentary deposits in the transfer
or depositional zones (Alloway et al., 2007; Zhang et al., 2001). However, reliable reconstructions of past conditions from





sedimentary deposits require a detailed understanding of sediment transport along the sediment-routing (or source-to-sink) system, including any potential alteration of signals through the transfer zone, as well as the preservation of the sedimentary deposits and its signals over time (Romans et al., 2016 and references therein).

Fluvial fill terraces represent transient sediment storage along river channels, and therefore they are an important component of the sediment-routing system (e.g., Allen, 2008). They are generated by variations in river-bed elevations due to sediment deposition followed by river incision into the formerly deposited sediments (Bull, 1990). As a result of incision, remnants of the former floodplain can be abandoned by the active channel and preserved as terraces, a process we refer to as "terrace cutting". Fill terraces, as such, are an indicator of unsteadiness in the parameters that control fluvial-channel geometry. Aggradation and incision can be triggered by changing conditions at the upstream end of the river, namely the sediment to water discharge ratio, $Q_{s,in}/Q_w$ (e.g., Buffington, 2012; Gilbert, 1877; Lane, 1955; Mackin, 1948), or by base-level changes at the downstream end (e.g., Fisk, 1944; Merritts et al., 1994; Shen et al., 2012). In some cases, internal dynamics of the system, sometimes referred to as "autogenic processes", may lead to terrace formation which cannot be directly linked to any external forcing at the upstream or downstream end of the channel (e.g., Erkens et al., 2009; Limaye and Lamb, 2016; Malatesta et al., 2017; Patton and Schumm, 1981; Womack and Schumm, 1977). The cutting of terraces can either coincide with or lag behind the onset of the perturbation that drives terrace formation. The formation of fill terraces in response to external perturbations has two major implications: (1) fill terraces potentially provide a record of past environmental conditions (e.g., Bridgland and Westaway, 2008; Bull, 1990; Merritts et al., 1994); and (2) the deposition and erosion of fill terraces can alter downstream sediment signals, complicating signal propagation from catchment headwaters to long-term depositional sinks (e.g,. Allen, 2008; Castelltort and Van Den Driessche, 2003; Romans et al., 2016).

Fill-terrace deposits have been used to infer past variability in discharge (Litty et al., 2016; Poisson and Avouac, 2004) or sediment supply (Bookhagen et al., 2006; Schaller et al., 2004). For a reliable reconstruction of such parameters, however, it is essential to understand how closely terrace formation tracks environmental perturbations. Because most studied fill terraces are thousands to millions of years old and form over the course of years to thousands of years (e.g., Bookhagen et al., 2006; Schaller et al., 2004; Schildgen et al., 2002, 2016; Tofelde et al., 2017), fill-terrace formation can rarely be observed directly in nature. Consequently, we need alternative ways to investigate the formation of fill terraces and their impacts on downstream sediment discharge.

Numerical models provide an opportunity to predict the evolution of alluvial river-bed elevation over time (Blom et al., 2017, 2016; Simpson and Castelltort, 2012; Slingerland and Snow, 1988; Wickert and Schildgen, 2018). However, those predictions commonly are limited to the evolution of the longitudinal profile and do not take into account modifications of the channel width or the cutting of terraces (Blom et al., 2017, 2016; Simpson and Castelltort, 2012; Slingerland and Snow, 1988). Hancock and Anderson (2002) modeled bedrock strath terrace formation, a partially analogous process, but their erosional stream-power-based approach cannot be easily translated to transport-limited systems, where slope and long-profile evolution result from both sediment and water inputs.



Physical experiments provide an alternative approach to studying terrace formation (Baynes et al., 2018; Frankel et al., 2007; Gardner, 1983; Lewis, 1944; Mizutani, 1998; Schumm and Parker, 1973; Wohl and Ikeda, 1997). Most experimental studies have tested the cutting of terraces due to base-level fall (Frankel et al., 2007; Gardner, 1983; Schumm and Parker, 1973) or explained their formation through autogenic processes (Lewis, 1944; Mizutani, 1998). Only one experimental study by Baynes et al. (2018) investigated terrace formation as a response to changes in sediment supply ($Q_{s,in}$) or water discharge ($Q_w$), but this study focused on vertical incision into bedrock and strath-terrace cutting. Van den Berg van Saparoea and Postma (2008) performed experiments to investigate the effects of pulses in $Q_w$ and $Q_{s,in}$ on the evolution of longitudinal channel profiles and sediment discharge at the basin outlet ($Q_{s,out}$), but they did not focus on terrace formation. To our knowledge, there are no experimental studies that systematically compare how fill terraces formed through various mechanisms may differ from one another, or investigate the impacts of terrace formation on downstream sediment discharge.

In this study, we present results from seven physical experiments of braided channels in non-cohesive sediment to test three potential mechanisms of fill-terrace cutting due to external perturbations: (1) an increase in $Q_w$, (2) a reduction in $Q_{s,in}$, and (3) a fall in base level. We furthermore monitor our experiments for terrace cutting related to autogenic processes. Subsequently, we discuss: (1) channel responses to perturbations in external forcing and conditions for terrace formation, (2) differences in lag-times between the onset of the perturbation and the timing of terrace cutting and consequent differences in terraces profiles, (3) the relationship between terrace surface slope and the terrace-formation mechanism, and (4) the effects of fluvial aggradation or bed incision on sediment discharge at the outlet of the river system ($Q_{s,out}$).

## 2 Formation of fluvial fill terraces

Fluvial terraces form in response to perturbations that happen either upstream ($Q_{s,in}$, $Q_w$), or downstream (base-level changes) along the river. Such perturbations may be the result of environmental changes (external or allogenic perturbations), or the result of internal (autogenic) dynamics within the system. For each external or internal forcing mechanism, we summarize below observations from field studies, numerical models, and physical experiments.

### 2.1 Sediment to water discharge ratio ($Q_{s,in}/Q_w$)

Alluvial rivers adjust their slopes and widths such that, in a graded (steady) state, the incoming water discharge ($Q_w$) can transport the incoming sediment ($Q_{s,in}$) downstream (Buffington, 2012; Gilbert, 1877; Lane, 1955; Mackin, 1948). Scherler et al. (2015) referred to terrace formation related to changes in $Q_w$ as the '*discharge-driven model*'. In this model, a reduction in $Q_w$ leads to valley aggradation due to deposition of sediment on the riverbed. A subsequent phase of increased $Q_w$ can then cause incision. In contrast, the '*hillslope-driven model*' requires variability in $Q_{s,in}$. When an increased $Q_{s,in}$ exceeds the sediment-transport capacity of the river, the excess sediment is deposited. Deposition of sediment elevates the channel bed,




increases its slope, and thereby increases the sediment-transport capacity of the river until it matches the incoming sediment supply, $Q_{s,in}$. If $Q_{s,in}$ is reduced such that the sediment-transport capacity exceeds the sediment supply, the river tends to incise. The incision both supplements $Q_{s,in}$ with material from the channel bed and lowers the channel slope, thereby decreasing its transport capacity towards an equilibrium with the new $Q_{s,in}$.

Terrace formation due to variability in $Q_w$ has mainly been related to climatic changes, such as those caused by glacial-interglacial cycles (Penck and Brückner, 1909). Field studies favor this model when times of valley aggradation coincide with drier conditions and incision coincides with wetter conditions (Hanson et al., 2006; Scherler et al., 2015; Schildgen et al., 2016; Tofelde et al., 2017). Variability in $Q_{s,in}$ to river channels can have a variety of causes, including climatically driven changes in regolith production rates on hillslopes (Bull, 1991; Norton et al., 2015; Savi et al., 2015), climatically driven vegetation

growth that stabilizes sediment on hillslopes (Fuller et al., 1998; Garcin et al., 2017; Huntington, 1907), and exposure of regolith following glacier retreat (Malatesta et al., 2018; Malatesta and Avouac, 2018; Savi et al., 2014; Schildgen et al., 2002). Landslides also deliver sediment to rivers, and the rate of landsliding can vary in response to changes in tectonic rock uplift rates or precipitation (e.g., Bookhagen et al., 2006; McPhillips et al., 2014; Scherler et al., 2016; Schildgen et al., 2016). Increases in precipitation can mobilize additional sediment from hillslopes until the climate returns to a drier state (Dey et al.,

2016) or until hillslopes are stripped bare (Steffen et al., 2010, 2009). All of the above interpretations are based on a temporal link between the formation of fill terraces and climate proxy data, and suggest that variability in $Q_w$ and/or $Q_{s,in}$ can drive terrace formation.

Numerical models have been developed to investigate the evolution of fluvial terraces in response to variable $Q_w$ and $Q_{s,in}$ (Boll et al., 1988; Veldkamp and Vermeulen, 1989; Veldkamp and Van Dijke, 1998), and model results have been compared

to different terrace sequences in Europe (Meuse River: Bogaart and van Balen, 2000; Tebbens et al., 2000; Maas River: Veldkamp and Van Dijke, 2000; Allier River: Veldkamp, 1992, Veldkamp and Van Dijke, 1998). Similarities between modeled terraces and field observations support the conclusion that terraces can form in response to variable $Q_w$ and/or $Q_{s,in}$.

## 2.2 Base-level changes

Fluvial terraces can also be the product of changes in base level at the downstream end of the river. A drop in base level locally creates a steeper channel gradient at the downstream end. To return to a steady-state profile, the channel typically incises into its bed through an upstream-propagating knickzone, which, in the case of alluvial channels, can be highly diffuse (Begin et al., 1981; Grimaud et al., 2015; Whipple and Tucker, 1999; Wickert and Schildgen, 2018). A rise in base level leads to a local reduction in channel slope at the downstream end. To return to a steady-state profile, the channel deposits sediment upstream

of the location of base-level rise to increase the slope again. Fluvial fill terraces can thus be formed in response to alternating phases of base level rise and fall.

Although either tectonic or climatic forcing can lead to changes in base level, alternating rises and falls are most commonly associated with climatic forcing. Early observations in the Lower Mississippi Valley (USA) related valley aggradation to a



glacio-eustatic sea-level highstand and marine transgression, whereas valley incision and consequent terrace cutting was linked to sea-level fall (Fisk, 1944; Shen et al., 2012). Other field studies have related terrace formation to climatically driven alternations of sea level (Merritts et al., 1994) or lake level (Farabaugh and Rigsby, 2005). Sediment aggradation associated with sea-level rise followed by incision during sea-level fall has also been shown by a numerical model that aimed to model

the evolution of the Meuse terrace sequence in Europe (Tebbens et al., 2000; Veldkamp and Tebbens, 2001). In addition, terrace cutting following base-level drop and upstream knickzone migration has been produced in flume experiments (Frankel et al., 2007; Gardner, 1983; Schumm and Parker, 1973).

## 2.3 Complex response and autogenic processes

In addition to external (i.e., allogenic) forcing described above, internal dynamics can also drive terrace formation. Internally-driven terrace formation can result from internal feedbacks in response to a change in boundary conditions ('complex response') or due to purely internal dynamics with constant boundary conditions ('autogenic' processes). Below, we distinguish between complex responses and autogenic processes, and we discuss how they may lead to terrace development.

A non-linear response within the channel system to a linear external change can be considered a complex response (Schumm,

1979, 1973). For example, field observations (Faulkner et al., 2016; Schumm, 1979; Womack and Schumm, 1977), physical experiments (Gardner, 1983; Schumm and Parker, 1973), and numerical models (Slingerland and Snow, 1988) indicate that several terrace levels may be cut in response to a single drop in base level. Schumm (1979, 1973) observed that incision of the main stem lowered the base level for the tributaries, which consequently started to incise and transport additional sediment to the main stem. The elevated sediment supply in turn exceeded the transport capacity of the main stem, triggering deposition

in the formerly incised channel. Once the tributaries were adjusted to the new base level, sediment supply decreased, which triggered renewed incision of the main stem into the recently deposited material. Whereas the initial, externally-driven base-level drop created a first terrace level, all subsequent terraces were formed in response to internal feedbacks within the fluvial system and therefore cannot be directly linked to an external perturbation.

In contrast to a complex response, we consider autogenic terraces to be those that are formed in response to non-linear processes

within the fluvial system under constant external boundary conditions. One example is a meander cut-off, which can occur without any external perturbation and leads to a local increase in channel slope. The resulting increase in bed shear stress triggers incision and subsequent terrace formation. This phenomenon has been observed in the field (Erkens et al., 2009; Gonzalez, 2001; Womack and Schumm, 1977) and has been replicated using numerical models (Limaye and Lamb, 2016). Another example is local storage and release of sediment, which results from and feeds back into locally non-uniform sediment

transport rates. By storing or releasing sediment, each section of the channel changes the local boundary condition on the segment directly downstream ($Q_{s,in}/Q_w$) or upstream (bed elevation and thus slope). Consequently, sediment deposition, channel incision, and terrace formation can happen simultaneously in different parts of the channel (Lewis, 1944; Patton and Schumm, 1981).





# 3 Methods

To test the dynamics of fill-terrace formation in response to different external forcing conditions and the impact of terrace formation on sediment transport across the transfer zone of a source-to-sink system, we performed seven experiments at the

Saint Anthony Falls Laboratory in Minneapolis, USA, in 2015. The experimental setup consisted of a wooden box with the dimensions of 4 m x 2.5 m x 0.4 m (Fig. 1A) that was filled with quartz sand with a mean grain size of 144 μm. At the inlet, sand and water were supplied through a cylindrical wire-mesh diffuser filled with gravel to ensure sufficient mixing of sand and water. Water discharge ($Q_w$) and sediment supply ($Q_{s,in}$) could be regulated separately. At the downstream end, water and sand ($Q_{s,out}$) exited the basin through a 20 cm-wide gap that opened onto the floor below. This downstream sink was required

to avoid deltaic sediment deposition that would, if allowed to grow, eventually raise the base level of the upstream fluvial system. At the beginning of each experiment, an initial channel was shaped by hand (Fig. 1A) and the experiments were run under reference conditions ($Q_{w,ref}$ = 95 ml/s, $Q_{s,ref}$ = 1.3 ml/s) for 240 minutes. This runtime was sufficient to reach a quasi-steady state in which the average $Q_{s\_out}$ approximately equaled $Q_{s,in}$. After this "spin-up" phase, the channel had a uniform equilibrium slope of approximately 7%.

Every 30 min we stopped the experiments to perform a scan with a laser scanner mounted on the railing of the basin that surrounded the wooden box. Digital elevation models (DEMs) created from the scans have a horizontal resolution of 1 mm (Fig. 1B). Using those DEMs, we measured the evolution of channel cross-sectional profiles, longitudinal channel profiles, and surface slopes. Long profiles were calculated by extracting the lowest elevation point in each cross-section at 1 mm increments. By plotting elevation against the distance down the long axis of the box rather than against channel length, resulting

slopes are slightly overestimated due to the minor sinuosity of the channels. To directly compare terrace and channel slopes, we extracted 5 cm wide swath profiles along the terrace surfaces and the equivalent stretch of the modern channel. The width of swath profiles had to be reduced on terraces of the $DQ_w\_IQ_w$ and the $IQ_{s,in}\_DQ_{s,in}$ experiments because terraces in these runs were narrower than 5 cm. Slopes were calculated based on a linear fit through the mean elevation profiles. To assess uncertainties, the root mean square error (RMSE) was calculated between the linear model and the observed data.

Overhead photos were taken every 20 s with a fish-eye lens (Fig. 1C). Distortions of the photos were ortho-rectified in Adobe Photoshop and photos were resampled at 1 mm horizontal resolution to directly overlap with the laser scans. Photos were turned into binary images with values of 1 for wet pixels and 0 for dry pixels. This binarization was performed by transforming the *rgb* (red, green, blue) images into *hsv* (hue, saturation, value) images and then manually defining a hue cut-off for each experiment that best separates wet and dry pixels in the image (Fig. 1D). From the binary images, the number of wet pixels in

each cross-section (perpendicular to the basin margin and therefore to the average flow direction) were counted. Analyses were restricted to the areas within the orange box (Fig. 1C, D), because terraces mainly developed in this part of the channel and because we considered this sector of the channel be unaffected by the fixed location of the outlet. To calculate average





channel width, the average number of wet pixels in 1200 cross sections perpendicular to the basin margin (therefore perpendicular to the average flow direction) were counted and are reported with one standard deviation. No overhead photos were taken for the *Ctrl_1* experiment, because of an error in the camera installation.

We manually measured $Q_{s,out}$ at 10-minute intervals by collecting the discharged sediment in a container over a 10-second period and measuring its volume. This approach allowed us to estimate whether the system had returned to steady state ($Q_{s,in}$ ≈ $Q_{s,out}$) during the runs. At the same 10-minute interval, we measured bed elevation at the inlet and at the outlet to estimate the spatially-averaged channel slope. We interpreted a constant slope for over more than 30 minutes as additional evidence for a graded (steady state) channel. The data can be found in the supplementary material.

We ran seven experiments to test the impacts of changes in $Q_{s,in}$, $Q_w$, and base level on the channel. The experiments are summarized in Table 1. To investigate the effect of $Q_w$, we ran two separate experiments: in one experiment we doubled $Q_w$ ($IQ_w$ = increase discharge) to 190 mL/s at 240 min (end of the spin-up time) and in the other experiment we first halved $Q_w$ to 48 ml/s at 240 min and then returned to the initial 95 mL/s at 480 min ($DQ_w\_IQ_w$ = decrease discharge, increase discharge). To test the effect of $Q_{s,in}$, we ran one experiment in which we reduced the $Q_{s,in}$ by 83% to 0.22 ml/s ($DQ_{s,in}$ = decrease sediment supply) at 240 min and another one in which we first doubled $Q_{s,in}$ to 2.6 ml/s at 240 min and then halved $Q_{s,in}$ again to the initial 1.3 ml/s at 480 min ($IQ_{s,in}\_DQ_{s,in}$ = increase sediment supply, decrease sediment supply). All $Q_{s,in}$ and $Q_w$ changes were imposed instantaneously, resulting in a step function in the forcing. Immediately before imposing these changes, we covered the near-channel surface with a thin layer of red sand to optically identify the area that is reworked after the change. We ran one experiment in which we dropped the base level by 10 cm gradually over 20 min starting at 240 min, resulting in a base-level lowering rate of 0.5 cm/min ($BLF$). For this experiment, we started with a base level higher than in the initial setting by flooding the basin surrounding the wooden box (Fig. 1A). The final base level equaled those of the other experiments. In this experiment, the red sand was applied immediately before the onset of base-level lowering. Additionally, we performed two control experiments in which we made no changes to the initial conditions in order to investigate whether terraces would form in our experiment without any change in external forcing (*Ctrl_1, Ctrl_2*).

## 4 Results

Fluvial terraces were cut in the experimental runs *$IQ_w$, $DQ_w\_IQ_w$* (in the *$IQ_w$* phase)*, $DQ_{s,in}$, $IQ_{s,in}\_DQ_{s,in}$* (in the *$DQ_{s,in}$* phase) and *BLF (*Fig. 2, 3*)*. No terraces were formed after the 'spin-up' time of *Ctrl_1* and *Ctrl_2*. The terraces visible in the cross-section of *Ctrl_2* formed in response to incision during the 'spin-up' phase and did not substantially develop after 240 min (Fig. 3B, red line).

To form fill terraces, changes in channel-bed elevation and channel width are required. In our experiments, channel-elevation changes occurred by sediment deposition or incision (Fig. 4). However, these bed-elevation changes were not uniform along the channel reach (Fig. 4). In the runs *Ctrl_1* and *Ctrl_2,* the longitudinal profiles were stable over time and only minor



lowering in bed elevation (max. 4 cm) occurred at the upstream end (Fig. 4A, B). A sudden increase in $Q_w$ ($IQ_w$, and the $IQ_w$ phase o*f* $DQ_w\_IQ_w$) or a decrease in $Q_{s,in}$ ($DQ_{s,in}$, and the $DQ_{s,in}$ phase of $IQ_{s,in}\_DQ_{s,in}$) both led to river incision, which was most pronounced at the upstream end (Fig. 4C, D, G, H) and was, in most cases, not recognizable at the downstream end (Fig. 4D, G, H), where the channel-bed elevation was fixed due to the steady base level. Sediment deposition in the channels

followed a decrease in $Q_w$ ($DQ_w$ phase of $DQ_w\_IQ_w$) or an increase in $Q_{s,in}$ ($IQ_{s,in}$ phase of $IQ_{s,in}\_DQ_{s,in}$), which was again most recognizable at the upstream end of the reach (Fig. 4E, F). The drop in base level, however, caused maximum incision at the downstream end, and the incision wave migrated upstream as a knickzone (Fig. 4I).

The evolution of slope and width of the active channel were tracked through time (Fig. 5). The *Ctrl_1* and *Ctrl_2* experiments only showed a marginal decrease of channel slopes after the 240 min 'spin-up' time from ~ 0.074 and 0.071 to around 0.070

and 0.067 (~6 % reduction; Fig. 5A). As such, we consider any change in slope after the 'spin-up' time that is on the same order as those observed in *Ctrl_1* and *Ctrl_2* as ongoing adjustment to the reference condition as opposed to the result of an external perturbation. Channel width in the control experiments varied slowly between ca. 20 cm and 35 cm.

An instant doubling of $Q_w$ ($IQ_w$; Fig. 5B) resulted in a rapid, exponential decrease in channel slope. After approximately 480 min, the slope was reduced from ~0.072 to ~0.043 (40% reduction), and new stable conditions were reached. The doubling of

$Q_w$ also triggered an instant narrowing of the channel from ~35 cm to ~15 cm (~57 % decrease), followed by subsequent slow widening.

A sudden reduction in $Q_w$ to half its initial value ($DQ_w\_IQ_w$; Fig. 5C) resulted in an increase in slope from ~0.072 to ~0.085 (18% increase) between 240 and 480 min runtime, and a widening of the channel from about 25 cm to about 45 cm (~80% increase) during the same time. The subsequent doubling in $Q_w$ back to its initial value triggered a rapid (nearly exponential)

reduction in slope back to the initial ~0.072 (~15% reduction) and an instantaneous narrowing of the channel (~45% reduction) followed by slow widening.

A reduction in $Q_{s,in}$ by 83% ($DQ_{s,in}$; Fig. 5D) triggered a decrease in channel slope. The rate of decrease was lower than in the $IQ_w$ run, and the new slope stabilized around 0.06 (24% reduction). An instantaneous decrease in channel width also occurred, but this change was again less pronounced than what we observed in the $IQ_w$ experiment (~33% reduction). No subsequent

widening of the channel was detectable.

An increase in $Q_{s,in}$ ($IQ_{s,in}\_DQ_{s,in}$; Fig. 5E) led to an increase in channel gradient from about 0.070 to about 0.078 (11% increase) and an increase in channel width from about 30 cm to about 55 cm (~83% increase). The subsequent reduction in $Q_{s,in}$ led to a decrease of the channel slope and an instantaneous channel narrowing to < 30 cm, followed by subsequent widening back to the initial width of ~30 cm

For the base-level fall experiment (*BLF*; Fig. 5F), channel slope instantly and rapidly increased after the onset of base-level fall from about 0.047 to 0.073 (55% increase), and it increased at a slower rate further to about 0.08, before lowering back to 0.072. However, these slope values are simply calculated based on the height difference at the inlet and outlet, ignoring any variability in slope along the experiment reach that is, in the *BLF* experiments, significant due to knickzone propagation. The drop in base level resulted in a sudden drop in channel width, followed by three cycles of channel widening and narrowing. In





summary, we observed that an increase in $Q_w$ and a decrease in $Q_{s,in}$, resulted in an immediate decrease in channel slope (through upstream incision) and an instant reduction in channel width, whereas a drop in base level caused an increase in channel slope (through downstream incision) and a reduction in channel width (Fig. 5).

The time of terrace cutting lagged minutes to hours behind the onset of the perturbation (Fig. 5). Lag-times were determined

from overhead photos and are defined as the time interval between the onset of the perturbation (at minute 240 or 480) and the last time the future terrace surface was occupied by water. The times given in Fig. 5 refer to the last occupation of the areas for which swath profiles were extracted (Fig. 6 right panel). In the two experiments in which we changed $Q_w$ and in the $IQ_{s,in}$ _$DQ_{s,in}$ experiment, terrace cutting in the upstream reach of the channel (Fig. 3 right column, Fig. 4; dashed arrows) began within ~5 minutes after the change in boundary conditions (Fig. 5; black arrows). In the $IQ_w$ experiment, for example, the

majority of the $T_A$ terrace was cut instantly (no removal of red sand) and only a small part at the downstream end was occupied again until 6 minutes after perturbation (Fig. 2A, B). In the $DQ_{s,in}$ experiment, however, the $T_A$ and $T_B$ terraces were cut 297 and 144 min after the perturbation (Fig. 5D). In the $BLF$ experiment, terraces in the downstream channel reach were cut immediately after the onset of base-level drop, but were mostly destroyed within 30 min (Fig. 2C, D). Terrace cutting in the upstream part of the basin began 112 and 117 min after the initial perturbation (Fig. 5F).

To analyze how well the channel-bed profiles immediately preceding the time of perturbation were  preserved by the terraces, we compared the elevation profiles of the two terraces on each side of the channel (yellow and orange lines) with the channel that existed at the onset of perturbation (red line) (Fig. 6). In experiments with increasing $Q_w$ ($IQ_w$, $IQ_w$ phase of $DQ_w$_$IQ_w$) or base-level changes ($BLF$), the elevation profiles of the terraces are similar to the initial channel profile (Fig. 6A, B and E). In cases of changes in $Q_{s,in}$ ($DQ_{s,in}$, $DQ_{s,in}$  phase of $IQ_{s,in}$_$DQ_{s,in}$), the terraces were cut at lower elevations than the former channel

(Fig. 6C, D). In the $DQ_{s,in}$ experiment, terraces on either side of the channel formed at different elevations, with one terrace about 3 cm below the other (Fig. 6C, 3E; unpaired terraces). In contrast, terraces in the other four experiments are at approximately the same elevation (paired terraces) (Fig. 4, 6). Despite being paired, the slopes of the two terraces differ from each other by between 5% ($IQ_{s,in}$ _$DQ_{s,in}$) and 33% ($IQ_w$). When comparing terrace slopes to the active channel slopes (blue lines) at the end of each run, terrace slopes are steeper in all experiments in which upstream conditions ($Q_w$, $Q_{s,in}$) were changed

(Fig. 6 A-D). In contrast, the slopes of the terraces and the active channel in the $BLF$ experiment are similar to each other (Fig. 6E).

Changes in boundary conditions also affected sediment discharge at the outlet (Fig. 5, lowest panels). An instantaneous doubling of $Q_w$ ($IQ_w$; Fig. 5B) resulted in an instant increase in $Q_{s,out}$ to more than 20 times $Q_{s,in}$. This rapid increase was followed by an exponential decay down to the initial $Q_{s,out}$ value. A sudden reduction in $Q_w$ to half its initial value ($DQ_w$_$IQ_w$;

Fig. 5C) resulted in a decrease in $Q_{s,out}$. The subsequent doubling in $Q_w$ back to its initial value triggered a rapid increase in $Q_{s,out}$ that decayed over time. In contrast, neither the instantaneous reduction in $Q_{s,in}$ by 83% ($DQ_{s,in}$; Fig. 5D) nor the doubling in $Q_{s,in}$ ($IQ_{s,in}$_$DQ_{s,in}$; Fig. 5E) triggered a measurable change in $Q_{s,out}$. For the base-level fall experiment ($BLF$; Fig. 5F), $Q_{s,out}$ could not be measured before and during the base level drop, because the basin surrounding the wooden box was flooded for this experiment. $Q_{s,out}$ was only measured from minute 280 onwards, which corresponds to minute 40 after the 'spin-up' of the



base level fall. At that time, $Q_{s,out}$ was still about 10 times higher than $Q_{s,in}$, and $Q_{s,out}$ decreased approximately linearly from that time onwards.

## 5 Discussion

### 5.1 Channel response to perturbations and conditions of terrace formation

The preservation of fluvial fill terraces requires that vertical incision outpaces lateral erosion on one or both sides of the active channel. Whether this occurs depends on the response of alluvial channels to changing boundary conditions, which can occur through adjustments to their slope, wetted perimeter (width and depth), and/or bed-surface texture (grain-size distribution) (Blom et al., 2017; Buffington, 2012 and references therein). Because the grain-size distribution in our experiments remained constant, we focus our discussion on the externally forced adjustments of channel slope ($S$) and width ($w$) during terrace formation.

In our experiments, river-bed aggradation and channel steepening occurred after a decrease in $Q_w$ and after an increase in $Q_{s,in}$, whereas river incision (with terrace cutting) and channel-slope lowering were driven by an increase in $Q_w$, a decrease in $Q_{s,in}$, or a fall in base level (Figs. 2, 3, 4). In the case of base-level fall, incision began at the downstream boundary and diffused upstream, producing a transient steepening. The evolution of longitudinal channel profiles in our experiments is in agreement with earlier flume studies that investigated channel response to upstream (van den Berg van Saparoea and Postma, 2008) and downstream (Begin et al., 1981; Frankel et al., 2007) perturbations, as well as with numerical models that predict the evolution of longitudinal profiles following variations in $Q_{s,in}$ or $Q_w$ (Blom et al., 2017; Simpson and Castelltort, 2012; Wickert and Schildgen, 2018). In addition to slope changes, channels can also adjust to external forcing by changing their width (Fig. 5; Buffington, 2012; Church, 1995; Curtis et al., 2010; Dade et al., 2011). In all experiments, an increase in channel width occurred during aggradation (reduced $Q_w$, increased $Q_{s,in}$), and an instantaneous decrease in channel width occurred at the start of incision (increased $Q_w$, reduced $Q_{s,in}$, BLF; Fig. 5).No terraces were formed during the two control experiments after the 'spin-up' time. However, this finding does not imply that autogenic terraces do not exist in natural systems, as meander bend cut-off (Erkens et al., 2009; Gonzalez, 2001; Limaye and Lamb, 2016; Womack and Schumm, 1977) could not be tested with our experimental setup. We observed internal variability in sediment storage and release, for example in the form of bank collapse due to lateral channel migration during the experiments. However, local lateral sediment input through bank collapse did not trigger terrace formation in our experiments. Our experimental set-up also precluded terrace formation in response to internal feedbacks between the main stem and tributaries (Schumm, 1979, 1973, Gardener 1983, Schumm and Parker 1973, Slingerland and Snow 1988).

In order to link drivers and response, we turn to the work of Wickert and Schildgen (2018), who coupled equations for flow, sediment transport, and channel morphodynamics to solve for long-profile changes in transport-limited rivers. From this work,

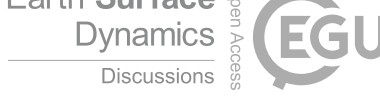

in which channel width is allowed to self-adjust following Parker (1978), we distill the following relationships between channel width ($w$), slope ($S$) and either $Q_{s,in}$ or $Q_w$:

$$Q_w \propto \frac{w}{S^{7/6}} \qquad (1)$$

and

$$Q_{s,in} \propto w \qquad (2)$$

Eq. 2 predicts the observed reduction in channel width after a decrease in $Q_{s,in}$ (Fig. 5, eq. 2). Eq. 1 predicts that slope should
decrease as water discharges increases, which is consistent with the observed decrease in slope from about 0.072 to 0.043 (Fig. 5B) in the $IQ_w$ experiment, in which water discharge doubled. However, this amount of slope decrease should be matched by an 8% increase in channel width, which runs contrary to the observed instantaneous reduction in channel width by ~57% followed by gradual widening. This response is transient, whereas Wickert and Schildgen (2018) assume an equilibrium width; the relationship between time-evolving slope, width, and basal shear stress is the most likely cause of this discrepancy. The equilibrium-width solution used by Wickert and Schildgen (2018) assumes a constant ratio between the basal shear stress at bankfull discharge ($\tau_b$) and the critical shear stress for the initiation of sediment motion ($\tau_c$), which can be described by (Parker, 1978):

$$\tau_b = (1 + \varepsilon)\tau_c \qquad (3)$$

Parker (1978) suggested that the fraction of excess shear stress at bankfull flow ($\varepsilon$) is about 0.2 for self-formed gravel-bed rivers with equilibrium widths. Empirical measurements have confirmed an epsilon of 0.2 in a large number of rivers across the US (Phillips and Jerolmack, 2016), but Pfeiffer et al. (2017) illustrated that $\varepsilon$ increases in tectonically active regions. It could be that rapid uplift is analogous to incision in our experiment during its transient-response phase, causing the channel to narrow and $\tau_b$ to increase, which further accelerates incision. Our experimental results demonstrate that accurately simulating long-profile evolution may require an improved understanding of the transient response of channel width.

## 5.2 Preservation of channel profiles

A common application of fluvial-terrace mapping is to reconstruct paleo-longitudinal channel profiles from terrace remnants (e.g., Faulkner et al., 2016; Hanson et al., 2006; Pederson et al., 2006; Poisson and Avouac, 2004). Reconstructed longitudinal profiles from terrace remnants are thought to be representative of the former channel profiles, ideally of conditions immediately prior to perturbations. However, morphological adjustments of a channel to external perturbations require time, such that the geomorphological response can lag behind the changes in environmental parameters (e.g., Blum and Tornqvist, 2000; Tebbens et al., 2000; Vandenberghe, 2003, 1995). The lag-time between external perturbations and the onset of terrace cutting



determines the degree of reworking of terrace material. Consequently, the shorter the lag-time, the better the preservation potential of environmental conditions that existed prior to the time of perturbation.

In our experiments, the terrace surfaces preserve the former channel elevation profiles in the two increased $Q_w$ experiments and in the *BLF* experiment (Fig. 6A, B and E). In contrast, in the decreased $Q_{s,in}$ experiments, terrace-elevation profiles are

lower than the river channel immediately preceding the perturbation and, in case of the $DQ_{s,in}$ run, the terraces are also unpaired (Fig. 6C, D). Focusing on the upstream-perturbation experiments first, we observed short lag-times between perturbations and terrace cutting in all $Q_w$ related experiments (Fig. 5B, C), which ensured good preservation of the channel profile prior to perturbation (Fig. 6A, B). Similarly, terrace cutting in the $IQ_{s,in}\_DQ_{s,in}$ experiment was characterized by short ($T_B$) or no ($T_A$) lag-times (Fig. 5E). The small discrepancy between terrace slopes and initial channel slopes is a result of slope variations

between the center of the channel belt (where initial and final channel profiles were measured), and the sides of the channel belt, where the terrace slopes were measured. In contrast, terrace cutting in the $DQ_{s,in}$ experiment occurred with a several hour delay. The difference in lag-times between the $T_A$ and $T_B$ terrace of about two and a half hours resulted in unpaired terraces, with elevation profiles several cm below the channel profile prior to perturbation (Fig. 6C).

The length of the lag-time between the perturbation and the abandonment of a terrace surface depends on how effectively

vertical incision outcompetes lateral erosion. Bufe et al. (2018) have shown that the rate of lateral channel migration scales inversely with the height of valley walls (elevation difference between a terrace surface and the active channel). As such, the higher the incision rate after perturbation, the faster wall-heights grow and the more lateral mobility is reduced. Due to this positive feedback, rapid incision after a perturbation should result in short lag-times between the onset of the perturbation and terrace cutting and a good preservation of the channel profile that existed prior to perturbation. In contrast, if the river incises

more slowly, terraces may be cut long after incision initiates, and the terrace profile will not directly reflect the channel profile prior to perturbation.

The lag time between the onset of base-level fall and the cutting of terraces in the upstream part of the valley is about ~115 min (Fig. 5I), which was the time required for the knickpoint to propagate upstream. As such, for base-level-fall-related terraces, the temporal lag between base-level fall and terrace cutting increases with increasing distance to the terrace upstream.

In other words, terrace surfaces created through upstream knickpoint migration are diachronous, become progressively younger upstream despite being physically a continuous unit. Faulkner et al. (2016) found decreasing OSL ages with upstream distance in a fill terrace along the Chippewa River, USA that formed in response to base-level fall. Similar conclusions were also reached by Pazzaglia (2013). In comparison, incision was initiated near-synchronously along the entire reach when incision was triggered by a change in upstream boundary conditions ($IQ_w$, $DQ_{s,in}$; Fig. 4C, D). In summary, lag-times between the onset

of the perturbation and terrace cutting depend on the combination of local incision rates after the perturbation and the trigger for incision (base-level fall vs. a change in upstream conditions).

Lag-times between the perturbation and the onset of terrace cutting can be important when dating the surfaces of fluvial fill terraces in the field. Common methods to date the onset of river incision include the dating of terrace surface material with cosmogenic exposure dating (e.g., Schildgen et al., 2016; Tofelde et al., 2017), dating sand or silt lenses with optically




stimulated luminescence close to the terrace surface (OSL; e.g., Fuller et al., 1998; Schildgen et al., 2016; Steffen et al., 2009) or dating embedded organic material with [14]C (Farabaugh and Rigsby, 2005; Scherler et al., 2015). When transferring our observations to a field scenario, the ~2h or more of channel material reworking before terraces were cut within the upstream part of the reach in the *BLF* and the *DQ$_{s,in}$* experiment would result in terrace ages that are younger than the time of perturbation.

The best temporal correlations between the perturbation and the terrace surface ages are achieved by those formed by changes in $Q_w$ due to the fast onset of vertical incision and minimal reworking of terrace surface material. To assess the significance of this time-lag in natural systems requires more work on how to scale the experiment to larger channels.

### 5.3 Differences in terrace surface slopes

To reliably use fluvial terraces to reconstruct paleo-environmental conditions (i.e., changes in base level, $Q_{s,in}$ or $Q_w$), the identification of the terrace formation mechanism is important. We found that for $Q_{s,in}$ or $Q_w$ related terraces, the slopes of terrace surfaces are always steeper than the active channel (the new steady state channel after the perturbation), whereas the slope of terraces formed due to downstream perturbations is very similar to that of the active channel (Fig. 6). Similar observations have been made in the field. Poisson and Avouac (2004) measured a reduction in channel slope between terraces

due to deeper incision at the upstream end of a flight of terraces in the Tien Shan. They related the changes in longitudinal profiles (inferred from the terraces) to changes in $Q_w$. In contrast, Faulkner et al. (2016) measured terraces in the Chippewa River, a tributary to the Mississippi River, which were created in response to base-level fall and upstream knickpoint migration due to incision of the Mississippi channel bed after deglaciation. They observed no major slope change between the longitudinal profile reconstructed from the terrace and the modern channel. According to Wickert and Schildgen (2018), the

relationship between slope $S$, $Q_{s,in}$ and $Q_w$, for alluvial rivers taking self-adjusting channel width and channel roughness into account, can be described as:

$$S \propto \left(\frac{Q_{s,in}}{Q_w}\right)^{6/7} \tag{4}$$

According to this relationship, a decrease in $Q_{s,in}$ or an increase in $Q_w$ results in a lower channel slope. A drop in base level should, after the signal has propagated upstream, result in a slope similar to the channel before the perturbation because the

$Q_{s,in}/Q_w$ ratio is unchanged. Hence, our findings suggest that slope comparisons between the terrace surfaces and the active channel could indicate whether an upstream or a downstream perturbation caused the cutting of the terraces. However, such comparisons are only informative if the active channel is still graded to the boundary conditions that initiated incision and terrace cutting. In addition, this approach to identifying the terrace-formation mechanism requires negligible tectonic tilting of the terraces after cutting.

In tectonically active regions, both strath and fill terraces have been used to infer tectonic deformation rates (e.g., Hu et al., 2017; Lavé and Avouac, 2000; Litchfield and Berryman, 2006; Peters and van Balen, 2007). Variability in slopes over time,





derived from reconstructed longitudinal channel profiles, have been used to infer local deformation rates (e.g., Hu et al., 2017; Lavé and Avouac, 2000). The observed slope differences between terrace surfaces and the active channel after upstream perturbations in our experiments (Fig. 6), however, imply that slope differences observed in the field can only be used to infer tectonic deformation rates if one can either rule out (Lavé and Avouac, 2000) or quantify slope changes related to changing

$Q_w$ and/or $Q_{s,in}$ (Pazzaglia, 2013). Because the slope changed in our experiments of upstream perturbations, incision rates were not uniform along the channel (Fig. 4). Litchfield and Berryman (2006) also measured variable fluvial incision rates based on terrace heights at several locations along 10 major rivers located along the Hikurangi Margin, New Zealand. Accordingly, gradients in incision rates along rivers should be interpreted in the context of potential changes to the shape of the longitudinal profile.

## 5.4 Signal propagation and implications for stratigraphy

Alluvial rivers adjust their channel geometry (slope, width, and depth) with regards to incoming $Q_w$ and $Q_{s,in}$ (Lane, 1955; Mackin, 1948). Consequently, a change in input parameters leads to an adjustment in channel geometry through the deposition or remobilization of sediment until new equilibrium conditions are reached (transient phase). The required adjustment time is

referred to as the response time of the channel (Paola et al., 1992). We expect that a change in $Q_w$ will trigger a transient response in $Q_{s,out}$ during that adjustment phase, but $Q_{s,out}$ is expected to return to the initial value once the new steady-state channel geometry is reached (Armitage et al., 2013, 2011). In contrast, a change in $Q_{s,in}$ will result in a permanent adjustment of $Q_{s,out}$ once the channel geometry is adjusted to the new conditions (Allen and Densmore, 2000; Armitage et al., 2011). According to Eq. 4, an increase in $Q_w$ is expected to result in a lower channel slope and, therefore, to initiate river incision. In

our $IQ_w$ experiment, we observed an up to 20-fold increase in $Q_{s,out}$ after the perturbation, followed by a return to previous $Q_{s,out}$ values at about 300 min after the perturbation (equivalent to 540 min runtime; Fig. 5B). As such, the $Q_{s,out}$ signal is generated during the transient phase of slope adjustment. This pattern is schematically shown in Fig. 7C. Because $Q_{s,in}$ was held constant during the experiment, the additional sediment that reached the outlet was remobilized from within the channel, in particular from the upstream part (Fig. 4C, G). This result corroborates previous observations from physical experiments

(van den Berg van Saparoea and Postma, 2008) and numerical models (Armitage et al., 2013; Simpson and Castelltort, 2012). In contrast, a decrease in $Q_w$ requires a steeper channel gradient, which is achieved through sediment deposition within the channel (Fig. 4E). In our experiments, $Q_{s,out}$ was reduced relative to the upstream sediment supply during the transient slope-adjustment phase (Fig. 5C and 7D).

A decrease in $Q_{s,in}$ should, following the achievement of a graded channel profile, also produce a reduced $Q_{s,out}$, whereas an

increase in $Q_{s,in}$ should result in enhanced sediment discharge at the outlet (Allen and Densmore, 2000; Armitage et al., 2011). According to Eq. 4, a reduction in $Q_{s,in}$ will trigger temporary incision, because a lower slope is required to transport less sediment with the same amount of $Q_w$, whereas an increase in $Q_{s,in}$ will require a steeper slope and thus trigger aggradation. We observed channel incision and slope reduction in the $DQ_{s,in}$ experiments (Fig. 4D, H and 5D, E) and aggradation and slope



increase following an increase in $Q_{s,in}$ (Fig. 4F and 5E). However, in none of the experiments with variable $Q_s$ is a clear signal in $Q_{s,out}$ recognizable during the transient phase of slope adjustment (Fig. 5D and E, 7E and F). We consider the negative feedback between $Q_{s,in}$ and the bed-elevation change during the transient channel-adjustment phase as the main reason for this lack of response (Simpson and Castelltort, 2012; van den Berg van Saparoea and Postma, 2008). The additional sediment

supplied upstream is deposited within the channel, resulting in aggradation, and is therefore not detectable at the outlet. When less sediment is supplied upstream, the channel incises and complements the supplied upstream sediment with remobilized sediment from within the reach, such that once again, no clear reduction in $Q_{s,out}$ is visible during the adjustment phase. We did not run the experiments long enough to analyze the adjusted steady-state phase, but we would expect that once the channel has adjusted to new equilibrium conditions, the changes in $Q_{s,in}$ will eventually become visible in $Q_{s,out}$ (Allen and Densmore,

2000; Armitage et al., 2011).

Internal dynamics within the channel can lead to variability in $Q_{s,out}$ even without external forcing. In the *Ctrl_1* and *Ctrl_2* experiments, scatter in the $Q_{s,out}$ signal was up to 5 times the value of $Q_{s,in}$ (Fig. 5a). This variability is due to continuous lateral movement of the channel and subsequent bank collapse, which results in stochastic contributions of additional sediment. Lateral channel mobility of a stream varies with water and sediment discharge (Bufe et al., 2018; Wickert et al., 2013).

However, if the volume of sediment mobilized from valley walls due to lateral migration is much larger than the change in $Q_{s,in}$, then no clear signal in $Q_{s,out}$ might be recognizable, even after channel adjustment. The channel instead will continually adjust to the stochastic lateral input of sediment.

Regarding $Q_{s,out}$ signals, we conclude that terraces, floodplains, and the channel itself act as a temporary storage space where sediment can be deposited or remobilized when boundary conditions change (Coulthard et al., 2005; Simpson and Castelltort,

2012; van den Berg van Saparoea and Postma, 2008). Our data support earlier findings by Simpson and Castelltort (2012) and van den Berg van Saparoea and Postma (2008), who concluded from their respective numerical model and physical experiments that signals of $Q_w$ variability create an amplified signal in $Q_{s,out}$, whereas changes in $Q_{s,in}$ create a dampened signal in $Q_{s,out}$ due to the a negative feedback between $Q_{s,in}$ and channel gradient. Our experiments, illustrated schematically in Fig. 7, also suggest that $Q_w$-driven $Q_{s,out}$ signals are transient, and that as the channel slope adjusts to the new input $Q_w$, $Q_{s,out}$ evolves

back to its initial steady-state value. In contrast, $Q_{s,out}$ signals driven by changes in $Q_{s,in}$ may not be observable during transient channel adjustment, but will occur and persist once the channel has adjusted to new steady-state conditions.

Our findings also have implications for geochemical signatures of sediment, for example the concentration of cosmogenic [10]Be, which is commonly measured to infer catchment mean denudation rates (Bierman and Steig, 1996; Brown et al., 1995; Granger et al., 1996). In cases of channel aggradation, $Q_{s,out}$ is reduced compared to $Q_{s,in}$ due to deposition within the channel

(Fig. 7B, D, F). The exported sediment could be sourced from incoming sediment that is not deposited (grey circles) and/or mixing with remobilized sediment within the channel (yellow circles). In general, net deposition along the channel leads to the majority of the grains at the outlet being freshly delivered from hillslopes, thus carrying the modern chemical composition at the time of transport. In contrast, during incision, older material stored within the channel, floodplain, and/or terraces is remobilized and contributes to the temporary peak in $Q_{s,out}$ (Fig. 7A, C, E). Shortly after the perturbation, most of the



remobilized sediment will be stratigraphically high and relatively young (yellow circles), but older material from deeper layers (orange and red circles) will progressively be remobilized and mixed with young material from upstream. Cosmogenic nuclide analyses along the eastern Altiplano margin (Hippe et al., 2012) and in the Amazon basin (Wittmann et al., 2011) indicate that sediment can be stored within the fluvial system over thousands to millions of years. Remobilization of formerly deposited

material and subsequent mixing with fresh hillslope material (incoming sediment) can temporally buffer signals stored in the geochemical composition of detrital river sediments (e.g., Tofelde et al., 2018; Wittmann et al., 2016, 2011). We conclude that modern chemical signals are more likely to be transmitted through the system during aggradation phases, whereas local sediment that has been transiently stored may strongly overprint the signal of modern sediments during times of incision.

## 6 Summary and Conclusion

We performed seven physical experiments to investigate the effects of changing boundary conditions ($Q_{s,in}$, $Q_w$, base level) on channel geometry, fill-terrace formation and signal propagation in fluvial sediments. In particular, we recorded the evolution of channel slope and width during adjustment to new boundary conditions. Furthermore, we explored the conditions under which fill terraces form and how well they preserve the channel profile prior to perturbation based on lag-times between the onset of perturbation and terrace cutting, synchronicity of incision along the length of the channel, and the relationship between

terrace-surface slopes and terrace-formation mechanisms. In addition, we examined the implications of changing boundary conditions on signal propagation through the sediment-routing system. Our experimental findings can be summarized as follows:

1. An increase in $Q_w$, a decrease in $Q_{s,in}$, or a drop in base level triggered river incision and terrace cutting, combined with an instantaneous reduction in channel width.

2. The observed reduction of channel width after an increase in $Q_w$ runs contrary to the expected channel widening under equilibrium conditions. This finding indicates that the transient response of the fluvial system – not captured in the equilibrium relationship between channel width ($w$), discharge ($Q_w$) and slope ($S$) from the coupled equations of Wickert and Schildgen (2018) – may be significant. We suggest that the transient channel-width response may lead to an excess shear stress at bankfull flow ($\varepsilon$) that differs from the

commonly assumed and encountered value of ~$0.2\tau_c$ (Parker, 1978; Phillips and Jerolmack, 2016).

3. The lag-time between an external perturbation and terrace cutting determines (i) how well terraces preserve and record the pre-perturbation channel longitudinal profile, and (ii) the degree of reworking of terrace-surface sediment. We found that rapid incision creates terraces that effectively track external forcing and record the pre-perturbation channel profile, whereas slower incision enables lateral migration of the channel,

with terraces cut during the transient phase that lag behind the timing of forcing and do not preserve the pre-perturbation channel profile.





4. In comparison to incision triggered by changes in upstream conditions ($Q_{s,in}$, $Q_w$), which occurred near synchronously along the entire channel reach, incision triggered by base-level fall created the upstream migration of a knickzone. Consequently, the lag-time between the drop in base level and the cutting of a terrace surface increased with distance upstream. Due to increased surface reworking with distance upstream, the preservation potential of the channel surface prior to perturbation decreases with distance upstream.

5. Terraces related to upstream perturbations ($Q_{s,in;}$ $Q_w$) were always steeper than the active channel at the end of the experiment. In contrast, the final, adjusted channel slope was similar to the initial channel slope in the base-level fall experiment. This difference can help to identify the terrace-formation mechanism in field settings, but complicates the interpretation of terraces as tectonic deformation markers.

6. Changes in $Q_w$ caused a measurable signal in $Q_{s,out}$ during the transient phase of channel adjustment, whereas $Q_{s,out}$ signals related to changes in $Q_{s,in}$ were not detectable during the transient phase due to buffering (sediment storage or release) of $Q_{s,in}$ as the channel adjusted its gradient. Changes in $Q_{s,in}$ are thought to become more recognizable once the channel has adjusted to new steady-state conditions. Because $Q_w$-driven signals generated an amplified $Q_{s,out}$ signal during the transient channel response phase, they have a higher potential to be preserved in the stratigraphic record than do changes in $Q_{s,in}$ if upstream conditions are changing periodically with a period that is shorter than the channel response time.

7. Signals extracted from the geochemical composition of sediments are more likely to represent modern-day conditions during times of aggradation, whereas the signal will be temporally buffered due to mixing with older, remobilized sediment during times of channel incision.

We experimentally demonstrated that fluvial fill terraces can form due to changes in water discharge (climate), sediment supply (climate or tectonics), or base level (climate or tectonics). We demonstrated major differences in lag-times between the onset of perturbation and terrace cutting, and consequently in the resulting terrace elevation profiles and slopes. Therefore, information on the initial channel and environmental conditions that existed prior to the time of perturbation are not always well preserved in the terraces. We conclude that identifying the mechanism of fluvial fill terrace formation is necessary to reconstruct past climatic or tectonic forcing accurately and that sediment storage and remobilization of sediment in alluvial channels can influence signals stored in the discharge ($Q_{s,out}$) or chemical composition of sediment.



*Author contribution*. S.T., S.S. and A.W. designed and built the experimental setup. S.T. and S.S. performed the experiments. S.T. analyzed the data with the help of S.S., A.W. and A.B. All authors discussed the data, designed the manuscript and commented on it. S.T. designed the artwork.

*Competing interests*. The authors declare that they have no conflict of interest.

*Acknowledgements*. We thank Ben Erickson, Richard Christopher, Chris Ellis, Jim Mullin, and Eric Steen for their help in
building the experimental setup and installing equipment. We are also thankful to Jean-Louis Grimaud and Chris Paola for fruitful discussions and suggestions. Financial support for this study was provided by the Emmy-Noether-Programme of the Deutsche Forschungsgemeinschaft (DFG), grant number SCHI 1241/1-1 awarded to T.S. and start-up funds provided to A.W. by the University of Minnesota. S.S acknowledges support from the Alexander von Humboldt Foundation, grant ITA 1154030 STP.

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



**Table 1. Water and sediment inputs to the experiments.**

| Experiment | 0 – 240 min (reference conditions) | | 240 – 480 min | | 480 min until end | | Graphical description |
|---|---|---|---|---|---|---|---|
| | $Q_w$ (ml s$^{-1}$) | $Q_{s,in}$ (ml s$^{-1}$) | $Q_w$ (ml s$^{-1}$) | $Q_{s,in}$ (ml s$^{-1}$) | $Q_w$ (ml s$^{-1}$) | $Q_{s,in}$ (ml s$^{-1}$) | |
| Ctrl_1 | 95 | 1.3 | 95 | 1.3 | 95 | 1.3 | |
| Ctrl_2 | 95 | 1.3 | 95 | 1.3 | 95 | 1.3 | |
| $IQ_w$ | 95 | 1.3 | 190 | 1.3 | 190 | 1.3 | |
| $DQ_w\_IQ_w$ | 95 | 1.3 | 47.5 | 1.3 | 95 | 1.3 | |
| $DQ_{s,in}$ | 95 | 1.3 | 95 | 0.22 | 95 | 0.22 | |
| $IQ_{s,in}\_DQ_{s,in}$ | 95 | 1.3 | 95 | 2.6 | 95 | 1.3 | |
| BLF | 95 | 1.3 | 95 | 1.3 | 95 | 1.3 | |





**Figure 1.** Experimental setup, data collection and analysis. (A) Overview of experimental setup. Sediment supply ($Q_{s,in}$) and water discharge ($Q_w$) can be regulated separately. For all but the base level fall (BLF) experiment, the base level was fixed. Water and sediment fell off of an edge at the outlet. For the BLF experiment (shown in the picture), the base level was controlled through the water level in the surrounding basin. (B) Digital elevation model (DEM) derived from laser scans showing the final topography of the increased water ($IQ_w$) experiment. (C) Overhead photograph of the $IQ_w$ experiment taken directly before the scan shown in B. The surface was covered with a thin layer of red sand before the instant increase in discharge was performed. The remnants of red sand on the terraces indicate no further reworking after the onset of increased discharge. (D) Overhead photographs were turned into binary (wet, dry) images from which the average channel width within the analyzed area (orange frame) can be calculated.




**Figure 2.** Fill terraces formed during experimental runs. Paired terraces were formed in the Increase $Q_w$ ($IQ_w$) experiment and are shown from top (A) and looking in the downstream direction (B) at the end of the experiment (540 min = 300 min after spin-up time). Remnants of red sand on the terrace surfaces indicate that those areas have not been flooded after the instant doubling in discharge. During the base-level fall (*BLF*) experiment, terraces at the downstream end were abandoned instantly after the onset of base level fall (250 min = 10 min after onset of *BLF*). Terraces are shown from above (C) and looking in the downstream direction (D). Those terraces were destroyed shorty after they were cut. A new set of terraces was formed in the upstream part ca. 120 min after the onset of *BLF*.





**Figure 3.** Evolution of cross-sections in the upper part of the reach (left panel). In each cross section, the lowest point is set equal to zero to track incision. The color scheme represents time since the last change in boundary conditions (equivalent to either 240 min or 480 min experiment time). For better comparison, we plot a maximum of 240 minutes for all experiments, despite longer recordings for some of the runs. Exact location of cross sections are indicated by the black lines in the DEMs displaying the last scan of each experiment (right panel). Cross-sections haven been chosen at the terrace midpoints and thus vary slightly between the experiments. The times given in parentheses are the absolute experiment runtimes.






**Figure 4.** Evolution of longitudinal river profiles from minute 240 (end of 'spin-up' phase) onwards. River profiles were extracted from the laser scans. Laser scans were recorded every 30 min, and an additional two scans at 10 and 20 minutes after the initiation of the base-level fall were conducted during the *BLF* experiment. Dashed arrows indicate down-basin distance along which terraces formed. Note that the $DQ_w\_IQ_w$ and $IQ_{s,in}\_DQ_{s,in}$ were split into two panels each, with one panel representing each phase.

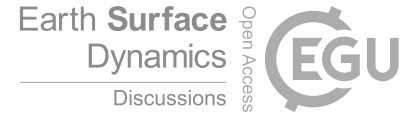

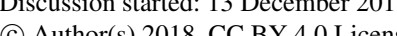




**Figure 5.** Input parameters and evolution of channel slope and channel width during the experiments. Input sediment ($Q_{s,in}$; orange solid line) and water ($Q_w$; blue solid line) discharge were normalized to the reference input values ($Q_{s,ref}$ = 1.3 ml/s and $Q_{w,ref}$ = 95 ml/s). Slope ($S$, grey circles) was calculated based on the bed elevation difference between the inlet and the outlet divided by the length of the system. Channel elevation measurements for slope calculations were performed manually during the runs. Channel width was calculated as the mean number (solid lines) of wet pixels in each of 1200 cross section within the box indicated in Fig. 1C, D. The colored shaded areas around the curves indicate the standard deviation of the 1200 measurements. The evolution of width without any external perturbation (*Ctrl_2*) is plotted for comparison with each other experiment in which external conditions were changed (B-F). Note that no measurements are available for the *Ctrl_1* experiment due to issues with the installation of the overhead camera. Sediment discharge at the outlet ($Q_{s,out}$) during the experimental runs is compared to input sediment ($Q_{s,in}$; orange solid line); both were normalized to reference input values ($Q_{s\_ref}$ = 1.3 ml/s). The first 240 min of each experiment were adjustment to the reference settings (grey box) and were not included in the analyses. Black arrows indicate times when terraces in the upstream part of the sandbox started to be cut.



**Figure 6.** Elevation profile and slope comparison of terrace surfaces and active channels. Elevation profiles are given as mean (solid lines) and minimum and maximum values (dashed lines), extracted along a 5 cm wide swaths as indicated on the right panel. Swath width was reduced in two cases of too narrow terraces to 1 cm ($DQ_w\_IQ_w$ T$_A$ terrace) and 2 cm ($DQ_{s,in}$ T$_A$ terrace). T$_A$ and T$_B$ indicate terraces on one side each and refer to labels of lag-times given in Fig.5. Slopes were calculated based on a linear fit through the mean elevation profiles. Numbers in parentheses give the RMSE between the linear fit and the measured data. For the four experiments in which upstream conditions



were changed (A-D), the slopes of the terraces are steeper than of the active channel at the end of the experiment. In contrast, in the *BLF* experiment, slopes of the terraces and the active channels are about the same. Note the different y-axis for the $IQ_w$ run for better visibility. Colors of elevation in right panel same as in Fig. 3.





**Figure 7.** Schematic model of the evolution of signals at the outlet stored in either sediment volume or the chemical composition of the sediment.