# Peer review of "Alluvial channel response to environmental perturbations: Fill-terrace formation and sediment-signal disruption"

_Earth Surface Dynamics, 2018_

## Referee Comment (RC1) · Malatesta (Referee) · 21 Jan 2019

Review of Tofelde et al. (2019, E Surf D): Alluvial channel response to environmental perturbations: Fill-terrace formation and sediment-signal disruption

Dear editor,

I had the pleasure to carefully read the interesting new manuscript of Tofelde and colleagues. In it, the authors present a set of seven flume experiments documenting the transient response of an alluvial channel to changes in sediment and water fluxes, and in base level. Their observations illustrate the transient behaviour of transport-limited

streams that was previously described in various scattered theoretical, experimental, and field-study articles. It is a useful and timely contribution. However, the text is written in a manner that often suggests that the results of the flume experiments are novel findings while they mostly confirm and support previous work. As such the manuscript is almost a review in disguise. Nevertheless, the experiments led the authors to some valuable novel observations, in particular the lagged response of incision to a drop in input sediment flux. It is particularly valuable in that regard because we precisely lack documentation of transient responses as equations describing fluvial geometry mostly deal with equilibrium conditions. I suggest that the manuscript be accepted after moderate revision.

The figures are of quality and support the text well. The text is also detailed enough such that the reader can grasp all they need to understand the experimental runs.

Below I propose three types of comments to the authors, 1) general comment about fill terraces; 2) a suggestion for the structure of the text; 3) line-by-line comments on science and bibliography.

1. Fill vs. Cut-in-fill terraces: The authors introduce the object "fill-terrace" on page 2 and thereafter it is inferred that all terraces recorded in their flume experiments are such. I would object to this use of the term. A fill terrace, as described on page 2, is a morphologic datum recording the culmination of sediment aggradation immediately preceding a phase of incision and thus abandonment (Howard, 1959; Bull, 1991; Pazzaglia, 2013). In several experimental runs, it seems that the entire active floodplain is being eroded before it narrows its width and starts entrenching, thus abandoning terraces. In that situation, these terraces are not "fill-terraces" but cut-in-fill as they record a moment during the incisional phase and not the culmination of alluvial aggradation. The title of the article needs to be accordingly modified. Then, the difficulty resides in reliably identifying if a given "top" terrace (top as in being the highest from the last incision episode) is indeed a fill terrace. To me it is very interesting that the authors identify cases where barely any fill terraces are abandoned. And that instead two large

cut-in-fill terraces replace the fill terraces one would commonly expect. It appears to capture the moment when vertical incision is promoted over lateral erosion leading to fast autogenic entrenchment of the channel (Malatesta et al., 2017; Bufe et al., 2018) but the two experiments with a drop in Qs suggest that this inflexion point does not always occur at a similar moment. Finally on that point, the rationale behind picking the terraces TA and TB should be fleshed out because at least in the case of the DQsin run, they capture cut-in-fill terraces. More about that with the comment on p. 12 l. 13.

**2. Structure of the manuscript**

I think that a weakness of the current manuscript structure is that it is difficult to understand what the novel advances are and what the narrative of the work is. That is especially true for readers who are familiar with the existing, extensive, body work on alluvial geometry dating starting with Gilbert and Murphy (1914). The results are presented as if they almost provided a first-time observation of such alluvial dynamics. However, most of the observations from the flume experiments have already been observed, predicted, or discussed in previous bodies of work. What is novel is the documentation of the transient response itself. The manuscript could be somewhat modified to make this clearer and better highlight the contribution of the authors to this larger body of work. In that spirit, I would suggest to move elements of the discussion to the review section "2 Formation of fluvial fill terraces" so as to clearly establish what is acquired knowledge and to underline the gap that the authors want to fill here. In particular, section 2.1 could be augmented with large parts of sections "5.1 Channel response to perturbations and conditions of terrace formation" and "5.3 Differences in terrace surface slope". By explicitly introducing the theoretical framework used to describe the relationships between alluvial slope and fluxes of sediment and water (Qs and Qw), the authors would build a better launchpad for their study in my opinion. The Meyer-Peter Müller (MPM) equation revised by Wong and Parker (2006) or more recent derivations of slope as a function of Qs and Qw (e.g. by Malatesta and Lamb, 2017 GSAB, or Wickert and Schildgen, 2019) can help establish clearly what is known so far,

and what is not. The latter being a good understanding of the transient behaviour from one equilibrium configuration to the next. I believe that this modification to the structure of the manuscript would help the reader better navigate the coexistence of the review and experimental aspects of the paper.

3. Science and bibliography comments

p. 1 l. 9-10: This is a pretty strong statement. I would argue that published work provide a pretty good understanding of the impacts of such forcing on terrace formation and sediment dynamics. What is lacking — and provided by the authors here — is rigorous observations of the transient response.

p. 2 l. 27-30: Malatesta, Prancevic and Avouac (2017, JGR) explicitly target lateral feedbacks with a numerical model.

p. 2 l. 31: Limaye and Lamb (2016, JGR) could also be mentioned here as an example of an excellent bedrock model.

p. 3 l. 8-10: I strongly encourage the authors to have a look at the 2003 Geology paper by Bonnet and Crave. Therein the authors investigate the impact of climatic (Qw) vs. tectonic forcing (base level) on an experimental landscape. While not targeting terraces in particular, it is one of the most insightful papers I've read on the subject. I strongly encourage the authors to read through it and incorporate some thoughts in their work.

p. 3 l. 20: "upstream" [and along stream] (to take into account extra Qs from local incision)

p. 4 l. 3: If incision supplies sediment to Qsin along stream, then Qsin is not the input sediment flux. It might be useful to separate Qsin, Qsc (sediment transport capacity at any point along stream), and Qsout.

p. 5 l. 10-13: I understand and appreciate the distinction here, and it is quite useful to separate the two. But is it a new refined definition? It seemed to me that fill-cut terraces are commonly considered both "complex response" and "autogenic" at the same time

(Schumm's work and Pazzaglia's review paper). If you indeed propose this new,useful, distinction here, I would encourage you to take ownership of it.

p. 5 l. 28: There is a new paper by Johnson and Finnegan that is in revision at Geology on "Tributary Channel Transience Triggered by Bedrock River Meander Cutoffs." I don't know when it will come out. But regardless, it might interest you for the future

p. 6 l. 5: As the reference codes of the experiments are going to be used thereafter, I would suggest to make a reference to Table 1 here.

p. 6 l. 16: what is the vertical resolution?

p. 6 l. 29: It could be helpful to mention that water is tainted blue in the photos.

p. 6 l. 32: why can it be considered unaffected?

p. 7 l. 30: I would argue that change in channel width is not required to form fill terraces. What needs to be reduced is the breadth of the active floodplain (in which the channel, of potentially fixed width, migrates left and right).

p. 9 l. 4: The nature of terraces TA and TB could be mentioned here to simplify the reading of the paragraph.

p.9 l. 21-22: Is there a threshold for what constitutes a pair? Is there a way to define that objectively, or at least in a consistently arbitrary way?

p. 10 l. 6-7: Not sure I understand the rationale behind the ratio of vertical and horizontal erosion. A terrace of width W is preserved for a time T with a river lateral erosion Eh such that T=W/Eh. Preservation is independent from the vertical incision rate. However, deep incision will result in higher walls that are costlier to erode.

p. 10 l. 15-19: Field studies such as Tofelde et al. (2018), Malatesta et al. (2017, Basin Research), or, and especially, Dzurisin (1975). More on the latter below.

p. 11 l. 4-5: a comment only valid if the theoretical framework for alluvial rivers is

not beefed up above: I suggest to state that +Qs leads to +S in order to preserve eq. 1 under constant Qw, just as to explain the rationale between Qs and S which is not directly derived from Eq. 1 and 2.

p. 11 l. 8: This dynamic is described and discussed by Malatesta et al. (2017, JGR). It is also worth noting two earlier flume experiments by Schumm et al. [1987, chapter 6] and Meyer et al. [1995] describe the evolution of a channel profile after it reaches a new equilibrium post-incision (see description of that work in section 5.1 in Malatesta et al. 2017, JGR).

p. 12 l. 1: What exactly is the degree of reworking of terrace material? The amount of vertical incision?

p. 12 l. 5: I am a little hung up on paired/unpaired and the threshold it implies. Wouldn't it be more informative to simply write that the terraces are abandoned successively?

p. 12 l. 13: Runs DQsin and IQsin_DQsin both lead to entrenchment when sediment flux drops. So, why does the same forcing cause very different terrace creation, or at least be considered as two different systems? To me, it seems that the different terrace record of the two runs could be explained as reflecting the inherent variability in the abandonment of cut-in-fill terraces. See point about fill terraces written at the begining of the review. It should be however noted that, in the experiment DQsin, there are two slivers of what was probably the original floodplain datum. As such, these slivers should be TA and TB for comparison with IQsin_DQsin.

p. 12 l. 17: this feedback has also been extensively discussed and explored by Malatesta et al. (2017, JGR).

p. 12 l. 19-21: yes, but the two effects mitigate each other. If the incision rate is slow, the later terrace will also not have been lowered that much such that the geometrical difference remains about the same.

p. 13 l. 11-12: The formulation used here suggests that the authors have observed

and established ("we found that") this relationship for the first time, along the 2018 Wickert & Schildgen paper. Yet, the fact that terraces have a steeper gradient than the stream's for Qs or Qw forcing is not a new observation or theoretical construct, it is built-in in theory since early fluvial geomorphology work (Mackin, 1948; Meyer-Peter & Müller, 1948; Léopold & Maddock, 1957; Hooke, 1968; Schumm, 1973; Leopold and Bull, 1979; Wells and Harvey, 1987; Harvey et al., 1999; DeLong et al., 2008; Rohais et al., 2012). Recently Malatesta & Lamb (2018) used a derivation of MPM to constrain alluvial slope as an explicit function of Qs and Qw. This passage is one that inspires my earlier suggestion to provide a more complete overview of current knowledge, in particular in terms of theories of transport and geometry.

p. 13 l. 14: I would also point to the absolutely remarkable site of the Gower Gulch alluvial fan in Death Valley. There, a man-made diversion instantaneously changed the hydrology of the catchment leading to sudden incision of the alluvial channel. Details are found in the work of - Troxel, B.W. (1974, Man■made diversion of Furnace Creek Wash, Zabriskie Point, Death Valley, California: California Geology, v. 27, p. 219–223), - Dzurisin (1975,Channel responses to artificial stream capture, Death Valley, California: Geology, v. 3, p. 309–312, doi:10.1130/0091■7613(1975)3<309 :CR-TASC>2.0.CO;2.), - Snyder & Kammer (2009), - Malatesta & Lamb (2017). [you will find the two 70's papers on Gower Gulch attached hereby]

p. 13 l.30 - p. 14 l. 9: I am not sure that I follow the argument here. When terrace treads are used to quantify tectonic deformation, the gradient of the terrace does not matter as it is always detrended to retrieve local deformation (e.g. from an anticline, Lavé Avouac, 2000). As long as the tread is straight, tectonic deformation can be well-constrained.

p. 14 l. 12-15: this context could be introduced much earlier in the manuscript to better motivate the study.

p. 15 l. 7: It can be noted that this illustrates predictions of laws like MPM whereby no

geometric change at the downstream end of the reach demands that the sediment flux transport capacity does not change either

p. 16 l. 6-7: Wouldn't chemical signals be best transferred during phases of bypass? Or is recycling more important in such phase than during aggradation?

p. 16 l. 18-19: I understand that these are observations from the runs, but I think it would be advisable to add that these "findings" validate existing theories. Though grammatically correct, the word suggests an unwarranted degree of novelty to my ears (non-native english hearing ears, mind you) . That is well known and demonstrated already. The same comment is also valid for point 5 of the conclusion.

I appreciate that some of my suggestions represent a considerable amount of rewriting and thank the authors for considering them. I hope that they will find this review helpful.

Kind regards, Luca Malatesta

Please also note the supplement to this comment:
https://www.earth-surf-dynam-discuss.net/esurf-2018-84/esurf-2018-84-RC1-supplement.zip

---

## Referee Comment (RC2) · Anonymous Referee #2 · 29 Jan 2019

General Comments: This well written paper describes a set of seven flume experiments in a sand box in order to mimic conditions and controls of fill-terrace formation. The main controls explored are changes in water Qw and sediment Qs discharge and changes in base level. The paper gives a nice and consistent description of current terrace formation theories, models and controls. It gives a clear description of the experiments and relates them in a transparent way to current model insights on fluvial dynamics. The derived conclusions are supported by the sand box experimental evidence but the translation to field evidence is not equally well considered and not always supported by evidence (there a quite some constraints related to the physical experiments). The main limitation of this investigation is that all results and relation-

ships found are only valid for a flume sand box system which cannot be linearly scaled up to real world system without some critical considerations and reflections. First of all is the sand box experiment dealing with a relatively short and steep fluvial system with $Q_{w,in} = Q_{w,out}$. The setup resembles, in a qualitative way, more an alluvial fan system than a large mature fluvial system, that are usually studied in the cited terrace studies. Secondly, is the 'fluvial system' studied a braided system only, while many studied and cited terrace systems are thought to be initiated when the fluvial system switched from a braided to (more) meandering state (and back). Finally has the used methodology the issue of reproducibility. If we would repeat the same experiments in the same sand box would we get the same terraces (properties) and results?. This is crucial to know because the laser scanning allows us to measure very small changes (with known uncertainties) but if there is significant other uncertainty ('noise') in the sand box data of a higher magnitude we might be over interpreting the data. As long as we do not know the 'noise' in the experiments we should be reluctant to draw too many conclusions from relative minor changes in elevation. I recommend to address these potential limitations in the discussion in a separate section. Having raised these concerns I do believe the experiments generate an interesting set of criteria and hypotheses that could and should be more rigorously tested on real world systems and be evaluated in numerical models. I will certainly test some of the proposed relationships on existing terrace field evidence and with numerical modelling. I therefore recommend to publish this publication after revisions.

Specific comments: The validity of the results and relationships observed are certainly more valid for fluvial fan type settings where also transport distances are relatively short and gradients are steep and we only observe braided behavior. In such real world systems we actually do observe ifferences in gradients between different fill type terraces. The large and longer fluvial systems are often characterized by almost parallel gradients of preserved terraces. Often terrace formation and preservation is linked to tributaries causing reach specific changes in Qs and Qw, something that has not been evaluated in the experiments. The link between landscape dynamics and $Q_{s,in}$ is

another scaling challenge. Landscapes often display a delay between environmental changes and sediment flux responses. These response lags can be even an order magnitudes larger than the lag-times within the fluvial system itself. This is related to coupling and decoupling of hillslope dynamics to the fluvial system. The autogenic dynamics analysis requires more thought. We can only discard them if they do not occur after longer repeated runs under 'stable' conditions. It seems there is more autogenic dynamics related in the transient response of channel width an aspect in the model results that are not as detailed analyzed as the terrace profiles, surface slopes and signal propagation. I like the prediction that net deposition along the channel leads to the majority of the grains at the outlet being freshly delivered from hillslopes (assuming hillslope coupling). While during incision older material is reworked in the outlet material, potentially yielding older ages (with cosmogenics). In terms of the boundary conditions of the physical experiments I have the following remarks/questions: How realistic is a constant Qs,in input? In reality sediments are released as sediment waves into the fluvial system. How important are the initial conditions? (referring to initial channel and 'spin-up' phase). What is the effect of stopping the experiment for the laser scanning? Doesn't this 'disturb' the experiment. A comparison between two equal runs with and without stopping could answer this issue? If this has been investigated before, please cite the relevant literature on this. You give temporal lags in measured time. How would you scale this up to reality? (see fig 5) A difference between the Qw and Qs experiments compared to the base level change scenarios is the there is far less accommodation space in the upper part for terrace preservation (a narrow steep incision) compared to the downstream section and its response to base level change. Shouldn't this not be included in the impact analysis of perturbations? I fully agree with the statement that simulating long-profile evolution requires an improved understanding of the transient response of channel width. I presume that the Wickert and Schildgen, 2018 relationship between S, Qs ,in and Qw are also only valid for braided sand box systems under transport limited conditions? This also implies uniform 'bedrock' lithology. In reality (all cited real world examples) tectonic stability doesn't exist, nor do

uniform lithologies or transport limited conditions. I am not suggesting to exclude the comparison but be more sensitive of the differences. The view of terraces/floodplains as temporal storage space is a realistic one. The percentage of Qs,in is in temporary storage during experiment in total in time, in Fig 5 could be used to quantify this effect and the possible effect on cosmogenic age.

---

## Referee Comment (RC3) · Anonymous Referee #3 · 1 Feb 2019

Tofelde et al. (2019, E Surf D): Alluvial channel response to environmental perturbations: Fill-terrace formation and sediment-signal disruption Review

January 2019

Dear Editors of Earth Surface Dynamics, Thank you for the opportunity to review for your journal. I have read the manuscript prepared by Tofelde et al., and here I will first comment on my impression of the overall quality and character of the paper, then offer my opinion on acceptance of the manuscript, and finally offer some minor line-by-line suggestions of an editorial nature.

First, I enjoyed reading this well written manuscript. I appreciate that the authors

crafted an accessible background literature review (from the perspective of a non-experimentalist). In their manuscript, Tofelde et al., develop interesting and timely scientific questions and knowledge gaps–what are the responses of alluvial fill terraces to modulation of base level, and changes in upstream water discharge and sediment supply (Qw, Qs respectively)–which they then address using seven experiments. I echo the sentiment of Reviewer 2 that this paper has the ring of a review paper, yet that is not a problem for me, and I actually appreciated the good explanations of current knowledge (theoretical, field, and experimental). I thought the amount of review in the introduction was appropriate to bring a non experimentalist/expert up to speed on the current thinking of how terrace incision-aggradation functions with respect to changes in upstream or downstream (base level) boundary conditions. I thought the figures are well made and that the captions are effective as well.

The results of the seven experiments performed by the authors show there are distinct responses in the slope of pre-perturbation and post-perturbation alluvial surface elevations that are dependent upon the type of forcing mechanism, and the authors document interesting transient behavior of fill terrace, channel elevations/width, and Qs out of the experimental system with time. In experiments with increased Qw or Qs, gradients in the new equilibrium channels decrease significantly compared to the pre upstream perturbation channel gradients. This is a somewhat intuitive, yet interesting result, and one that presumably has the potential to be tested in the sedimentary/geomorphic record. I thought that the rationale for the experiments and the results are thought provoking to those interested in not only morphologic response of alluvial fill terraces to external forcing, but also the implications of their response to external forcing in terms of chemical signatures preserved (or not) in sediment/sedimentary systems (end of Section 5).

The experimental design did not include simulations of increased Qw + Qs, or decreased Qw + Qs, as conceivably might occur/be expected in a natural sedimentary system undergoing upstream changes in boundary conditions. Thus its possible the

results of these experiments (pure perturbations in Qw or Qs) may be difficult to invert from sedimentary records or be more pronounced in experiments than nature. I don't consider this a shortcoming of the manuscript, its just an observation, and perhaps the authors could include a statement about this in the discussion? Other reviewers have suggested ideas to help improve the communication of what results are novel by the restructuring of the literature review and parts of the discussion (e.g. Malatesta's comment #2). I concur that the authors should consider improving the way in which they communicate how to interpret these experimental results in the context of existing theoretical and experimental knowledge.

Recommendation: I recommend that this manuscript ultimately be accepted for publication after the authors implement minor revisions.

Line-by-line comments: P1 L9 suggest "...tectonic histories" rather than "...tectonic conditions"?

P2 L20-21 You may want to specify that (at least for Schaller et al 2004) the methods used to interpret paleo discharge were in part based on cosmogenic nuclide concentrations, not simply the age of terrace formation. Interpretations from those concentrations are in turn subject to assumptions of the systematics of cosmogenic nuclides and sedimentary dynamics.

P3 L10 The following sentence needs to be rewritten: "To our knowledge, there are no experimental studies that systematically compare how fill terraces formed through various mechanisms may differ from one another, or investigate the impacts of terrace formation on downstream sediment discharge."

P5 L33 The end of the second Section (2 Formation of fluvial fill terraces) seems abrupt; would it help to provide one or two statements that help summarize and transition into Section 3 here?

P8 L2 Suggest "channel incision" rather than "river incision"?

P9 L23-26 "When comparing terrace slopes to the active channel slopes (blue lines) at the end of each run, terrace slopes are steeper in all experiments in which upstream conditions (Qw, Qs,in) were changed 25 (Fig. 6 A-D). In contrast, the slopes of the terraces and the active channel in the BLF experiment are similar to each other (Fig. 6E)." This is a really interesting relationship, and one I would not have expected (though I don't often think about these kinds of experiments), but that does seem intuitive. Is this pre-perturbation terrace slope and upstream-downstream boundary condition relationship something that is seen in other experimental studies? In nature? I see your discussion includes some mention of this explicitly, and introduces the active tectonic aspect that unfortunately complicates interpretations and adds non uniqueness to potential interpretations of terrace slope history. Can you predict/offer guidelines for which kind of natural systems your experimental results would be best applied?

P10 L22 add a space after "...Fig 5)."

P15 L30-31 Perhaps cite the figure # again for clarity, for which grey vs. yellow circles relate to this sentence.

P16 L6-8 The last sentence of Section 5 suggests chemical signals may be propagated more efficiently through systems during phases of aggradation, rather than phases of incision when mixing of older stored sediment might overprint the chemical signature of "fresh" hillslope derived sediment. This is interesting... Your statement makes sense, however would it also be fair to say that the chemical signature would be a function of the ratio of the "fresh" to recycled sediment (and obviously the erosion rate upstream)? And that those ratios could vary greatly given different system scales (I'm thinking about the ratio of upstream derived Qs vs excavated volume)? Perhaps this is a tangential idea more suitable for its own paper?!

---

## Author Comment (AC1) · 4 Mar 2019

Response to review by Luca Malatesta:

First, we would like to thank the reviewer for his detailed and constructive review of our submission. We believe that addressing his comments will increase the quality of the manuscript. His two main comments were related to (1) the usage of terminology and (2) the structure of the manuscript. In addition, he provided several line comments related to science and bibliography. While the original review comments are shown in *italics*, our responses are given in regular blue font.

**1. Fill vs. Cut-in-fill terraces:**

*The authors introduce the object "fill-terrace" on page 2 and thereafter it is inferred that all terraces recorded in their flume experiments are such. I would object to this use of the term. A fill terrace, as described on page 2, is a morphologic datum recording the culmination of sediment aggradation immediately preceding a phase of incision and thus abandonment (Howard, 1959; Bull, 1991; Pazzaglia, 2013). In several experimental runs, it seems that the entire active floodplain is being eroded before it narrows its width and starts entrenching, thus abandoning terraces. In that situation, these terraces are not "fill-terraces" but cut-in-fill as they record a moment during the incisional phase and not the culmination of alluvial aggradation. The title of the article needs to be accordingly modified. Then, the difficulty resides in reliably identifying if a given "top" terrace (top as in being the highest from the last incision episode) is indeed a fill terrace. To me it is very interesting that the authors identify cases where barely any fill terraces are abandoned. And that instead two large cut-in-fill terraces replace the fill terraces one would commonly expect. It appears to capture the moment when vertical incision is promoted over lateral erosion leading to fast autogenic entrenchment of the channel (Malatesta et al., 2017; Bufe et al., 2018) but the two experiments with a drop in Qs suggest that this inflexion point does not always occur at a similar moment. Finally on that point, the rationale behind picking the terraces TA and TB should be fleshed out because at least in the case of the DQsin run, they capture cut-in-fill terraces. More about that with the comment on p. 12 l. 13.*

Indeed, the terrace terminology in the literature is rather inconsistent. Often, terraces are subdivided into two main categories: strath and fill (e.g. Howard 1959, Pazzaglia 2013). Fill terraces have been further subdivided into the 'highest' terrace that preserves the original deposited surface and 'lower' terraces with surfaces below the original deposited surface that have been eroded laterally into the fill. While the first type is referred to as 'filltop' (Howard 1959) or just 'fill terrace' (e.g., Bull 1990, Merritts et al., 1994), the second type has been described as 'fill-strath' (Howard 1959), 'cut-terrace' (e.g. Merritts et al., 1994), 'fill-cut terrace' (Mizutani 1998, Bull 1990, Pazzaglia 2013, Malatesta et al. 2017) or 'cut-fill terrace' (e.g., Norton et al., 2015). As such, for simplification, we only referred to fill terraces in general with the aim to include both subtypes. Especially since a distinction between the two subtypes in the field is often not possible without detailed stratigraphic or geochronological analysis.

However, we agree that a distinction of the two fill terraces sub-categories would be helpful to clarify the description and especially several points within the discussion. A distinction of the two subtypes within the experiments can easily be made, as we covered the surface with a thin layer of red sand prior to each perturbation. The preservation of the red sand is a clear indication for no further overwash after the perturbation and as such identifies the first subtype of fill terraces (filltop). Any later formed terraces will consequently be cut-terraces. In a way, we already made the distinction as the filltop terraces are those with a lag-time of 0 min (Fig. 5), while the cut terraces have lag-times of > 0 min. But a formal definition of the two subtypes will clarify the difference. For a better visualization, we will include photos of each terrace section and label the terraces accordingly. For later analysis (e.g. terrace surface slope) we always chose the most extensive terrace surfaces on each side of the river. With this approach we aim to mimic common field approaches.

We think that such a distinction between the two terrace subtypes will also clarify the discussion about the degree of reworking of terrace material. Different techniques can be applied to date terrace surfaces. Most of them, however, include sample collection at the terrace surface or within the upper couple of meters. As this part is often equivalent the active layer of the river bed (the depth range over which gains are actively remobilized and deposited), the lag-time between the onset of perturbation and the abandonment of a surface determines what we referred to as 'reworking of terrace material'. As this point was not clear in the discussion (see reviewer comment p.12 l. 1 below), we will clarify this point.

We also agree that the definition of what constitutes a paired or unpaired terrace is not clear (see reviewer comments p.9 l. 21-22 and p. 12 l. 5). Often, paired or unpaired terraces are distinguished based on height similarities or differences. However, as far as we are aware, there is no common rule where to draw the threshold. Instead, we will follow the reviewer's suggestion and instead refer to the ages/ lag-times of the terrace surfaces and describe successive abandonment instead of referring to 'unpaired' terraces.

Also, we agree that the cut-terraces capture the moment when vertical incision outcompetes lateral erosion. However, we disagree that this process should necessarily be referred to as 'autogenic entrenchment'. In the literature, the term 'autogenic' has been used inconsistently and no one definition of the term seems to exist. For the purpose of this manuscript, we have decided to define the term to include terraces that are formed without any external perturbation (i.e. under constant external boundary conditions). Terraces formed due to meander-bend cutoffs are one example of such autogenic terraces. In contrast, the cut-terraces observed in our experiments, which were formed after a lag-time with respect to the time of perturbation, are clearly linked to the external perturbation and are formed during a transient phase in which the channel adjusts to the new conditions. The transient is observed in a number of measureable quantities such as the channel slope and the discharge of sediment out of the basin. We will suggest a differentiation of the terms "autogenic" and "allogenic" for future use (see comment to p. 5 l. 10-13 below).

**2. Structure of the manuscript**

*I think that a weakness of the current manuscript structure is that it is difficult to understand what the novel advances are and what the narrative of the work is. That is especially true for readers who are familiar with the existing, extensive, body work on alluvial geometry dating starting with Gilbert and Murphy (1914). The results are presented as if they almost provided a first-time observation of such alluvial dynamics. However, most of the observations from the flume experiments have already been observed, predicted, or discussed in previous bodies of work. What is novel is the documentation of the transient response itself. The manuscript could be somewhat modified to make this clearer and better highlight the contribution of the authors to this larger body of work. In that spirit, I would suggest to move elements of the discussion to the review section "2 Formation of fluvial fill terraces" so as to clearly establish what is acquired knowledge and to underline the gap that the authors want to fill here. In particular, section 2.1 could be augmented with large parts of sections "5.1 Channel response to perturbations and conditions of terrace formation" and "5.3 Differences in terrace surface slope". By explicitly introducing the theoretical framework used to describe the relationships between alluvial slope and fluxes of sediment and water (Qs and Qw), the authors would build a better launchpad for their study in my opinion. The Meyer-Peter Müller (MPM) equation revised byWong and Parker (2006) or more recent derivations of slope as a function of Qs and Qw (e.g. by Malatesta and Lamb, 2017 GSAB, or Wickert and Schildgen, 2019) can help establish clearly what is known so far, and what is not. The latter being a good understanding of the transient behaviour from one equilibrium configuration to the next. I believe that this modification to the structure of the manuscript would help the reader better navigate the coexistence of the review and experimental aspects of the paper.*

We apologize for giving the impression that all our observations were novel. This was not our intention. The reason to not include the theoretical framework on channel geometry (relationship between Qs, Qw, S and W) in section 2 and only bring it up during the discussion was to keep the focus of the paper on fluvial terraces. But we agree that it might be better to expand section 2 to better distinguish our experimental results between novel observations and those validating existing theories.

As such, we will follow the reviewer's suggestions of and (1) rearrange Section 2 by including background on channel geometry (moving parts of the sections 5.1 and 5.3 into section 2 and expand it) and (2) rephrase the sentences that implied our observations are novel despite being a confirmation of earlier observations or ideas (e.g. p. 13 l. 11-12, p. 16 l. 18-19). We hope that both of these adjustments will help to better focus our work on the 'transient response of an alluvial channel to external perturbation'.

**3. Science and bibliography comments**

*p. 1 l. 9-10: This is a pretty strong statement. I would argue that published work provide a pretty good understanding of the impacts of such forcing on terrace formation and sediment dynamics. What is lacking and provided by the authors here is rigorous observations of the transient response.*

We agree that the original statement was rather vague and as such could be understood in several ways. Therefore we will adjust the sentence to: "However, we currently lack a systematic understanding of the timescales of terrace formation, the transient channel evolution, and associated sediment storage and release in response to changes in base-level, water discharge, and sediment discharge"

*p. 2 l. 27-30: Malatesta, Prancevic and Avouac (2017, JGR) explicitly target lateral feedbacks with a numerical model.*

We will include this reference.

*p. 2 l. 31: Limaye and Lamb (2016, JGR) could also be mentioned here as an example of an excellent bedrock model.*

We agree that the work of Limaye and Lamb (2016) is an important paper. But as our main focus is on alluvial rivers formed in response to external perturbation, we prefer to not include another bedrock model in the introduction. Please note though, that we cite this paper in the section on autogenic terrace formation, as it particularly focuses on the formation of autogenic terraces (p. 2 l. 13 & p. 5 l. 28).

*p. 3 l. 8-10: I strongly encourage the authors to have a look at the 2003 Geology paper by Bonnet and Crave. Therein the authors investigate the impact of climatic (Qw) vs. tectonic forcing (base level) on an experimental landscape. While not targeting terraces in particular, it is one of the most insightful papers I've read on the subject. I strongly encourage the authors to read through it and incorporate some thoughts in their work.*

Although not investigating terrace formation, their measurements of denudation rates go well along with our Qs,out measurements and we will incorporate their findings within the discussion section on 'Signal propagation and implications for stratigraphy'.

*p. 3 l. 20: "upstream" [and along stream] (to take into account extra Qs from local incision)*
Will be clarified.

*p. 4 l. 3: If incision supplies sediment to Qsin along stream, then Qsin is not the input sediment flux. It might be useful to separate Qsin, Qsc (sediment transport capacity at any point along stream), and Qsout.*

As also suggested by the other two reviewers, our measurements on $Q_{s,in}$ and $Q_{s,out}$ actually allow a quantification of the contribution of upstream supplied sediment ($Q_{s,in}$) and remobilized sediment from within the channel to the total sediment discharge ($Q_{s,out}$). We will include those absolute values to the lowest panels of Fig. 5 and adjust the text accordingly, including a clear differentiation between sediment supplied upstream ($Q_{s,in}$) and sediment remobilized within the bed.

*p. 5 l. 10-13: I understand and appreciate the distinction here, and it is quite useful to separate the two. But is it a new refined definition? It seemed to me that fill-cut terraces are commonly considered both "complex response" and "autogenic" at the same time (Schumm's work and Pazzaglia's review paper). If you indeed propose this new,useful, distinction here, I would encourage you to take ownership of it.*

Please see comment to 1.

*p. 5 l. 28: There is a new paper by Johnson and Finnegan that is in revision at Geology on "Tributary Channel Transience Triggered by Bedrock River Meander Cutoffs." I don't know when it will come out. But regardless, it might interest you for the future.*

Thanks for the suggestion.

*p. 6 l. 5: As the reference codes of the experiments are going to be used thereafter, I would suggest to make a reference to Table 1 here.*

Will be included.

*p. 6 l. 16: what is the vertical resolution?*

Will be included.

*p. 6 l. 29: It could be helpful to mention that water is tainted blue in the photos.*

Will be included.

*p. 6 l. 32: why can it be considered unaffected?*

We agree that this statement was too strong as we cannot 'prove' the upstream part to be unaffected. Instead, we will correct the statement to "we consider this part as least affected by the fixed position at the outlet". The second and more important reason to analyze the upstream part is because the terraces were preferentially formed in this part.

*p. 7 l. 30: I would argue that change in channel width is not required to form fill terraces. What needs to be reduced is the breadth of the active floodplain (in which the channel, of potentially fixed width, migrates left and right).*

We agree and will adjust the text accordingly.

*p. 9 l. 4: The nature of terraces TA and TB could be mentioned here to simplify the reading of the paragraph.*

This section will be adjusted in accordance with the subdivision into the two sub-types of fill terraces. See comment to 1.

*p.9 l. 21-22: Is there a threshold for what constitutes a pair? Is there a way to define that objectively, or at least in a consistently arbitrary way?*

Please see comment to 1.

*p. 10 l. 6-7: Not sure I understand the rationale behind the ratio of vertical and horizontal erosion. A terrace of width W is preserved for a time T with a river lateral erosion Eh such that T=W/Eh. Preservation is independent from the vertical incision rate. However, deep incision will result in higher walls that are costlier to erode.*

What we meant is that vertical incision needs to outcompete lateral erosion to even form terraces. We agree that the term 'preservation' used in the text was misleading. It will be corrected. However, the preservation is in that sense dependent on the vertical incision rate, as faster vertical incision will reduce lateral erosion due to a limited transport capacity of the available water.

*p. 10 l. 15-19: Field studies such as Tofelde et al. (2018), Malatesta et al. (2017, Basin Research), or, and especially, Dzurisin (1975). More on the latter below.*

Yes, we can also refer to the field studies here. Will be included.

*p. 11 l. 4-5: a comment only valid if the theoretical framework for alluvial rivers is not beefed up above: I suggest to state that +Qs leads to +S in order to preserve eq. 1 under constant Qw, just as to explain the rationale between Qs and S which is not directly derived from Eq. 1 and 2.*

The theoretical framework will be included in section 2. See response to 1.

*p. 11 l. 8: This dynamic is described and discussed by Malatesta et al. (2017, JGR). It is also worth noting two earlier flume experiments by Schumm et al. [1987, chapter 6] and Meyer et al. [1995] describe the evolution of a channel profile after it reaches a new equilibrium post-incision (see description of that work in section 5.1 in Malatesta et al. 2017, JGR).*

We agree that it is a good idea to compare our observations to other studies that have observed a channel widening after the slope has adjusted to its new equilibrium profile. In the three mentioned studies, widening is related to the input of sediment from the walls or the channel bed. We will expand our discussion to include those studies.

*p. 12 l. 1: What exactly is the degree of reworking of terrace material? The amount of vertical incision?*

Please see comment to 1.

*p. 12 l. 5: I am a little hung up on paired/unpaired and the threshold it implies. Wouldn't it be more informative to simply write that the terraces are abandoned successively?*

Please see comment to 1.

*p. 12 l. 13: Runs DQsin and IQsin_DQsin both lead to entrenchment when sediment flux drops. So, why does the same forcing cause very different terrace creation, or at least be considered as two different systems? To me, it seems that the different terrace record of the two runs could be explained as reflecting the inherent variability in the abandonment of cut-in-fill terraces. See point about fill terraces written at the beginning of the review. It should be however noted that, in the experiment DQsin, there are*

*two slivers of what was probably the original floodplain datum. As such, these slivers should be TA and TB for comparison with IQsin_DQsin.*

The discussion about this point will be adjusted as we will distinguish between the two different types of fill terraces (see response to 1). We agree that the different responses to the same forcing probably indicate the inherent variability.

*p. 12 l. 17: this feedback has also been extensively discussed and explored by Malatesta et al. (2017, JGR).* We will add this reference.

*p. 12 l. 19-21: yes, but the two effects mitigate each other. If the incision rate is slow, the later terrace will also not have been lowered that much such that the geometrical difference remains about the same.* The sentence refers to the time when the switch from dominantly lateral erosion to dominantly vertical incision happens. The earlier the switch, the better the preservation of the initial profile. When the channel continues to planate laterally, it lowers the entire bed surface and when rapid incision initiates, the cut-fill terrace has a lower slope than the channel at the onset of the perturbation. Given the observations we make (Fig. 6), lateral erosion and incision do not seem to completely trade-off so as to keep the geometry constant as suggested by the comment. Instead, we see a good preservation of profiles in cases of instant incision (very low lag-times), compared to lower channel profiles in cases with longer lag-times.

*p. 13 l. 11-12: The formulation used here suggests that the authors have observed and established ("we found that") this relationship for the first time, along the 2018 Wickert & Schildgen paper. Yet, the fact that terraces have a steeper gradient than the stream's for Qs or Qw forcing is not a new observation or theoretical construct, it is built-in in theory since early fluvial geomorphology work (Mackin, 1948; Meyer-Peter & Müller, 1948; Léopold & Maddock, 1957; Hooke, 1968; Schumm, 1973; Leopold and Bull, 1979; Wells and Harvey, 1987; Harvey et al., 1999; DeLong et al., 2008; Rohais et al., 2012). Recently Malatesta & Lamb (2018) used a derivation of MPM to constrain alluvial slope as an explicit function of Qs and Qw. This passage is one that inspires my earlier suggestion to provide a more complete overview of current knowledge, in particular in terms of theories of transport and geometry.* As already stated in our response to comment 2, we will expand section 2 in include information on sediment transport and channel geometry. Also, we will carefully rephrase the sentences that were pointed out as misleading.

*p. 13 l. 14: I would also point to the absolutely remarkable site of the Gower Gulch alluvial fan in Death Valley. There, a man-made diversion instantaneously changed the hydrology of the catchment leading to sudden incision of the alluvial channel. Details are found in the work of - Troxel, B.W. (1974, Man made diversion of Furnace Creek Wash, Zabriskie Point, Death Valley, California: California Geology, v. 27, p. 219– 223), - Dzurisin (1975,Channel responses to artificial stream capture, Death Valley, California: Geology, v. 3, p. 309–312, doi:10.1130/0091 7613(1975)3<309 :CRTASC> 2.0.CO;2.), - Snyder & Kammer (2009), - Malatesta & Lamb (2017). [you will find the two 70's papers on Gower Gulch attached hereby]* We will carefully study the suggested manuscripts and implement the findings of that work.

*p. 13 l.30 - p. 14 l. 9: I am not sure that I follow the argument here. When terrace treads are used to quantify tectonic deformation, the gradient of the terrace does not matter as it is always detrended to retrieve local deformation (e.g. from an anticline, Lavé Avouac, 2000). As long as the tread is straight, tectonic deformation can be well-constrained.* We think the confusion is between using deformation of the tread of a single terrace and using slope differences between different terraces to reconstruct tectonic deformation rates. We agree that we have not clearly differentiated between the two in the manuscript. We will clarify the differences and adjust the text as well as the references accordingly.

*p. 14 l. 12-15: this context could be introduced much earlier in the manuscript to better motivate the study.*

This part will be moved to section 2.

*p. 15 l. 7: It can be noted that this illustrates predictions of laws like MPM whereby no geometric change at the downstream end of the reach demands that the sediment flux transport capacity does not change either.*

Unfortunately, we do not follow the comment of the reviewer. The sentence refers to changes in upstream sediment supply and the according adjustment of the channel reach. Although the base level at the downstream end is fixed, changed in upstream sediment supply do result in changes of channel geometry, i.e. slope and width of the channel reach.

*p. 16 l. 6-7: Wouldn't chemical signals be best transferred during phases of bypass? Or is recycling more important in such phase than during aggradation?*

We would expect that recycling due to lateral movement plays a greater role during bypass that during an aggradation event. Bypass, in the sense of no net deposition or erosion because the channel is in equilibrium, does not exclude the mixing of older and younger material during lateral movement.

*p. 16 l. 18-19: I understand that these are observations from the runs, but I think it would be advisable to add that these "findings" validate existing theories. Though grammatically correct, the word suggests an unwarranted degree of novelty to my ears (non-native english hearing ears, mind you) . That is well known and demonstrated already. The same comment is also valid for point 5 of the conclusion.*

We will rephrase the sentence accordingly (see reply to 2.).

---

## Author Comment (AC3) · 4 Mar 2019

Response to Reviewer #3:

First, we would like to thank reviewer #3 for taking the time to review our submission. Reviewer #3 provided general, as well as line-by-line comments, which we will address below. While the original review comments are shown in *italics*, our responses are given in regular blue font.

*First, I enjoyed reading this well written manuscript. I appreciate that the authors crafted an accessible background literature review (from the perspective of a nonexperimentalist). In their manuscript, Tofelde et al., develop interesting and timely scientific questions and knowledge gaps–what are the responses of alluvial fill terraces to modulation of base level, and changes in upstream water discharge and sediment supply ($Q_w$, $Q_s$ respectively)–which they then address using seven experiments. I echo the sentiment of Reviewer 2 that this paper has the ring of a review paper, yet that is not a problem for me, and I actually appreciated the good explanations of current knowledge (theoretical, field, and experimental). I thought the amount of review in the introduction was appropriate to bring a non experimentalist/expert up to speed on the current thinking of how terrace incision-aggradation functions with respect to changes in upstream or downstream (base level) boundary conditions. I thought the figures are well made and that the captions are effective as well.*
Thanks for this kind assessment.

*The results of the seven experiments performed by the authors show there are distinct responses in the slope of pre-perturbation and post-perturbation alluvial surface elevations that are dependent upon the type of forcing mechanism, and the authors document interesting transient behavior of fill terrace, channel elevations/width, and Qs out of the experimental system with time. In experiments with increased Qw or Qs, gradients in the new equilibrium channels decrease significantly compared to the pre upstream perturbation channel gradients. This is a somewhat intuitive, yet interesting result, and one that presumably has the potential to be tested in the sedimentary/geomorphic record. I thought that the rationale for the experiments and the results are thought provoking to those interested in not only morphologic response of alluvial fill terraces to external forcing, but also the implications of their response to external forcing in terms of chemical signatures preserved (or not) in sediment/sedimentary systems (end of Section 5).*

*The experimental design did not include simulations of increased Qw + Qs, or decreased Qw + Qs, as conceivably might occur/be expected in a natural sedimentary system undergoing upstream changes in boundary conditions. Thus its possible the C2 results of these experiments (pure perturbations in Qw or Qs) may be difficult to invert from sedimentary records or be more pronounced in experiments than nature. I don't consider this a shortcoming of the manuscript, it's just an observation, and perhaps the authors could include a statement about this in the discussion?*
We agree that changes in environmental conditions (e.g. tectonics, climate) that have the potential to affect either $Q_s$ or $Q_w$ are likely to affect both in reality. For example, a change to wetter conditions (increase in $Q_w$) might also trigger a pulse of sediment release from the hillslopes to the channels (e.g. Steffen et. al (2009, 2010)). Thus, considering the entire sediment routing system, $Q_s$ and $Q_w$ are often coupled. With our experimental set-up, however, we only investigate the response of the transfer sub-system to changes in surrounding conditions, and we de-couple $Q_s$ and $Q_w$ to investigate the potential effect that each of those two parameters can have on the evolution of channel morphology. Also, although both parameters are thought to vary simultaneously, thick fluvial fills and fill terrace formation in the field are often related to either significant changes in either $Q_s$ or $Q_w$ (hillslope-driven and discharge-driven models as described in Scherler et al. (2015); see p.3 l.27 to p.4 l.4). As such, we investigate the

two end-members of those models. Many variations in-between those endmembers are possible though. For clarification, we will include the above mentioned points within the manuscript.

*Other reviewers have suggested ideas to help improve the communication of what results are novel by the restructuring of the literature review and parts of the discussion (e.g. Malatesta's comment #2). I concur that the authors should consider improving the way in which they communicate how to interpret these experimental results in the context of existing theoretical and experimental knowledge.*
We agree with both reviewers that the introduction on the theoretical background should be extended. Please see our reply to Malatesta's comments for details on how we intend to adjust the section on background knowledge.

*Recommendation: I recommend that this manuscript ultimately be accepted for publication after the authors implement minor revisions.*

**Line-by-line comments:**
*P1 L9 suggest "...tectonic histories" rather than "...tectonic conditions"?*
We appreciate this suggestion, however, we prefer to stick to the term 'tectonic conditions' for the following reason: Terraces form under certain environmental conditions. As such, the terraces can be used to reconstruct those certain conditions that persisted at a certain point in time. They are not a continuous archive (as for example a varved lake core would be). Therefore, fill terraces cannot be used to infer entire climatic or tectonic histories.

*P2 L20-21 You may want to specify that (at least for Schaller et al 2004) the methods used to interpret paleo discharge were in part based on cosmogenic nuclide concentrations, not simply the age of terrace formation. Interpretations from those concentrations are in turn subject to assumptions of the systematics of cosmogenic nuclides and sedimentary dynamics.*
The main point about this sentence was to state that fill-terrace deposits have been used in various ways, including for example the reconstruction of paleo-discharge or paleo-denudation rates. Schaller et al. (2004) did not reconstruct discharge, but paleo-denudations rates. If we explained the Schaller work in detail, we would also need to explain the other applied approaches, which would not benefit the purpose of the sentence. As such, we prefer to leave the sentence as it is.

*P3 L10 The following sentence needs to be rewritten: "To our knowledge, there are no experimental studies that systematically compare how fill terraces formed through various mechanisms may differ from one another, or investigate the impacts of terrace formation on downstream sediment discharge."*
We hope that the following adjustment of the sentence will help to clarify its structure: "To our knowledge, there are no experimental studies that systematically (i) compare differences in fill terrace geometry, location, and formation timescales that occur for various formation mechanisms, or (ii) investigate the impacts of terrace formation on downstream sediment discharge."

*P5 L33 The end of the second Section (2 Formation of fluvial fill terraces) seems abrupt; would it help to provide one or two statements that help summarize and transition into Section 3 here?*
For a better transition, we will add the following sentence at the end of section 2: "In summary, different processes within the sediment-routing system, including variability in sediment supply, water discharge, base-level changes, autogenic processes, or complex channel responses to a single allogenic perturbation, could lead to similar geomorphic responses of the alluvial river - the formation of fluvial fill terraces."

*P8 L2 Suggest "channel incision" rather than "river incision"?*
OK, will be changed.

*P9 L23-26 "When comparing terrace slopes to the active channel slopes (blue lines) at the end of each run, terrace slopes are steeper in all experiments in which upstream conditions (Qw, Qs,in) were changed 25 (Fig. 6 A-D). In contrast, the slopes of the terraces and the active channel in the BLF experiment are similar to each other (Fig. 6E)." This is a really interesting relationship, and one I would not have expected (though I don't often think about these kinds of experiments), but that does seem intuitive. Is this pre-perturbation terrace slope and upstream-downstream boundary condition relationship something that is seen in other experimental studies? In nature? I see your discussion includes some mention of this explicitly, and introduces the active tectonic aspect that unfortunately complicates interpretations and adds non uniqueness to potential interpretations of terrace slope history. Can you predict/offer guidelines for which kind of natural systems your experimental results would be best applied?*
Variability in terrace slopes has been reported from field studies (e.g., Tofelde at el. (2017), Baker and Gosse (2009), Burgette et al. (2017), Poisson and Avouac (2004)). As such, if (1) base-level changes, autogenic processes and complex responses can be ruled out as the driving formation mechanism of those terraces and if (2) the terrace surfaces have not been tectonically deformed, then those fill terrace slopes can potentially be used to infer paleo $Q_s$ and $Q_w$. The idea of paleo-hydrology inferred from terrace surfaces has already proposed by Leopold and Miller (1954). With modern techniques, however, we hope to improve the possibilities for more quantitative assessments of $Q_w$ and $Q_s$ of the present and the past. We will include this potential application of the findings within the discussion.

*P10 L22 add a space after "...Fig 5)."*
OK.

*P15 L30-31 Perhaps cite the figure # again for clarity, for which grey vs. yellow circles relate to this sentence.*
Will be done.

*P16 L6-8 The last sentence of Section 5 suggests chemical signals may be propagated more efficiently through systems during phases of aggradation, rather than phases of incision when mixing of older stored sediment might overprint the chemical signature of "fresh" hillslope derived sediment. This is interesting...Your statement makes sense, however would it also be fair to say that the chemical signature would be a function of the ratio of the "fresh" to recycled sediment (and obviously the erosion rate upstream)? And that those ratios could vary greatly given different system scales (I'm thinking about the ratio of upstream derived Qs vs excavated volume)? Perhaps this is a tangential idea more suitable for its own paper?!*
We appreciate this suggestion to consider absolute ratios between upstream supplied and remobilized sediment. As we have the absolute input and output $Q_s$ values, we can generate such a plot, which we will include in the lowest panel of Fig. 5. We will adapt the text of the manuscript accordingly and include the point that the degree of signal modification is dependent on the ratio of fresh vs. recycled sediments.

---

## Author Response (AR1)

Dear Sébastien Castelltort,

We submit a revised version of our manuscript entitled '*Alluvial channel response to environmental perturbations: Fill-terrace formation and sediment-signal disruption*' for consideration for publication in *Earth Surface Dynamics*. First, we would like to apologize for our delay in submitting the revised version. We appreciated the reviews and the efforts made to help improve our manuscript and we heavily modified the manuscript even beyond the suggestions of the reviewers. Therefore, before replying to the reviewer's comments in detail below, we would like to summarize the main modifications to the manuscript.

Reviewer #1 suggested to better emphasize the main focus of the manuscript, as the differentiation between novel observations and confirmation of earlier ideas was not clear. Initially, the manuscript had a strong focus on fluvial terraces only. However, we think that the major strength of our experimental setup is the opportunity to track the evolution of two different records of landscape evolution simultaneously – namely (1) fluvial-fill terraces in the transfer zone and (2) sediment discharge ($Q_{s,out}$) to the deposition zone. We have thus reoriented the scope of the manuscript in this direction. The main modifications involve:

- A strong reorientation of the introduction. We included information on the general behavior of alluvial channels as well as background knowledge on variability in $Q_{s,out}$. To balance this, we have removed the extensive background section on fluvial-fill terraces (former chapter 2) and included the most important information in the introduction.
- For clarification and a better visualization we included two new figures: figure 1 and 9. Figure 1 is a conceptual summary of sediment-routing systems (modified after Castelltort and van den Driessche, 2003), the important parameters that shape the transfer zone as well as the two landscape-evolution records that we investigate in our experiments. Figure 9 is a conceptual summary of our observations.
- With the modified focus of the paper, we also strongly modified the structure and (partly) the content of the discussion section. Within the new structure, we discusse (1) fill-terrace records; (2) $Q_{s,out}$ records; (3) what can be learned about the coupling of the two; (4) the limitations of our experimental approach, especially when compared to natural settings (following the main concerns of reviewer #2); and (5) implications of our observations for future field studies.

In addition to the changes listed above or stated below within the detailed responses to the reviewers, we performed slight modifications to the figures, captions or to the wording of the main text. These modifications were either only stylistic for increased precision and clarity, or are discussed in the detailed responses below.

Yours sincerely,

Steffi Tofelde and co-authors

**Response to review by Luca Malatesta:**

First, we would like to thank Luca Malatesta for his detailed and constructive review of our submission. We believe that addressing his thoughtful comments has increased the quality of the manuscript. Luca Malatesta's two main comments were related to (1) the usage of terminology and (2) the structure of the manuscript. In addition, he provided several line-by-line comments related to science and bibliography. While the original review comments are shown in *italics*, our responses are given in regular, blue font. Our line numbers refer to the newly submitted version of the manuscript.

*1. Fill vs. Cut-in-fill terraces:*
*The authors introduce the object "fill-terrace" on page 2 and thereafter it is inferred that all terraces recorded in their flume experiments are such. I would object to this use of the term. A fill terrace, as described on page 2, is a morphologic datum recording the culmination of sediment aggradation immediately preceding a phase of incision and thus abandonment (Howard, 1959; Bull, 1991; Pazzaglia, 2013). In several experimental runs, it seems that the entire active floodplain is being eroded before it narrows its width and starts entrenching, thus abandoning terraces. In that situation, these terraces are not "fill-terraces" but cut-in-fill as they record a moment during the incisional phase and not the culmination of alluvial aggradation. The title of the article needs to be accordingly modified. Then, the difficulty resides in reliably identifying if a given "top" terrace (top as in being the highest from the last incision episode) is indeed a fill terrace. To me it is very interesting that the authors identify cases where barely any fill terraces are abandoned. And that instead two large cut-in-fill terraces replace the fill terraces one would commonly expect. It appears to capture the moment when vertical incision is promoted over lateral erosion leading to fast autogenic entrenchment of the channel (Malatesta et al., 2017; Bufe et al., 2018) but the two experiments with a drop in Qs suggest that this inflexion point does not always occur at a similar moment. Finally on that point, the rationale behind picking the terraces TA and TB should be fleshed out because at least in the case of the DQsin run, they capture cut-in-fill terraces. More about that with the comment on p. 12 l. 13.*

Indeed, the terrace terminology in the literature is rather inconsistent. Often, terraces are subdivided into two main categories: strath and fill (e.g. Howard 1959, Pazzaglia 2013). Fill terraces have been further subdivided into the 'highest' terrace that preserves the original deposited surface and 'lower' terraces with surfaces below the original deposited surface that have been eroded laterally into the fill. While the first type is referred to as 'filltop' (Howard 1959) or just 'fill terrace' (e.g., Bull 1990, Merritts et al., 1994), the second type has been described as 'fill-strath' (Howard 1959), 'cut-terrace' (e.g. Merritts et al., 1994), 'fill-cut terrace' (Mizutani 1998, Bull 1990, Pazzaglia 2013, Malatesta et al. 2017) or 'cut-fill terrace' (e.g., Norton et al., 2015). As such, for simplification, we only referred to fill terraces in general with the aim to include both subtypes. Especially since a distinction between the two subtypes in the field is often not possible without detailed stratigraphic or geochronological analysis.

However, we agree that a distinction of the two fill terraces sub-categories would be helpful to clarify the terrace description, and also the discussion on lag-times between perturbation and terrace cutting. A distinction of the two subtypes within the experiments can easily be made, as we covered the surface with a thin layer of red sand prior to each perturbation. The preservation of the red sand is a clear indication for no further overwash after the perturbation and, as such, identifies the first subtype of fill terraces (filltop). Any later formed terraces will consequently be fill-cut terraces. Therefore, we added a formal definition of the two subtypes to the introduction (l. 72-73) and the methodological distinction to the methods section (l. 162-164). For a better visualization, we modified Fig. 3 (former Fig. 2) and included photos of each terrace section with the according labels of terrace type and lag-times. We chose, however,

to keep the original title as the general term 'fill terraces' covers all types. For later analysis (e.g. terrace surface slope) we always chose the most extensive terrace surfaces on each side of the river. With this approach we aim to mimic common field approaches. We clarified this in the results section (l. 235-236).

We also agree that the definition of what constitutes a paired or unpaired terrace is not clear (see reviewer comments p.9 l. 21-22 and p. 12 l. 5). Often, paired or unpaired terraces are distinguished based on height similarities or differences. However, as far as we are aware, there is no common rule where to draw the threshold. Instead, we followed the reviewer's suggestion and instead referred to the ages/ lag-times of the terrace surfaces and describe successive abandonment instead of referring to 'unpaired' terraces. All former statements of paired/unpaired terraces have been removed.

Also, we agree that the cut-terraces capture the moment when vertical incision outcompetes lateral erosion. However, we disagree that this process should necessarily be referred to as 'autogenic entrenchment'. In the literature, the term 'autogenic' has been used inconsistently and no one definition of the term seems to exist. In the new version of the manuscript, we have decided to reduce this topic to the following statement: "In some cases, internal dynamics of the system, sometimes referred to as "autogenic processes", lead to terrace formation that cannot be directly linked to external forcing (e.g., Erkens et al., 2009; Limaye and Lamb, 2016; Malatesta et al., 2017; Patton and Schumm, 1981; Womack and Schumm, 1977) ."

**2. Structure of the manuscript**
*I think that a weakness of the current manuscript structure is that it is difficult to understand what the novel advances are and what the narrative of the work is. That is especially true for readers who are familiar with the existing, extensive, body work on alluvial geometry dating starting with Gilbert and Murphy (1914). The results are presented as if they almost provided a first-time observation of such alluvial dynamics. However, most of the observations from the flume experiments have already been observed, predicted, or discussed in previous bodies of work. What is novel is the documentation of the transient response itself. The manuscript could be somewhat modified to make this clearer and better highlight the contribution of the authors to this larger body of work. In that spirit, I would suggest to move elements of the discussion to the review section "2 Formation of fluvial fill terraces" so as to clearly establish what is acquired knowledge and to underline the gap that the authors want to fill here. In particular, section 2.1 could be augmented with large parts of sections "5.1 Channel response to perturbations and conditions of terrace formation" and "5.3 Differences in terrace surface slope". By explicitly introducing the theoretical framework used to describe the relationships between alluvial slope and fluxes of sediment and water (Qs and Qw), the authors would build a better launchpad for their study in my opinion. The Meyer-Peter Müller (MPM) equation revised byWong and Parker (2006) or more recent derivations of slope as a function of Qs and Qw (e.g. by Malatesta and Lamb, 2017 GSAB, or Wickert and Schildgen, 2019) can help establish clearly what is known so far, and what is not. The latter being a good understanding of the transient behaviour from one equilibrium configuration to the next. I believe that this modification to the structure of the manuscript would help the reader better navigate the coexistence of the review and experimental aspects of the paper.*

We apologize for giving the impression that all our observations were novel. This was not our intention. We followed the suggestion of the reviewer to better elaborate the main focus of the paper. We think that the strength of the paper is the tracking of the simultaneous evolution of two different records of landscape evolution – fluvial fill terraces in the transfer zone and sediment deposition rate ($\sim O_{s,out}$) in the deposition zone. The majority of the changes of the manuscript relate to better structure the manuscript around this point. Following adjustments were made:
- We significantly rewrote the introduction to better lead to the point of comparing the two records.

- Within the new introduction, we follow the reviewer's suggestion and extend the background on channel geometry and sediment transport. To do so, we have moved parts of the former discussion (section 5.3) to the introduction, and we have also included an explanation of the relationship between the geometrical adjustment, bedload transport and transport capacity after Meyer-Peter Müller (1948), as was both suggested by reviewer #1. In line with these modifications, we have moved the most important information on terrace formation (former section 2) to the introduction and removed section 2 instead.
- We also added a new overview figure 1 that summarized the source-to-sink framework as well as the important parameters for our study.
- Within the results, we slightly rearranged the order by moving the description of terrace abandonment to the beginning and by distinguishing between fill-top and fill-cut terraces (l. 182-201).
- With the more defined focus of the paper, we basically discuss for each of the two records (fill-terraces and $Q_{s,out}$/sedimentation rate) (1) how the information stored in them can be modified and (2) how different forcings can lead to a similar stratigraphy (ambiguity) (new discussion sections 4.1 and 4.2). Therefore, we decided to remove the discussion part on channel width evolution, as this point is interesting, but does not directly contribute to our main focus anymore. Instead, we are currently working on a manuscript that addresses this point separately. We enhanced the discussion on why we observed different lag-times between perturbation and terrace surface abandonment for the different forcing mechanism (l. 290-325).
- We also expanded the discussion on $Q_{s,out}$ by including a paragraph about the times of $Q_s$ signal modification (new section 4.2.1), as well as an entire section on the combination of the two records (new section 4.3), and included a new figure 9 to summarize these points.
- Last, we adjusted the conclusion by focusing more on the coupling of the two records.
- Also, we carefully rephrased the sentences that implied our observations are novel despite being a confirmation of earlier observations or ideas (e.g. p. 13 l. 11-12, p. 16 l. 18-19), if they have not been removed from the new version.

**3. Science and bibliography comments**

*p. 1 l. 9-10: This is a pretty strong statement. I would argue that published work provide a pretty good understanding of the impacts of such forcing on terrace formation and sediment dynamics. What is lacking and provided by the authors here is rigorous observations of the transient response.*
We agree that the original statement was rather vague and as such could be understood in several ways. Therefore we adjusted the sentence to: "However, we currently lack a systematic and quantitative understanding of the transient evolution of fluvial systems and their associated sediment storage and release in response to changes in base level, water input, and sediment input."

*p. 2 l. 27-30: Malatesta, Prancevic and Avouac (2017, JGR) explicitly target lateral feedbacks with a numerical model.*
We did include the reference.

*p. 2 l. 31: Limaye and Lamb (2016, JGR) could also be mentioned here as an example of an excellent bedrock model.*
We agree that the work of Limaye and Lamb (2016) is an important paper. The sentence the reviewer refers to has been removed from the modified introduction. Please note though, that we cite this paper in the section on autogenic terrace formation, as it particularly focuses on the formation of autogenic terraces.

*p. 3 l. 8-10: I strongly encourage the authors to have a look at the 2003 Geology paper by Bonnet and Crave. Therein the authors investigate the impact of climatic (Qw) vs. tectonic forcing (base level) on an experimental landscape. While not targeting terraces in particular, it is one of the most insightful papers I've read on the subject. I strongly encourage the authors to read through it and incorporate some thoughts in their work.*

We thank the reviewer for this suggestions. We have implemented the paper in the introduction (l. 79, 82 and 84) as well as in the discussion (section 4.2.2, l. 367, 376 and 389).

*p. 3 l. 20: "upstream" [and along stream] (to take into account extra Qs from local incision)*

This sentence has been removed.

*p. 4 l. 3: If incision supplies sediment to Qsin along stream, then Qsin is not the input sediment flux. It might be useful to separate Qsin, Qsc (sediment transport capacity at any point along stream), and Qsout.*

We agree that this point was confusing. Thus, we have now separated sediment flux into: $Q_{s,in}$ (=entrance to the transfer zone), $Q_s$ (=sediment flux at any point within the transfer zone) and $Q_{s,out}$ (=sediment discharge at the outlet of the transfer zone). We have added the three parameters to our new overview figure 1. Also, we have adjusted the text accordingly.

*p. 5 l. 10-13: I understand and appreciate the distinction here, and it is quite useful to separate the two. But is it a new refined definition? It seemed to me that fill-cut terraces are commonly considered both "complex response" and "autogenic" at the same time (Schumm's work and Pazzaglia's review paper). If you indeed propose this new,useful, distinction here, I would encourage you to take ownership of it.*

The section on autogenic terraces has been reduced to the following statement (l. 68-70) "In some cases, internal dynamics of the system, sometimes referred to as "autogenic processes", lead to terrace formation that cannot be directly linked to external forcing (e.g., Erkens et al., 2009; Limaye and Lamb, 2016; Malatesta et al., 2017; Patton and Schumm, 1981; Womack and Schumm, 1977)."

*p. 5 l. 28: There is a new paper by Johnson and Finnegan that is in revision at Geology on "Tributary Channel Transience Triggered by Bedrock River Meander Cutoffs." I don't know when it will come out. But regardless, it might interest you for the future.*

We thank the reviewer for this suggestion.

*p. 6 l. 5: As the reference codes of the experiments are going to be used thereafter, I would suggest to make a reference to Table 1 here.*

We included it.

*p. 6 l. 16: what is the vertical resolution?*

We included the information on the vertical resolution (1 mm).

*p. 6 l. 29: It could be helpful to mention that water is tainted blue in the photos.*

We added a sentence within the methods section explaining that the water was dyed blue (l. 141-142).

*p. 6 l. 32: why can it be considered unaffected?*

We agree that this statement was too strong as we cannot 'prove' the upstream part to be unaffected. Instead, we changed the statement to "we considered this sector of the channel to be least unaffected by the fixed location of the outlet." The second and more important reason to analyze the upstream part is because the terraces were preferentially formed in this part.

*p. 7 l. 30: I would argue that change in channel width is not required to form fill terraces. What needs to be reduced is the breadth of the active floodplain (in which the channel, of potentially fixed width, migrates left and right).*

We agree with the reviewer. We have changed the sentence to 'Fill-terrace formation requires changes in the channel-bed elevation and width of the active floodplain'.

*p. 9 l. 4: The nature of terraces TA and TB could be mentioned here to simplify the reading of the paragraph.*
We moved the description of the terraces to the beginning of the results section (in accordance with the new details on all terraces in the updated Fig. 3) and included an explaining sentence regarding $T_L$ (former TA) and $T_R$ (former TB) at its very beginning (l. 174-175).

*p.9 l. 21-22: Is there a threshold for what constitutes a pair? Is there a way to define that objectively, or at least in a consistently arbitrary way?*
We removed the terms 'paired' and 'unpaired' as no clear definition exists. Instead, we refer to the terraces based on their lag-times. Please also see our comment to 1.

*p. 10 l. 6-7: Not sure I understand the rationale behind the ratio of vertical and horizontal erosion. A terrace of width W is preserved for a time T with a river lateral erosion Eh such that T=W/Eh. Preservation is independent from the vertical incision rate. However, deep incision will result in higher walls that are costlier to erode.*
What we meant is that vertical incision needs to outcompete lateral erosion to even form terraces. We agree that the term 'preservation' used in the text was misleading. To avoid further confusion, we changed the sentence to "The cutting of fluvial-fill terraces requires vertical incision and a simultaneous reduction of the active floodplain width."

*p. 10 l. 15-19: Field studies such as Tofelde et al. (2018), Malatesta et al. (2017, Basin Research), or, and especially, Dzurisin (1975). More on the latter below.*
In this paragraph, we discuss the evolution of the longitudinal channel profiles and how our results relate to other physical or numerical studies that applied the same or similar forcings. The difference between physical and numerical model studies compared to field studies is that in physical and numerical modelling studies the input forcing parameters are known, such that the resulting profiles can be directly related to the forcing mechanism. For field studies, however, the longitudinal profiles can be reconstructed from terraces, but the main driver can only be inferred, but not known. Therefore, we prefer not to compare our results to field studies at this point.

*p. 11 l. 4-5: a comment only valid if the theoretical framework for alluvial rivers is not beefed up above: I suggest to state that +Qs leads to +S in order to preserve eq. 1 under constant Qw, just as to explain the rationale between Qs and S which is not directly derived from Eq. 1 and 2.*
The theoretical framework is now included in in the introduction and this part of the discussion has been removed.

*p. 11 l. 8: This dynamic is described and discussed by Malatesta et al. (2017, JGR). It is also worth noting two earlier flume experiments by Schumm et al. [1987, chapter 6] and Meyer et al. [1995] describe the evolution of a channel profile after it reaches a new equilibrium post-incision (see description of that work in section 5.1 in Malatesta et al. 2017, JGR).*
With the new scope of the paper, we have decided to remove the detailed discussion on channel width changes as it does not contribute to the adjusted focus of the paper.

*p. 12 l. 1: What exactly is the degree of reworking of terrace material? The amount of vertical incision?*
We consider the degree of reworking rather as the time the river still reworks the active layer before the terrace surfaces get abandoned and the sediment 'trapped'. For clarification, we changed the sentence to: "The lag time between an external perturbation and the onset of terrace cutting determines how much time the fluvial system has to modify the terrace sediments before their abandonment."

*p. 12 l. 5: I am a little hung up on paired/unpaired and the threshold it implies. Wouldn't it be more informative to simply write that the terraces are abandoned successively?*
We have removed the terms 'paired' and 'unpaired' entirely. Please see comment to 1 for details.

*p. 12 l. 13: Runs DQsin and IQsin_DQsin both lead to entrenchment when sediment flux drops. So, why does the same forcing cause very different terrace creation, or at least be considered as two different systems? To me, it seems that the different terrace record of the two runs could be explained as reflecting the inherent variability in the abandonment of cut-in-fill terraces. See point about fill terraces written at the beginning of the review. It should be however noted that, in the experiment DQsin, there are two slivers of what was probably the original floodplain datum. As such, these slivers should be TA and TB for comparison with IQsin_DQsin.*
We clarified in the manuscript that the long profiles and lag-times shown in Fig. 7 and 6, respectively, were extracted from the most extensive terraces surface to resemble common field approaches. We expanded the discussion on incision rates, by relating incision rates to excess transport capacity following Wickert & Schildgen (2019). We included a potential explanation why one of the experiments, in which we reduce $Q_{s,in}$ behaves differently than the other (l.313 -316): "However, while one of the two experiments with a reduction in $Q_{s,in}$ ($DQ_{s,in}$) is consistent with this theory (Fig. 6D), in the other one ($\underline{IQ}_{s,in}$ _ $DQ_{s,in}$), we observed relatively short lag times (Fig. 6E). These unexpectedly short lag times might be related to how the incision phase was preceded by an aggradation phase (due to an increase in $Q_{s,in}$). Possibly, the system rapidly settled back to the initial conditions because it had not completely adjusted to the preceding increase in $Q_{s,in}$."

*p. 12 l. 17: this feedback has also been extensively discussed and explored by Malatesta et al. (2017, JGR).*
We implemented this reference (l. 302).

*p. 12 l. 19-21: yes, but the two effects mitigate each other. If the incision rate is slow, the later terrace will also not have been lowered that much such that the geometrical difference remains about the same.*
The sentence refers to the time when the switch from dominantly lateral erosion to dominantly vertical incision happens. The earlier the switch, the better the preservation of the initial profile. When the channel continues to planate laterally, it lowers the entire bed surface and when rapid incision initiates, the cut-fill terrace has a lower slope than the channel at the onset of the perturbation. Given the observations we make (Fig. 7), lateral erosion and incision do not seem to completely trade-off so as to keep the geometry constant as suggested by the comment. Instead, we see a good preservation of profiles in cases of instant incision (very low lag-times), compared to lower channel profiles in cases with longer lag-times.

*p. 13 l. 11-12: The formulation used here suggests that the authors have observed and established ("we found that") this relationship for the first time, along the 2018 Wickert & Schildgen paper. Yet, the fact that terraces have a steeper gradient than the stream's for Qs or Qw forcing is not a new observation or theoretical construct, it is built-in in theory since early fluvial geomorphology work (Mackin, 1948; Meyer-Peter & Müller, 1948; Léopold & Maddock, 1957; Hooke, 1968; Schumm, 1973; Leopold and Bull, 1979; Wells and Harvey, 1987; Harvey et al., 1999; DeLong et al., 2008; Rohais et al., 2012). Recently Malatesta & Lamb (2018) used a derivation of MPM to constrain alluvial slope as an explicit function of Qs and Qw. This passage is one that inspires my earlier suggestion to provide a more complete overview of current knowledge, in particular in terms of theories of transport and geometry.*
We did not intend to pretend that we observed those relationships for the first time. We have removed the expression "we found that" entirely from the manuscript. And we expanded the discussion by comparing our observations to predictions from theory (l. 352-355), as well as with the numerical model results (Lane, 1955; Mackin, 1948; Malatesta and Lamb, 2018; Meyer-Peter and Müller, 1948; Wickert and Schildgen, 2019; Wobus et al., 2010). In addition, we added another field example (Pepin et al., 2013) to the one that was already included (Poisson and Avouac, 2004). Taken all those adjustments together, we are convinced that we do not create an impression of novelty about this point anymore.

*p. 13 l. 14: I would also point to the absolutely remarkable site of the Gower Gulch alluvial fan in Death Valley. There, a man-made diversion instantaneously changed the hydrology of the catchment leading to sudden incision of the alluvial channel. Details are found in the work of - Troxel, B.W. (1974, Man made diversion of Furnace Creek Wash, Zabriskie Point, Death Valley, California: California Geology, v. 27, p. 219– 223), - Dzurisin (1975,Channel responses to artificial stream capture, Death Valley, California: Geology, v. 3, p. 309–312, doi:10.1130/0091 7613(1975)3<309 :CRTASC> 2.0.CO;2.), - Snyder & Kammer (2009), - Malatesta & Lamb (2017). [you will find the two 70's papers on Gower Gulch attached hereby]*
We thank the reviewer for this suggestions. However, instead of artificial river capture, we decided to implement another natural example of terraces with reduced slopes due to climatic changes by Pepin et al. (2013) from the southern Central Andes as well as the numerical model exercise by Wobus et al. (2010). Both of their observations are in agreement with our experimental results as well.

*p. 13 l.30 - p. 14 l. 9: I am not sure that I follow the argument here. When terrace treads are used to quantify tectonic deformation, the gradient of the terrace does not matter as it is always detrended to retrieve local deformation (e.g. from an anticline, Lavé Avouac, 2000). As long as the tread is straight, tectonic deformation can be well-constrained.*
We agree that this paragraph was a little confusing. We have clarified that the slope changes after upstream perturbations ($Q_w$, $Q_{s,in}$) mainly affect the approach, in which incision rates (and thus uplift rates) are inferred from terrace height-age plots. As incision is higher at the upstream end after upstream perturbation, the terrace height varies along the profile. We have adjusted the text and references accordingly (l. 494-500).

*p. 14 l. 12-15: this context could be introduced much earlier in the manuscript to better motivate the study.*
This part has been moved to the introduction.

*p. 15 l. 7: It can be noted that this illustrates predictions of laws like MPM whereby no geometric change at the downstream end of the reach demands that the sediment flux transport capacity does not change either.*
Unfortunately, we do not follow the comment of the reviewer. The sentence refers to changes in upstream sediment supply and the according adjustment of the channel reach. Although the base level at the downstream end is fixed, changed in upstream sediment supply do result in changes of channel geometry, i.e. slope and width of the channel reach.

*p. 16 l. 6-7: Wouldn't chemical signals be best transferred during phases of bypass? Or is recycling more important in such phase than during aggradation?*
We would expect that recycling due to lateral movement plays a greater role during bypass that during an aggradation event. Bypass, in the sense of no net deposition or erosion because the channel is in equilibrium, does not exclude the mixing of older and younger material during lateral movement. However, we have added another sentence stating that the degree of signal modification is a function of the mixing- ratio of fresh and remobilized material (l. 516-518).

*p. 16 l. 18-19: I understand that these are observations from the runs, but I think it would be advisable to add that these "findings" validate existing theories. Though grammatically correct, the word suggests an unwarranted degree of novelty to my ears (non-native english hearing ears, mind you). That is well known and demonstrated already. The same comment is also valid for point 5 of the conclusion.*
This part of the conclusion has been rewritten and the sentence, the reviewer refers to, was deleted.

**Response to Reviewer #2:**

First, we would like to thank reviewer #2 for the feedback on our manuscript. Reviewer #2 provided general as well as specific comments which we will address below. While the original review comments are shown in italics, our responses are given in regular font. Our line numbers refer to the newly submitted version of the manuscript.

*General Comments:*

*This well written paper describes a set of seven flume experiments in a sand box in order to mimic conditions and controls of fill-terrace formation. The main controls explored are changes in water Qw and sediment Qs discharge and changes in base level. The paper gives a nice and consistent description of current terrace formation theories, models and controls. It gives a clear description of the experiments and relates them in a transparent way to current model insights on fluvial dynamics. The derived conclusions are supported by the sand box experimental evidence but the translation to field evidence is not equally well considered and not always supported by evidence (there a quite some constraints related to the physical experiments). The main limitation of this investigation is that all results and relationships found are only valid for a flume sand box system which cannot be linearly scaled up to real world system without some critical considerations and reflections.*

*First of all is the sand box experiment dealing with a relatively short and steep fluvial system with Qw,in = Qw,out. The setup resembles, in a qualitative way, more an alluvial fan system than a large mature fluvial system that are usually studied in the cited terrace studies.*

First, we thank the reviewer for the positive feedback on our work.

We further acknowledge that upscaling is common problem when transferring experimental results to natural settings. Therefore, we included a new section within the discussion that addresses the different limitations of our experimental setup (new section 4.4). In this section we discuss the following limitations:

(1) We investigate the transfer zone separated from the erosion zone, such that any natural coupling between hillslope processes and channel activity is not included.
(2) We treat $Q_{s,in}$ and $Q_w$ as two independent parameters, although they are known to be coupled in natural systems.
(3) We only investigate a single, braided channel and can therefore make no statements about tributary – main stem interactions or terraces forming in meandering rivers.
(4) We have geomorphically effective flow 100% of the time. As natural rivers have variable flow conditions, including times of no geomorphic activity, a direct comparison of lag times or response times is complicated.
(5) Discussion on number of experiments and reproducibility.

We agree that the channel system is relatively short and steep, but includes the fundamental feedbacks: water and sediment inputs, base level, and a channel that responds to these forcings. And we do believe that our setup differs from an alluvial fan setting because it has a narrow, defined outlet, which ensures that the river stays within a confined valley. Typical processes observed on alluvial fans during aggradation are gradual channel migration and avulsion (sudden changes in channel position), which results in an overall widening of the actively reworked alluvial fan area in downstream direction. This can be seen for example in experimental setups from Whipple et al. (1998), Kim et al. (2006a,b) and Martin et al. (2009). In our experimental setup, however, the confined outlet forces the river to stay 'in place' and limits avulsions. In addition, alluvial fans are often characterized by superelevation, i.e. elevation in the central part can be higher compared to the fan margins. As our main purpose is to study process-behavior

of a system, we assume that the preservation of processes (e.g., dominance of lateral migration over avulsion) is more important than the absolute scaling of the slope. The slope of the river is a function of the sediment supply and water discharge. As such, we could have chosen a gentler slope of the river. The reason for the stepper references slopes was to produce pronounced differences in channel geometry within all the different settings.

Also, the results of the experiment are qualitatively similar to those of the numerical alluvial channel simulations of Wickert & Schildgen (2019).

*Secondly, is the 'fluvial system' studied a braided system only, while many studied and cited terrace systems are thought to be initiated when the fluvial system switched from a braided to (more) meandering state (and back).*

We indeed cite field studies of terraces that were formed within braided as well as within meandering channel systems. The purpose of the introduction is to give an overall overview of the different processes of terrace formation.

However, a large number of the studies cited refer to terraces that were formed in braided channels system only. These studies include for example: Scherler et al. (2015), Schildgen et al. (2016), Tofelde et al. (2017), Norton et al. (2015), Faulkner et al. (2016), Fuller et al. (1998), Malatesta et al. (2018), Malatesta and Avouac (2018), Bookhagen et al. (2016), McPhillips et al. (2014), Dey et al. (2016), Steffen et al. (2009), Steffen et al. (2010) and Litty et al. (2016).

Nevertheless, we agree that our experimental approach is restricted to terrace formation in braided systems only. For clarification, we added an extra paragraph to the discussion stating that our setup restricts us to only investigate terraces that form parallel to the main stem and that we cannot investigate any terrace formation at the junctions between the main stem and tributaries or terraces that form due to meander cut-off (l. 449 - 458). We also clarified in the abstract and introduction, that our transfer zone is represented by "a single braided channel in non-cohesive sediment".

*Finally has the used methodology the issue of reproducibility. If we would repeat the same experiments in the same sand box would we get the same terraces (properties) and results? This is crucial to know because the laser scanning allows us to measure very small changes (with known uncertainties) but if there is significant other uncertainty ('noise') in the sand box data of a higher magnitude we might be over interpreting the data. As long as we do not know the 'noise' in the experiments we should be reluctant to draw too many conclusions from relative minor changes in elevation. I recommend to address these potential limitations in the discussion in a separate section.*

This is a good point and we agree that reproducibility is crucial. In the set of experiments contained within this manuscript we only repeated the control experiment (*Ctrl_1 and Ctrl_2*). The purpose of the control experiments was to investigate 'noise' within the system. We only interpret changes in morphology that are beyond the variability within the control experiments as externally driven adjustments.

Although we did not repeat the experiments that included external perturbations with exactly the same settings, we consider the last phase of the two experiments during which we performed two changes ($DQ_w\_IQ_w$ and $IQ_{s,in}\_DQ_{s,in}$) as repetition of the experiments with only one perturbation ($IQ_w$ and $DQ_{s,in}$), although with different absolute values of $Q_w$ and $Q_{s,in}$. The comparison of those experiments with each other reveals that the trajectories of channel evolution (longitudinal profiles, slope, width (Fig. 5 and 6)) is robust. In addition, the final $Q_s$ and $Q_w$ settings of the experiments with two changes ($DQ_w\_IQ_w$ and $IQ_{s,in}\_DQ_{s,in}$) were equal to the reference settings (*Ctrl_1, Ctrl_2* and 'spin-up' time setting of all experiments but *BLF*). When comparing the slope values to which all those sub-experiments evolve, the final slope values are very similar (around 0.07). Although not being exact repetitions of the same

experiments, the evolution to the same equilibrium conditions is an indicator that the results are reproducible.

However, we agree that despite the apparent repeatability based on two different experiments, our number of repeat experiments is very limited. We therefore included a new paragraph to the discussion elaborating on these points (l. 464-473).

*Having raised these concerns I do believe the experiments generate an interesting set of criteria and hypotheses that could and should be more rigorously tested on real world systems and be evaluated in numerical models. I will certainly test some of the proposed relationships on existing terrace field evidence and with numerical modelling. I therefore recommend to publish this publication after revisions.*

Thank you.

**Specific comments:**

*The validity of the results and relationships observed are certainly more valid for fluvial fan type settings where also transport distances are relatively short and gradients are steep and we only observe braided behavior. In such real world systems we actually do observe differences in gradients between different fill type terraces. The large and longer fluvial systems are often characterized by almost parallel gradients of preserved terraces. Often terrace formation and preservation is linked to tributaries causing reach specific changes in Qs and Qw, something that has not been evaluated in the experiments.*

As already discussed above, we think that our setting is not entirely representative of an alluvial fan system due to the confined outlet, the absence of superelevation and the dominance of lateral migration over avulsion. In real world systems, terraces along the main stem can be parallel to the active channel (e.g., Hanson et al., 2006; Faulkner et al., 2016), but they can also vary in gradient (e.g., Tofelde et al., 2017; Poisson and Avouac, 2004; Baker and Gosse, 2009; Burgette et al., 2017). As terrace sequences along the main stem can be up to tens of kilometers in extent, changes in slope might not be so obvious locally and can only bet determined by detailed surface elevation surveys of those terraces.

We agree that many terraces are preserved at confluences of tributary channels and the main stem. Within this set of experiments, we only focus on terraces that form along the main stem to keep the setting as simple as possible and investigate the direct effects of changes in $Q_s$, $Q_w$ or base-level on changes in bed elevation and terrace formation. Adding a tributary channel adds another level of complexity due to possible internal feedback mechanism between the main stem and the tributary. We also have performed experiments in which we focus on the interaction of a tributary and the main stem. This work is currently in preparation. We think that including another set of experiments, with a detailed focus on tributary-main stem interactions, would overload this manuscript and also draw the focus in a different direction. However, for clarification, we will add an explaining sentence that this study only investigates terrace formation along the main stem.

*The link between landscape dynamics and Qs,in is another scaling challenge. Landscapes often display a delay between environmental changes and sediment flux responses. These response lags can be even an order magnitudes larger than the lag-times within the fluvial system itself. This is related to coupling and decoupling of hillslope dynamics to the fluvial system.*

We agree that changes in sediment supply from the hillslopes to the channels can lag behind any changes in environmental conditions that might cause an adjustment of the supply rate. In this study, we only investigate the response of the fluvial part (=transfer zone) to variations in input conditions, and we do not have the ability to address lag times between environmental forcing and hillslope responses (erosion zone). Following pioneers like Stanley Schumm and Philipp Allen, we consider a sedimentary source-to-sink system as systems that can be subdivided into three zones – the erosion zone, the transfer zone and the deposition zone. Each of those zones has its own responses and response timescales to external

perturbations. We only investigate the transfer sub-system of a source-to-sink sediment transport system. The transfer sub-system connects the erosion zone (hillslopes) with the final deposition zone (e.g. a terrestrial or marine basin). As such, we only investigate response or lag-times of the transfer sub-system and do not investigate delays between sediment supply from hillslopes to river channels. Although we have stated this in the original manuscript (p. 2 l. 2, p.2 l. 17-19, p. 6 l. 4 of original version), we clarify this in the introduction by introducing the new figure 1, as well as by stating that we only investigate Qs-modifications within the transfer zone (e.g., l. 110-114, 529-530 and many more).

In additions, we include a paragraph in the discussion stating that we cannot investigate the potential coupling between the hillslopes (erosion zone) and channel (transfer zone) with our setup (l. 436-441).

*The autogenic dynamics analysis requires more thought. We can only discard them if they do not occur after longer repeated runs under 'stable' conditions. It seems there is more autogenic dynamics related in the transient response of channel width, an aspect in the model results that are not as detailed analyzed as the terrace profiles, surface slopes and signal propagation.*

We apologize, but this seems to be a misunderstanding. We do not discard autogenic terrace formation. On p. 11 l. 3-10 (original version) we state that we did not observe any autogenic terrace formation after the 'spin-up' time, but that the absence of such terraces does not mean that autogenic terraces do not exist. We also state that most mechanisms of autogenic terrace formation, could not be tested with our experimental setup. This part of the discussion has been moved to the section 4.4 (Limitation of experiments) and states:

"…the lack of terrace formation in the two control experiments after the 'spin-up' time does not imply that autogenic terraces do not exist in natural systems, because several potential mechanisms of autogenic or complex-response terrace formation like meander-bend cut-off (Erkens et al., 2009; Gonzalez, 2001; Limaye and Lamb, 2016; Womack and Schumm, 1977) or internal feedbacks between the main-stem and tributaries (Schumm, 1979, 1973, Gardener 1983, Schumm and Parker 1973, Slingerland and Snow 1988) could not be tested with our experimental set-up.

*I like the prediction that net deposition along the channel leads to the majority of the grains at the outlet being freshly delivered from hillslopes (assuming hillslope coupling). While during incision older material is reworked in the outlet material, potentially yielding older ages (with cosmogenics).*

Thank you.

*In terms of the boundary conditions of the physical experiments I have the following remarks/questions: How realistic is a constant Qs,in input? In reality sediments are released as sediment waves into the fluvial system.*

We agree that sediment supply from hillslopes to the channel (erosion zone to transfer zone) can be highly variable. The further downstream transport of the sediment in the river, however, is then limited by the availability of water. As alluvial rivers are limited by their transport capacity and not by the availability of sediment, we consider the $Q_{s,in}$ for a given channel reach within the transfer zone as less variable compared to sediment supply from hillslopes to the channel itself. For clarification, we have adjusted the text such that our experiments only investigate the geomorphic response of the transfer zone of a source-to-sink system (see comment above). The constant water discharge prescribed in the experiments is also a difference to natural channels that are dominated by variable discharges. In a way, we are 'compressing time' and assume that the experiments integrate over a number of large floods in natural channels; therefore, the timescales cannot be scaled directly (see new paragraph in section 4.4., l. 459-463).

*How important are the initial conditions? (referring to initial channel and 'spin-up' phase).*

We assume that the initial conditions play a minor role as we only look at changes in the system once the system is close to equilibrium. If the initial conditions were different, we expect the time to reach steady conditions to be longer or shorter (depending on the initial conditions). The two experiments during which we performed two changes ($IQ_s\_DQ_s$ and $DQ_w\_IQ_w$), both result at the initial slope value after the conditions have been changed back to reference conditions (Fig. 6C, E). As such, we expect the initial conditions mainly affect the 'spin-up' time required to reach stable conditions.

*What is the effect of stopping the experiment for the laser scanning? Doesn't this 'disturb' the experiment? A comparison between two equal runs with and without stopping could answer this issue? If this has been investigated before, please cite the relevant literature on this.*

Unfortunately, it is not possible to scan the surface without stopping the experiment for two reasons: (1) The laser scanner is mounted directly above the setting and it scans the surface in five lines parallel to the flow direction. Those five lines largely overlap and are merged after finishing the scans. The scanning of all five lines requires about 5 min. A continuation of the experiment would alter the surface morphology during the scanning time, such that the overlapping parts could not be merged anymore. (2) The water supplied to the experiments is dyed blue (Fig. 2). The reason is to enable the automatic detection of wet and dry pixels from the overhead photos. For the automatic detection, significant color differences between the water and the surrounding sand is necessary. The laser scanner, however, cannot penetrate the dyed water. As such, the experiments have to be stopped to be able to scan the surface topography. For the two reasons listed above, a comparison as suggested by the reviewer is unfortunately not possible.

However, the experiments have also been stopped overnight. In those cases, a laser scan was performed after stopping the experiment in the evening and before starting it again in the morning. The DEM of difference (DOD) between those scans reveal no major changes in topography for example through drying of the surface and collapse of channel banks. Finally, the time to drain the system took only a few minutes, and therefore does not leave a lot of time for significant reworking of the surface. Our approach is common for these type of experiment, and so far, there is no indication in the literature that it causes significant problems.

*You give temporal lags in measured time. How would you scale this up to reality? (see fig 5)*

As already mentioned above (see new paragraph in section 4.4., l. 459-463), our experiments are simple in a way that sediment supply and water discharge are constant through time, such that we assume that the experiments integrate over a number of large floods in natural channels. This makes an absolute scaling of channel response time and lag-times between perturbation and terrace abandonment complicated. Rather, we see the advantage of our approach that we can observe the form of the response (e.g. decrease in slope follows and exponential pattern and not a linear one). As such, we can differentiate whether a terrace was abandoned instantly after the onset of perturbation or rather later during the transient channel response phase.

*A difference between the Qw and Qs experiments compared to the base level change scenarios is the there is far less accommodation space in the upper part for terrace preservation (a narrow steep incision) compared to the downstream section and its response to base level change. Shouldn't this not be included in the impact analysis of perturbations?*

This is an interesting point. We agree that if the channel widens downstream (as channels tend to do in real systems), there is indeed more "space" to accommodate terraces downstream than upstream. That might be reflected in the width of terraces formed upstream and downstream. Because of the fixed outlet, we have a limited capacity of the system to widen downstream, and therefore are not sure we can make a strong statement about downstream changes in accommodation space with our setup.

*I fully agree with the statement that simulating long-profile evolution requires an improved understanding of the transient response of channel width. I presume that the Wickert and Schildgen, 2018 relationship between S, Qs ,in and Qw are also only valid for braided sand box systems under transport limited conditions?*

Wickert and Schildgen (2019) derive a general set of equations for gravel-bed river long-profile evolution -- meaning that flows are bedload-dominated and lack bedforms. They also note that transient width response is a needed direction of future research, and limit their approach to the assumption that such channels will tend to have a near-equilibrium width (e.g., Parker, 1978), which is appropriate for gradual changes discharge or other drivers of width change. This equilibrium width is set such that the Shields stress against the bank is equal to the critical Shields stress for initiation of motion, which is also appropriate for experiments such as ours, in which the banks are not held together by cohesive forces. For further questions on this study, we refer the reviewer to the final (2019) published paper.

Also, the detailed discussion of channel width evolution has been removed from the revised version of the paper.

*This also implies uniform 'bedrock' lithology. In reality (all cited real world examples) tectonic stability doesn't exist, nor do uniform lithologies or transport limited conditions. I am not suggesting to exclude the comparison but be more sensitive of the differences.*

In our experimental setup, we only study alluvial rivers. Therefore, uniform bedrock lithologies are not of major concern compared to studies of bedrock channels, in which a lowering in slopes requires the erosion of bedrock, which indeed is influenced by lithology. In our case, the material that needs to be moved is sediment, and we consider its lithology of minor importance.

As already noted above, we make the assumption that the system is always in 'transport-limited' conditions. The same conditions characterize some of the cited real world examples. Several of the cited field studies refer to braided, alluvial rivers in mountain basins that are characterized by massive alluvial fills (Tofelde et al. (2017), Schildgen et al. (2016), Dey et al. (2016), Malatesta et al. (2018), Malatesta and Avouac (2018), Scherler et al. (2015), Huntington (1907), Litty et al. (2016), Steffen et al. (2009, 2010)). In those settings, the amount of sediment that is transported out of the basin is restricted by the transport capacity of the river. As such, we consider those sites to be in transport-limited conditions. Concerning tectonic stability, we agree that it is unlikely to be maintained over very long time periods, but even over the millennial timescales that many alluvial features are formed, it is not uncommon to find areas where there is no substantial change in tectonic forcing.

*The view of terraces/floodplains as temporal storage space is a realistic one. The percentage of Qs,in is in temporary storage during experiment in total in time, in Fig 5 could be used to quantify this effect and the possible effect on cosmogenic age.*

Thanks for the suggestion. Indeed, as the absolute $Q_{s,in}$ values are known, we can quantify the percentage of sediment discharge ($Q_{s,out}$) that has been supplied from upstream ($Q_{s,in}$) and that has been remobilized from within the channel. In figure 6 (bottom panel) we plot $Q_{s,in}$ and $Q_{s,out}$, both normalized to the reference value of 1.29 ml/s (as stated in the methods). As such, the numbers read from the y-axis multiplied by 1.29 give the absolute volumes of $Q_{s,in}$ and $Q_{s,out}$ for each point in time. As the values are all normalized by the same value (1.29), the ratio of $Q_{s,out}$ and $Q_{s,in}$ tells us how much sediment has been remobilized within the channel compared to $Q_{s,in}$. For example in the $IQ_w$ experiment, the $Q_{s,out}$ increases to about 20 right after the doubling in discharge, while the $Q_{s,in}$ stays at 1. Consequently, 20 times as much sediment has been remobilized from the transfer zone compared to the upstream supply.

**Response to Reviewer #3:**

First, we would like to thank reviewer #3 for taking the time to review our submission. Reviewer #3 provided general, as well as line-by-line comments, which we will address below. While the original review comments are shown in *italics*, our responses are given in regular blue font. Our line numbers refer to the newly submitted version of the manuscript.

*First, I enjoyed reading this well written manuscript. I appreciate that the authors crafted an accessible background literature review (from the perspective of a nonexperimentalist). In their manuscript, Tofelde et al., develop interesting and timely scientific questions and knowledge gaps–what are the responses of alluvial fill terraces to modulation of base level, and changes in upstream water discharge and sediment supply (Qw, Qs respectively)–which they then address using seven experiments. I echo the sentiment of Reviewer 2 that this paper has the ring of a review paper, yet that is not a problem for me, and I actually appreciated the good explanations of current knowledge (theoretical, field, and experimental). I thought the amount of review in the introduction was appropriate to bring a non experimentalist/expert up to speed on the current thinking of how terrace incision-aggradation functions with respect to changes in upstream or downstream (base level) boundary conditions. I thought the figures are well made and that the captions are effective as well.*
Thanks for this kind assessment.

*The results of the seven experiments performed by the authors show there are distinct responses in the slope of pre-perturbation and post-perturbation alluvial surface elevations that are dependent upon the type of forcing mechanism, and the authors document interesting transient behavior of fill terrace, channel elevations/width, and Qs out of the experimental system with time. In experiments with increased Qw or Qs, gradients in the new equilibrium channels decrease significantly compared to the pre upstream perturbation channel gradients. This is a somewhat intuitive, yet interesting result, and one that presumably has the potential to be tested in the sedimentary/geomorphic record. I thought that the rationale for the experiments and the results are thought provoking to those interested in not only morphologic response of alluvial fill terraces to external forcing, but also the implications of their response to external forcing in terms of chemical signatures preserved (or not) in sediment/sedimentary systems (end of Section 5).*

*The experimental design did not include simulations of increased Qw + Qs, or decreased Qw + Qs, as conceivably might occur/be expected in a natural sedimentary system undergoing upstream changes in boundary conditions. Thus its possible the C2 results of these experiments (pure perturbations in Qw or Qs) may be difficult to invert from sedimentary records or be more pronounced in experiments than nature. I don't consider this a shortcoming of the manuscript, it's just an observation, and perhaps the authors could include a statement about this in the discussion?*
We agree that changes in environmental conditions (e.g. tectonics, climate) that have the potential to affect either $Q_s$ or $Q_w$ are likely to affect both in reality. For example, a change to wetter conditions (increase in $Q_w$) might also trigger a pulse of sediment release from the hillslopes to the channels (e.g. Steffen et. al (2009, 2010)). Thus, considering the entire sediment routing system, $Q_s$ and $Q_w$ are often coupled. With our experimental set-up, however, we only investigate the response of the transfer sub-system to changes in surrounding conditions, and we de-couple $Q_s$ and $Q_w$ to investigate the potential effect that each of those two parameters can have on the evolution of channel morphology. Also, although both parameters are thought to vary simultaneously, thick fluvial fills and fill terrace formation in the field are often related to either significant changes in either $Q_s$ or $Q_w$ (hillslope-driven and dischargedriven models as described in Scherler et al. (2015); see p.3 l.27 to p.4 l.4 of the original version). As such, we investigate the two end-members of those models. Many variations in-between those endmembers are possible though. For clarification, we included the above mentioned points within the discussion section on 'Limitations of experiments' (l. 442-448).

*Other reviewers have suggested ideas to help improve the communication of what results are novel by the restructuring of the literature review and parts of the discussion (e.g. Malatesta's comment #2). I concur that the authors should consider improving the way in which they communicate how to interpret these experimental results in the context of existing theoretical and experimental knowledge.*
We agree with both reviewers that the introduction on the theoretical background should be extended. Please see our reply to Malatesta's comments for details on how we intend to adjust the section on background knowledge.

*Recommendation: I recommend that this manuscript ultimately be accepted for publication after the authors implement minor revisions.*

**Line-by-line comments:**
*P1 L9 suggest "...tectonic histories" rather than "...tectonic conditions"?*
We appreciate this suggestion, however, we prefer to maintain the term 'tectonic conditions' for the following reason: Terraces form under certain environmental conditions. As such, the terraces can be used to reconstruct those certain conditions that persisted at a certain point in time. They are not a continuous archive (as for example a varved lake core would be). Therefore, fill terraces cannot be used to infer entire climatic or tectonic histories.

*P2 L20-21 You may want to specify that (at least for Schaller et al 2004) the methods used to interpret paleo discharge were in part based on cosmogenic nuclide concentrations, not simply the age of terrace formation. Interpretations from those concentrations are in turn subject to assumptions of the systematics of cosmogenic nuclides and sedimentary dynamics.*
The main point about this sentence was to state that fill-terrace deposits have been used in various ways, including for example the reconstruction of paleo-discharge or paleo-denudation rates. Schaller et al. (2004) did not reconstruct discharge, but paleo-denudations rates. If we explained the Schaller work in detail, we would also need to explain the other applied approaches, which would not benefit the purpose of the sentence. As such, we prefer to not extend the explanation. Please also note that this sentence has been moved and slightly rearranged within the new structure of the introduction.

*P3 L10 The following sentence needs to be rewritten: "To our knowledge, there are no experimental studies that systematically compare how fill terraces formed through various mechanisms may differ from one another, or investigate the impacts of terrace formation on downstream sediment discharge."*
With the new structure of the introduction, the sentence has been changed to "To our knowledge, there are no experimental studies that consider the combined evolution of two records of landscape evolution – fill terraces in the transfer zone and sediment discharge to the deposition zone – in response to environmental perturbations."

*P5 L33 The end of the second Section (2 Formation of fluvial fill terraces) seems abrupt; would it help to provide one or two statements that help summarize and transition into Section 3 here?*
Section 2 has been removed from the new version of the manuscript.

*P8 L2 Suggest "channel incision" rather than "river incision"?*
OK, was changed.

*P9 L23-26 "When comparing terrace slopes to the active channel slopes (blue lines) at the end of each run, terrace slopes are steeper in all experiments in which upstream conditions (Qw, Qs,in) were changed 25 (Fig. 6 A-D). In contrast, the slopes of the terraces and the active channel in the BLF experiment are similar to each other (Fig. 6E)." This is a really interesting relationship, and one I would not have expected (though I don't often think about these kinds of experiments), but that does seem intuitive. Is this pre-perturbation terrace slope and upstream-downstream boundary condition relationship something that is seen in other experimental studies? In nature? I see your discussion includes some mention of this explicitly, and introduces the active tectonic aspect that unfortunately complicates interpretations and adds non uniqueness to potential interpretations of terrace slope history. Can you predict/offer guidelines for which kind of natural systems your experimental results would be best applied?*
Variability in terrace slopes has been reported from field studies (e.g., Tofelde at el., 2017; Baker and Gosse, 2009; Burgette et al., 2017; Poisson and Avouac, 2004), while others have observed parallel or semi-parallel terrace surface slopes (e.g., Faulkner et al., 2016; Hanson et al., 2006). The slope-comparison is one of the parameters that should be investigated to identify the main terrace driving mechanism. However, it cannot stand alone, as both changes in $Q_w$ or $Q_{s,in}$ can results in a reduction in slope. As such, this characteristic should be seen in combination with other observations, as we have summarized in the new figure 9.

*P10 L22 add a space after "...Fig 5)."*
OK.

*P15 L30-31 Perhaps cite the figure # again for clarity, for which grey vs. yellow circles relate to this sentence.*
Done.

*P16 L6-8 The last sentence of Section 5 suggests chemical signals may be propagated more efficiently through systems during phases of aggradation, rather than phases of incision when mixing of older stored sediment might overprint the chemical signature of "fresh" hillslope derived sediment. This is interesting...Your statement makes sense, however would it also be fair to say that the chemical signature would be a function of the ratio of the "fresh" to recycled sediment (and obviously the erosion rate upstream)? And that those ratios could vary greatly given different system scales (I'm thinking about the ratio of upstream derived Qs vs excavated volume)? Perhaps this is a tangential idea more suitable for its own paper?!*
Following the thoughts of the reviewer, we have added a sentence stating that the degree of signal modification is a function of the mixing- ratio of fresh and remobilized material (l. 516-518).

**Alluvial channel response to environmental perturbations: Fill-terrace formation and sediment-signal disruption**

Stefanie Tofelde[1,2], Sara Savi[3]Savi[1], Andrew D. Wickert[4]Wickert[2], Aaron Bufe[2]Bufe[3], Taylor F. Schildgen[2]Schildgen[1,3]

[1]Institut für Erd- und Umweltwissenschaften und Geographie, Universität Potsdam, 14476 Potsdam, Germany
[2]Helmholtz Zentrum Potsdam, GeoForschungsZentrum (GFZ) Potsdam, 14473 Potsdam, Germany
[3]Institut für Geowissenschaften, Universität Potsdam, 14476 Potsdam, Germany
[4]Department[2]Department of Earth Sciences and Saint Anthony Falls Laboratory, University of Minnesota, Minneapolis, MN 55455, USA
[2]Helmholtz Zentrum Potsdam, GeoForschungsZentrum (GFZ) Potsdam, 14473 Potsdam, Germany

*Correspondence to*: Stefanie Tofelde (tofelde@uni-potsdam.de)

**Abstract.** The sensitivity of fluvial systems fill terraces to tectonic and climatic boundary conditions allows us to use the geomorphic and stratigraphic records as quantitative make them potentially useful archives of past climatic and tectonic conditions. Thus, fluvial terraces that form on alluvial fans and floodplains as well as the rate of sediment export to oceanic and continental basins are commonly used to reconstruct paleo-environments. However, we currently lack a systematic and quantitative understanding of the transient evolution of fluvial systems and their impacts of base-level, water discharge, and sediment discharge changes on terrace formation and associated sediment storage and release in response to changes in base level, water input, and sediment input. Such. This knowledge is necessary to quantify gap precludes a quantitative inversion of past environmental changechanges from terrace records or sedimentary deposits, and to disentangle the multiple possible causes for terrace formation and sediment deposition.terraces. Here, we use a set of seven physical experiments to explore terrace formation and sediment export from a single, braided channel system that is perturbed by changes in upstream water discharge orand sediment supply, or through downstream base-level fall. Each perturbation differently affects (1) the geometry of terraces and channels, (2) the timing of terrace cuttingformation, and (3) the transient response of sediment export from the basin.discharge. In general, an increase in water discharge leads to near-instantaneous channel incision across the entire fluvial system and consequent local terrace cutting, thus preserving preservation of the initial channel slopeprofile on terrace surfaces, and it also produces a transient increase in sediment export from the system. that eventually returns to its pre-perturbation rate. In contrast, a decreased changes in the upstream sediment-supply rate may result in longer lag-times before terrace cutting, leading to terrace slopes that differ from the initiala less well preserved pre-perturbation channel slopeprofile, and may also lagged responsesproduce a gradual change in sediment exportoutput towards a new steady-state value. Finally, downstream base-level fall triggers the upstream propagationmigration of a diffuse knickzone, forming terraces with upstream-decreasing ages. The slopegradient of terraces triggered by base-level fall mimicsmimicks that of the newly-adjusted active channel, whereas slopesgradients of terraces triggered by a decreasevariability in upstream sediment discharge or an increase in upstreamor water discharge are steeper compared to the new equilibrium channel. By combining fill-terrace records with

constraints on sediment export, we can distinguish among environmental  perturbations that would otherwise remain unresolved when using just one of these records.

**1 Introduction**

 Sediment-routing systems are commonly subdivided into three zones: a sediment-production zone, typically a mountainous region; a transfer zone of alluvial and fluvial systems that transport and/or temporarily store sediment; and a sedimentation (or deposition) zone, comprising continental or oceanic basins (Fig. 1; Allen, 2017; Castelltort and Van Den Driessche, 2003). Because climate and tectonics can affect sediment production rates, any changes in those conditions may lead to the formation of fluvial terraces in the transfer zone or changes in sedimentation rates in the deposition zone (Alloway et al., 2007; Bull, 1990; Scherler et al., 2015; Zhang et al., 2001). Many past studies have used such records to reconstruct paleoenvironmental conditions (fluvial terraces: Litty et al., 2016; Poisson and Avouac, 2004; Schaller et al., 2004; sedimentation rates: Hay et al., 1988; Zhang et al., 2001). Quantitative interpretations of either record, however, require a clear understanding of how terraces are formed or how sedimentary signals are altered in the transfer zone (Romans et al., 2016 and references therein). In addition, both records suffer from ambiguity, because variability in different environmental parameters can produce similar sedimentary responses. For example, changes in either sediment or water inputs can create fill terraces (Scherler et al., 2015) and affect sediment deposition rates (e.g., Armitage et al., 2011; Simpson and Castelltort, 2012).

 Alluvial rivers adjust their slope and width with respect to the local base-level such that in a graded (steady) state, the incoming water discharge ($Q_w$) can transport the incoming sediment supply ($Q_{s,in}$) downstream (Buffington, 2012; Gilbert, 1877; Lane, 1955; Mackin, 1948). When graded, the slope ($S$) scales nearly linearly with the ratio of $Q_{s,in}$ and $Q_w$ (e.g., Blom et al., 2017; Malatesta and Lamb, 2018; Parker, 1979; Wickert and Schildgen, 2019):

$$S \propto \left( \frac{Q_{s,in}}{Q_w} \right) \qquad (1)$$

 Changes in boundary conditions ($Q_w$, $Q_{s,in}$, and base level) therefore cause alluvial rivers to adjust their geometries through sediment deposition (aggradation) or incision, until a new graded profile is reached. Incision or aggradation result from the dependence of bedload-transport capacity on slope and water discharge (Meyer-Peter and Müller, 1948). For example, if $Q_w$ increases while $Q_{s,in}$ is held constant, the transport capacity exceeds $Q_{s,in}$, which leads to the entrainment of additional sediment from the channel bed, thus incision. As incision proceeds, the channel slope decreases until the transport capacity drops to match $Q_{s,in}$. Conversely, if $Q_{s,in}$ exceeds the transport capacity of the channel, sediment will be deposited to steepen the channel, thus increasing the transport capacity until it matches $Q_{s,in}$. These adjustments can be recorded through (1) fill-terrace formation in the transfer zone (e.g., Bridgland and Westaway, 2008; Bull, 1990; Merritts et al., 1994) and (2) changes in sediment export to basins (e.g., Allen, 2008; Castelltort and Van Den Driessche, 2003; Romans et al., 2016).

Fluvial fill terraces form when rivers incise their formerly deposited sediments (Bull, 1990; Howard, 1959), preserving former channel floodplains as terrace surfaces in a process we call "terrace cutting". Such changes in channel-bed elevation can be triggered by changes at the upstream end of the river, namely the sediment to water discharge ratio, $Q_{s,in}/Q_w$ (eq. 1; e.g., Dey et al., 2016; Scherler et al., 2015; Schildgen et al., 2016; Tofelde et al., 2017), or by base-level changes at the downstream end (e.g., Fisk, 1944; Merritts et al., 1994; Shen et al., 2012). Drivers for terrace formation through the first mechanism include climatically driven variability in $Q_w$ (Hanson et al., 2006; Penck and Brückner, 1909; Scherler et al., 2015; Schildgen et al., 2016; Tofelde et al., 2017) and variability in $Q_{s,in}$, due to, for example, changes in regolith-production rates on hillslopes (Bull, 1991; Norton et al., 2015; Savi et al., 2015), changes in vegetation cover (Fuller et al., 1998; Garcin et al., 2017; Huntington, 1907), exposure of additional regolith following glacier retreat (Malatesta et al., 2018; Malatesta and Avouac, 2018; Savi et al., 2014; Schildgen et al., 2002) or changes in landslide activity (e.g., Bookhagen et al., 2006; McPhillips et al., 2014; Scherler et al., 2016; Schildgen et al., 2016). River incision and terrace cutting through an upstream-migrating knickzone have been related to changes in glacio-eustatic sea-level (Fisk, 1944; Merritts et al., 1994; Shen et al., 2012) or lake-level (Farabaugh and Rigsby, 2005). In some cases, internal dynamics of the system, sometimes referred to as "autogenic processes", lead to terrace formation that cannot be directly linked to external forcing (e.g., Erkens et al., 2009; Limaye and Lamb, 2016; Malatesta et al., 2017; Patton and Schumm, 1981; Womack and Schumm, 1977) .

When studying terraces in the field, it can be difficult to distinguish between terraces that mark a sudden switch from aggradation or stable conditions to incision ("fill-top" terraces of Howard, 1959) and those that preserve surfaces that were cut by a river moving laterally during a period of overall incision ("fill-cut" terraces of Bull, 1990 and Pazzaglia, 2013). In the latter case, there can be a substantial lag between the onset of the environmental perturbation and the abandonment of the terrace surface (e.g., Steffen et al., 2010, 2009). Consequently, from fill terraces alone, both the formation mechanism (change in $Q_w$, $Q_{s,in}$ or base level) and the timing of the perturbation can be ambiguous.

Numerical and experimental work has demonstrated that the geometrical adjustment of alluvial rivers to external perturbations not only creates fluvial terraces, but also affects sediment discharge at the outlet ($Q_{s,out}$; Allen and Densmore, 2000; Armitage et al., 2013, 2011; Bonnet and Crave, 2003; Simpson and Castelltort, 2012; Tucker and Slingerland, 1997; van den Berg van Saparoea and Postma, 2008; Wickert and Schildgen, 2019), which may be recorded by changes in sedimentation rates in the deposition zone. For example, increases in either $Q_w$ or $Q_{s,in}$ increase $Q_{s,out}$ (Allen and Densmore, 2000; Armitage et al., 2013, 2011; Bonnet and Crave, 2003; Simpson and Castelltort, 2012), but each has a characteristic signature. Whereas a change in $Q_{s,in}$ triggers a permanent change in $Q_{s,out}$, a change in $Q_w$ leads to a transient change in $Q_{s,out}$ (Armitage et al., 2011; Bonnet and Crave, 2003; Wickert and Schildgen, 2019). However, because environmental forcings can be cyclic rather than step changes, it can be difficult to relate variability in sedimentation rates to a distinct forcing. Moreover, changes in $Q_{s,out}$ in response to changes in $Q_{s,in}$ or $Q_w$ may be buffered, amplified, or directly transmitted through sediment routing systems (Armitage et al., 2013; Godard et al., 2013; Romans et al., 2016 and references therein; Simpson and Castelltort, 2012). We propose that to correctly interpret changes in sedimentation rates, the modifications of $Q_{s,in}$ within the

transfer zone (then referred to as $Q_s$) due to sediment deposition (channel aggradation) and remobilization (channel incision) must be understood.

The long temporal and broad spatial scales of fill terrace-formation and sediment deposition preclude direct observations of their potential links in nature. Numerical models provide an inroad to understand the evolution of river long profiles and/or $Q_{s,out}$ after perturbations (Blom et al., 2017, 2016; Malatesta et al., 2017~~Sediment is moved across the Earth's surface from the production zone (mountainous regions), through the transfer zone (fluvial channels and floodplains), to the final depositional zone (continental and oceanic sedimentary basins) (Allen, 2017; Castelltort and Van Den Driessche, 2003). Because sediment production in mountainous regions is thought to vary with climatic and tectonic conditions, any changes in those conditions may be reflected in the sedimentary deposits in the transfer or depositional zones (Alloway et al., 2007; Zhang et al., 2001). However, reliable reconstructions of past conditions from sedimentary deposits require a detailed understanding of sediment transport along the sediment routing (or source-to-sink) system, including any potential alteration of signals through the transfer zone, as well as the preservation of the sedimentary deposits and its signals over time (Romans et al., 2016 and references therein).~~

~~Fluvial fill terraces represent transient sediment storage along river channels, and therefore they are an important component of the sediment-routing system (e.g., Allen, 2008). They are generated by variations in river-bed elevations due to sediment deposition followed by river incision into the formerly deposited sediments (Bull, 1990). As a result of incision, remnants of the former floodplain can be abandoned by the active channel and preserved as terraces, a process we refer to as "terrace cutting". Fill terraces, as such, are an indicator of unsteadiness in the parameters that control fluvial-channel geometry. Aggradation and incision can be triggered by changing conditions at the upstream end of the river, namely the sediment to water discharge ratio, $Q_{s,in}/Q_w$ (e.g., Buffington, 2012; Gilbert, 1877; Lane, 1955; Mackin, 1948), or by base-level changes at the downstream end (e.g., Fisk, 1944; Merritts et al., 1994; Shen et al., 2012). In some cases, internal dynamics of the system, sometimes referred to as "autogenic processes", may lead to terrace formation which cannot be directly linked to any external forcing at the upstream or downstream end of the channel (e.g., Erkens et al., 2009; Limaye and Lamb, 2016; Malatesta et al., 2017; Patton and Schumm, 1981; Womack and Schumm, 1977). The cutting of terraces can either coincide with or lag behind the onset of the perturbation that drives terrace formation. The formation of fill terraces in response to external perturbations has two major implications: (1) fill terraces potentially provide a record of past environmental conditions (e.g., Bridgland and Westaway, 2008; Bull, 1990; Merritts et al., 1994); and (2) the deposition and erosion of fill terraces can alter downstream sediment signals, complicating signal propagation from catchment headwaters to long-term depositional sinks (e.g,. Allen, 2008; Castelltort and Van Den Driessche, 2003; Romans et al., 2016).~~

~~Fill-terrace deposits have been used to infer past variability in discharge (Litty et al., 2016; Poisson and Avouac, 2004) or sediment supply (Bookhagen et al., 2006; Schaller et al., 2004). For a reliable reconstruction of such parameters, however, it is essential to understand how closely terrace formation tracks environmental perturbations. Because most studied fill terraces are thousands to millions of years old and form over the course of years to thousands of years (e.g., Bookhagen et al., 2006; Schaller et al., 2004; Schildgen et al., 2002, 2016; Tofelde et al., 2017), fill-terrace formation can rarely be observed~~

130 directly in nature. Consequently, we need alternative ways to investigate the formation of fill terraces and their impacts on downstream sediment discharge.

Numerical models provide an opportunity to predict the evolution of alluvial river-bed elevation over time (Blom et al., 2017, 2016; Simpson and Castelltort, 2012; Slingerland and Snow, 1988; Wickert and Schildgen, 2019), but most simulate river-profile 2018). However, those predictions commonly are limited to the evolution without takingof the longitudinal profile

135 and do not take into account modifications of the channel width or terrace formationthe cutting of terraces (Blom et al., 2017, 2016; Simpson and Castelltort, 2012; Slingerland and Snow, 1988; Wickert and Schildgen, 2019). In addition, most numerical models for river-profile evolution rely on equations derived for the steady-state case. As such, they may not accurately simulate transient responses, which are important for capturing terrace formation and modifications of $Q_{s,in}$ in the transfer zone. ). Hancock and Anderson (2002) modeled bedrock strath terrace formation, a partially analogous process, but their erosional

140 stream-power-based approach cannot be easily translated to transport-limited systems, where slope and long-profile evolution result from both sediment and water inputs.

Physical experiments provide an alternative approach to studying the dynamics of the transfer zone, including terrace formation (Baynes et al., 2018; Frankel et al., 2007; Gardner, 1983; Lewis, 1944; Mizutani, 1998; Schumm and Parker, 1973; Wohl and Ikeda, 1997) and the evolution of $Q_{s,out}$ (Bonnet and Crave, 2003; Van den Berg van Saparoea and Postma 2008).).

145 Most experimental studies have tested the cutting of terraces due to base-level fall (Frankel et al., 2007; Gardner, 1983; Schumm and Parker, 1973).), or explained their cuttingformation through autogenic processes (Lewis, 1944; Mizutani, 1998). Only one experimental study by Baynes et al. (2018) investigated terrace formation as a response to changes in sediment supply ($Q_{s,in}$), or water discharge ($Q_{w}$), but this study focused on vertical incision into bedrock and strath-terrace cutting. Van den Berg van Saparoea and Postma (2008) and Bonnet and Crave (2003) investigatedperformed experiments to investigate the

150 effects of variabilitypulses in $Q_{w}$ and $Q_{s,in}$ on topograpicthe evolution of longitudinal channel profiles and sediment discharge at the basin outlet ($Q_{s,out}$), but neither considered how these processes may be linked to they did not focus on terrace formation. To our knowledge, there are no experimental studies that consider the combined evolution of two records of landscape evolution – systematically compare how fill terraces in the transfer zone and sediment discharge toformed through various mechanisms may differ from one another, or investigate the deposition zone – in response to environmental

155 perturbationsimpacts of terrace formation on downstream sediment discharge.

In this study, we present results from seven physical experiments of the transfer zone, represented by a single braided channelchannels in non-cohesive sediment, in which we perturb $Q_{w}$, $Q_{s,in}$, and base level. We investigate the timing and geometrical response (slope, width) of the alluvial channel in the transfer zone (with a particular focus on to test three potential mechanisms of fill-terrace cutting) and patterns and response rates of $Q_{s,out}$, with a particular focus on how the records may be

160 linked and if a combination of both records can be diagnostic of specific changes in boundary conditions.

**2 Methods**

[revised manuscript text omitted]
. 2D). To distinguish wet and dry pixels by color, the supplied water was dyed blue (Fig. 2C).  From the binary images, the number of wet pixels in each cross-section (perpendicular to the basin margin and therefore to the average flow direction) were counted. Analyses were restricted to the areas within the orange box (Fig. 2C, D), because terraces mainly developed in this part of the channel and because we considered this sector at the upstream side of the basin to be least affected by the fixed location of the outlet. To calculate average channel width, the average number of wet pixels in 1200 cross sections  were counted and are reported with one standard deviation. No overhead photos were taken for the *Ctrl_1* experiment because of an error in the camera installation.

We manually measured $Q_{s,out}$ at 10-minute intervals by collecting the discharged sediment in a container over a 10-second period and measuring its volume. This approach allowed us to estimate whether the system had returned to steady state

9

290     ($Q_{s,in} \approx Q_{s,out}$) during the runs. At the same 10-minute interval, we measured bed elevation at the inlet and at the outlet to estimate the spatially-averaged channel slope. We interpreted a constant slope for over more than 30 minutes as additional evidence for a graded (steady state) channel. The data can be found in the supplementary material.

       We ran seven experiments to monitor how  changes in $Q_{s,in}$, $Q_w$, and base level affect channel adjustment, the evolution of fill terraces along the main-stem and sediment discharge at the outlet ($Q_{s,out}$) through time. The

295   experiments are summarized in Table 1. To investigate the effect of $Q_w$, we ran two separate experiments: in one experiment we doubled $Q_w$ ($IQ_w$ = increase discharge) to 190 mL/s at 240 min (end of the 'spin-up' time) and in the other experiment we first halved $Q_w$ to 48 ml/s at 240 min and then returned to the initial 95 mL/s at 480 min ($DQ_w\_IQ_w$ = decrease discharge, increase discharge). To test the effect of $Q_{s,in}$, we ran one experiment in which we reduced the $Q_{s,in}$ by 83% to 0.22 ml/s ($DQ_{s,in}$ = decrease sediment supply) at 240 min and another one in which we first doubled $Q_{s,in}$ to 2.6 ml/s at 240 min and then halved

300   $Q_{s,in}$ again to the initial 1.3 ml/s at 480 min ($IQ_{s,in}\_DQ_{s,in}$ = increase sediment supply, decrease sediment supply). All $Q_{s,in}$ and $Q_w$ changes were imposed instantaneously, resulting in a step function in the forcing (Table 1). Immediately before imposing these changes, we covered the near-channel surface with a thin layer of red sand to optically identify the area that was reworked after the change. This method allowed us to distinguish visually between fill-top (covered in red sand) and fill-cut terraces (red sand removed due to continuous overwash). We ran one experiment in which we dropped the base level by 10

305   cm gradually over 20 min starting at 240 min, resulting in a base-level lowering rate of 0.5 cm/min ($BLF$). For this experiment, we started with a base level higher than in the initial setting by flooding the basin surrounding the wooden box (Fig. 2A). The final base level equaled that of the other experiments. In this experiment, the red sand was applied immediately before the onset of base-level lowering. Additionally, we performed two control experiments in which we made no changes to the initial conditions in  to investigate whether terraces would form in our experiment without any change in external

310   forcing (*Ctrl_1, Ctrl_2*).

**3 Results**

      Fluvial terraces were cut in the experimental runs $IQ_w$, $DQ_w\_IQ_w$ (in the $IQ_w$ phase), $DQ_{s,in}$, $IQ_{s,in}\_DQ_{s,in}$ (in the $DQ_{s,in}$ phase) and *BLF* (Fig. 2, 3, 4). No terraces were formed after the 'spin-up' time of *Ctrl_1* and *Ctrl_2*. The terraces visible in

the cross-section of *Ctrl_2* formed in response to incision during the 'spin-up' phase and did not substantially develop after

315  240 min (Fig. 4B, red line). We named the terraces to the left of the channel (in downstream direction) $T_L$ and the terraces to the right $T_R$. In all terrace-forming experiments, both fill-top (red sand) and fill-cut terraces (red sand removed) formed (Fig. 3). Only in the $IQ_w$, $DQ_w\_IQ_w$ and $IQ_{s,in}\_DQ_{s,in}$ experiments were the fill-top terraces preserved as the most extensive terrace surface, at least on one side of the channel (Fig. 3A, B, D). In the $DQ_{s,in}$ experiment, only a fraction of the fill-top terrace ($T_L$) survived the transient channel adjustment phase (Fig. 3C). In all experiments that experienced upstream perturbation, fill-top

320  and fill-cut terraces formed only in the upstream half of the sandbox. In contrast, in the BLF experiment, terraces formed in

the downstream channel reach immediately after the onset of base-level drop, but were mostly destroyed within 30 min (Fig. 3E, G). Later during the BLF experiment, terraces formed in the upstream portion of the sandbox (Fig. 3F).

Fill-cut terrace cutting lagged minutes to hours behind the onset of the imposed perturbation (Fig. 3). We determined lag times from overhead photos, defined as the time interval between the onset of the perturbation (at minute 240 or 480) and the final time that the future terrace surface was occupied by water. In the two experiments during which we changed $Q_w$ and in the $IQ_{s,in}\_DQ_{s,in}$ experiment, the cutting of fill-cut terraces began within 6 minutes after the change in boundary conditions (Fig. 3A, B, D). In the $IQ_w$ experiment, for example, the majority of the $T_L$ terrace is a fill-top terrace (0 min lag-time) and only a small part at the downstream end was occupied until 6 minutes after perturbation (Fig. 3A, H). In the $DQ_{s,in}$ experiment, however, several fill-cut terraces formed successively with lag-times between ~14 min and 289 min (Fig. 3C). This experiment was the only one in which a sequence of terraces, instead of a single major surface, developed. In the *BLF* experiment, terrace cutting in the upstream part of the basin began 112 and 117 min after the onset of base-level lowering (Fig. 3F).

Fill-terrace formation requires, changes in the channel-bed elevation and  width of the active floodplain. In our experiments,  sediment deposition (aggradation) or erosion (incision) altered the channel-bed elevation (Fig. 5). However, these  changes were not uniform along the channel.  In the runs *Ctrl_1* and *Ctrl_2,* the longitudinal profiles were stable over time and experienced only minor lowering in bed elevation (max. 4 cm)  at the upstream end (Fig. 5A, B). A sudden increase in $Q_w$ ($IQ_w$, and the $IQ_w$ phase o*f* $DQ_w\_IQ_w$) or a decrease in $Q_{s,in}$ ($DQ_{s,in}$, and the $DQ_{s,in}$ phase of $IQ_{s,in}\_DQ_{s,in}$) both led to channel incision (Fig. 5C, D, G, H). This incision,  was most pronounced at the upstream end, near the changing boundary condition, but  not recognizable at the downstream end (Fig. 5D, G, H), where the channel-bed elevation was fixed due to the steady base level. Sediment deposition in the channels followed a decrease in $Q_w$ ($DQ_w$ phase of $DQ_w\_IQ_w$) or an increase in $Q_{s,in}$ ($IQ_{s,in}$ phase of $IQ_{s,in}\_DQ_{s,in}$), which was, again, most recognizable at the upstream end of the channel (Fig. 5E, F). The drop in base level, however, caused maximum incision at the downstream end, and the incision wave migrated upstream as a diffuse knickzone (Fig. 5I).

Channel  slope and width changes were observed in the absence of external perturbations. Channel slopes in the *Ctrl_1* and *Ctrl_2* marginally decreased  after the 240 min 'spin-up' time from ~~~0.074 and 0.071, respectively, to around 0.070 and 0.067 (~6 % reduction; Fig. 6A). As such, we consider any change in slope after the 'spin-up' time that is on the same order as those observed in *Ctrl_1* and *Ctrl_2* as ongoing adjustment to the reference condition as opposed to the result of an external perturbation. Channel width in the control experiments varied slowly between ca. 20 cm and 35 cm.

External perturbations in water and sediment inputs forced the channel width and slope to evolve. An instant doubling of $Q_w$ ($IQ_w$; Fig. 6B) resulted in a rapid,  decrease in channel slope that decayed exponentially as the channel approached a new graded state. After approximately 480 min, the slope was reduced from ~0.072 to ~0.043 (40% reduction),

and new stable conditions were reached. The doubling of $Q_w$ also triggered an instant narrowing of the channel from ~35 cm to ~15 cm (~57 % decrease), followed by subsequent slow widening.

In contrast, suddenly reducing A sudden reduction in $Q_w$ to half its initial value ($DQ_w$ $IQ_w$; Fig. 6C) increased channel5C) resulted in an increase in slope from ~0.072 to ~0.085 (18% increase) between 240 and 480 min runtime, and causeda widening of the channel to widen from about 25 cm to about 45 cm (~80% increase) during the same time period. The subsequent doubling in $Q_w$ back to its initial value triggered a rapid (nearly exponential) reduction in slope back to the initial ~0.072 (~15% reduction, again following an exponential decay) and an instantaneous narrowing of the channel (~45% reduction) followed by slow widening.

ReducingA reduction in $Q_{s,in}$ by 83% ($DQ_{s,in}$; Fig. 6D5D) triggered a decrease in channel slope to decrease at a slower. The rate of decrease was lower than in the $IQ_w$ run, and the new slope stabilized around 0.06006 (24% reduction from the initial 0.079). An instantaneous decrease in channel width also occurred, but this change was again less pronounced than what we observed in the $IQ_w$ experiment (~33% reduction). We detected noNo subsequent widening of the channel was detectable.

Finally, increasing An increase in $Q_{s,in}$ ($IQ_{s,in}$ $DQ_{s,in}$; Fig. 6E5E) led to an increase in channel steepeninggradient from a slope ofabout 0.070 to about 0.078 (11% increase) and increasedan increase in channel width from about 30 cm to about 55 cm (~83% increase). The subsequent reduction in $Q_{s,in}$ decreased led to a decrease of the channel slope and caused an instantaneous channel narrowing to < 30 cm, followed by subsequent widening back to the initial width of ~30 cm.

DuringFor the base-level fall experiment ($BLF$; Fig. 6F), mean5F), channel slope instantly and rapidly increased after the onset of base-level fall from about 0.047 to 0.073 (55% increase), and continued to increaseit increased at a slower rate further to about 0.08, before decreasinglowering back to 0.072. These meanHowever, these slope values, however, average over are simply calculated based on the height difference at the inlet and outlet, ignoring any spatial variability in incision, meaningslope along the experiment reach that they do not resolve the details of the diffusiveis, in the $BLF$ experiments, significant due to knickzone propagation of the knickzone. Beyond impacts on slope, the. The drop in base level resulted in a sudden decreasedrop in channel width, followed by three cycles of channel widening and narrowing. In summary, we observed that an increase in $Q_w$ and a decrease in $Q_{s,in}$ resulted in an immediate decrease in channel slope (through upstream incision) and an instant reduction in channel width, whereas a drop in base level caused an increase in channel slope (through downstream incision) and a reduction in channel width (Fig. 65).

The time of terrace cutting lagged minutes to hours behind the onset of the perturbation (Fig. 5). Lag times were determined from overhead photos and are defined as the time interval between the onset of the perturbation (at minute 240 or 480) and the last time the future terrace surface was occupied by water. The times given in Fig. 5 refer to the last occupation of the areas for which swath profiles were extracted (Fig. 6 right panel). In the two experiments in which we changed $Q_w$ and in the $IQ_{s,in}$ $DQ_{s,in}$ experiment, terrace cutting in the upstream reach of the channel (Fig. 3 right column, Fig. 4; dashed arrows) began within ~5 minutes after the change in boundary conditions (Fig. 5; black arrows). In the $IQ_w$ experiment, for example, the majority of the $T_A$ terrace was cut instantly (no removal of red sand) and only a small part at the downstream end was occupied again until 6 minutes after perturbation (Fig. 2A, B). In the $DQ_{s,in}$ experiment, however, the $T_A$ and $T_B$ terraces were

To analyze how well terrace surfaces represent channel slopes  immediately preceding the time of perturbation , we compared the elevation profiles of the  terraces on each side of the channel (yellow and orange lines) with the channel that existed at the onset of the perturbation (red line) (Fig. 7). We sampled elevations across the most extensively preserved terrace surface, regardless of its lag time, in a way that is similar to terrace mapping in the field. In experiments with increasing $Q_w$ ($IQ_w$, $IQ_w$ phase of $DQ_w\_IQ_w$) or base-level changes (*BLF*), the elevation profiles of the terraces are similar to the initial floodplain profile (Fig. 7A, B and E). In cases of changes in $Q_{s,in}$ ($DQ_{s,in}$, $DQ_{s,in}$ phase of $IQ_{s,in}\_DQ_{s,in}$), the terraces were cut at lower elevations than the former channel (Fig. 7C, D). In the $DQ_{s,in}$ experiment, fill-cut terraces on either side of the channel formed at different elevations, with one surface about 3 cm below the other (Fig. 7C, 4E). In contrast, terrace surfaces in the other four experiments are at approximately the same elevation  (Fig. 4, 7). Despite similar elevations, the slope differences  between $T_L$ and $T_R$ range from about 5% ($IQ_{s,in}\_DQ_{s,in}$) to 33% ($IQ_w$). When comparing terrace slopes to the active channel slopes  at the end of each run (blue lines), terrace slopes are steeper (by 20–122%) in all experiments in which upstream conditions ($Q_w$, $Q_{s,in}$)  changed (Fig. 7 A-D). In contrast, the slopes of the terraces and the active channel in the *BLF* experiment are similar to one another (within 11%) (Fig. 7E).

Changes in boundary conditions also affected $Q_{s,out}$ (Fig. 6, lowest panels). An instantaneous doubling of $Q_w$ ($IQ_w$; Fig. 6B) resulted in an instant increase in $Q_{s,out}$ to more than 20 times $Q_{s,in}$. This rapid increase was followed by an exponential decay down to the initial $Q_{s,out}$ value. A sudden reduction in $Q_w$ to half its initial value ($DQ_w\_IQ_w$; Fig. 6C) resulted in a decrease in $Q_{s,out}$. The subsequent doubling in $Q_w$ back to its initial value triggered a rapid increase in $Q_{s,out}$ that decayed over time. In contrast, neither the instantaneous reduction in $Q_{s,in}$ by 83% ($DQ_{s,in}$; Fig. 6D) nor the doubling in $Q_{s,in}$ ($IQ_{s,in}\_DQ_{s,in}$; Fig. 6E) triggered a measurable change in $Q_{s,out}$. For the base-level fall experiment (*BLF*; Fig. 6F), $Q_{s,out}$ could not be measured before and during the base level drop, because the basin surrounding the wooden box was flooded for this experiment. $Q_{s,out}$ was only measured from minute 280 onwards, which corresponds to minute 40 after the 'spin-up' of the base level fall. At that time, $Q_{s,out}$ was still about 10 times higher than $Q_{s,in}$, and $Q_{s,out}$ decreased approximately linearly from that time onwards.

**4 Discussion**

When attempting to use geomorphic or depositional records to reconstruct paleo-environmental conditions, we face a range of challenges. One challenge is to understand how the information on environmental boundary conditions is translated

420 into the sedimentary record, considering the potential modification of sediment signals during fluvial transport. A second, related challenge is that depositional records and fluvial terraces often cannot unambiguously be associated with a particular forcing mechanism. In the following, we will discuss these two challenges, and we will focus on both records that we monitored in our experiments – fill terraces in the transfer zone and sediment discharge to the deposition zone ($Q_{s,out}$). Further, we will discuss the use of an integrated set of observations to address these challenges, the limitations that arise when comparing the

425 experimental work to natural settings, as well as potential implications of our observations for future field studies.

**4.1 Terrace formation in the transfer zone**

**4.1.1 Conditions of terrace formation, lag times, and the 5.1 Channel response to perturbations and conditions of terrace formationThe preservation of pre-perturbation channel profiles**

The cutting of fluvial fill terraces requires that vertical incision and a simultaneous reductionoutpaces lateral erosion

430 on one or both sides of the active floodplain width.channel. Whether this occurs depends on the response of alluvial channels to changing boundary conditions, which can includeoccur through adjustments ofto their slope, wetted perimeter (width and depth), and/or bed-surface texture (grain-size distribution) (e.g., Blom et al., 2017; Buffington, 2012 and references therein; Wickert and Schildgen, 2019).(Blom et al., 2017; Buffington, 2012 and references therein). Because the grain-size distribution in our experiments remained constant, we focus our discussion on the externally forced adjustments of channel slope ($S$) and

435 width ($w$) during terrace formation.

In our experiments, river incision (with terrace cutting) was driven by an increase in $Q_w$, a decrease in $Q_{s,in}$, or a fall in base level (Figs. 3 - 6). In the case of base-level fall, incision began at the downstream boundary and diffused upstream, producing a transient steepening. Enhanced $Q_w$ or reduced $Q_{s,in}$, on the other hand, decreased channel slope. The evolution of longitudinal channel profiles in our experiments is in agreement with earlier flume studies that investigated channel response

440 to upstream (van den Berg van Saparoea and Postma, 2008) and downstream (Begin et al., 1981; Frankel et al., 2007) perturbations, as well as with numerical models that predict the evolution of longitudinal profiles following variations in $Q_{s,in}$, $Q_w$ or base level (Blom et al., 2017; Simpson and Castelltort, 2012; Wickert and Schildgen, 2019). In all experiments, incision and terrace cutting coincided with an instantaneous decrease in channel width, while aggradation corresponded to an increase in channel width (Fig. 7).

445 A common application of fluvial-terrace mapping is to reconstruct paleo-longitudinal channel profiles from terrace remnants (e.g., Faulkner et al., 2016; Hanson et al., 2006; Pederson et al., 2006; Poisson and Avouac, 2004). These profiles are thought to be representative of the former channel profiles, ideally reflecting their geometries immediately prior to a perturbation. However, morphological adjustments of a channel to external perturbations require time, such that the geomorphological response can lag behind the changes in environmental parameters (e.g., Blum and Tornqvist, 2000; Tebbens

450 et al., 2000; Vandenberghe, 2003, 1995). The lag time between an external perturbation and the onset of terrace cutting determines how much time the fluvial system has to modify the terrace sediments before their abandonment. In the following,

we first discuss the relationship between lag-times and the preserved terrace profiles related to upstream perturbations ($Q_{s,in}$, $Q_w$), followed by those related to downstream perturbations (*BLF*).

In our experiments, In our experiments, river-bed aggradation and channel steepening occurred after a decrease in $Q_w$ and after an increase in $Q_{s,in}$, whereas river incision (with terrace cutting) and channel-slope lowering were driven by an increase in $Q_w$, a decrease in $Q_{s,in}$, or a fall in base level (Figs. 2, 3, 4). In the case of base-level fall, incision began at the downstream boundary and diffused upstream, producing a transient steepening. The evolution of longitudinal channel profiles in our experiments is in agreement with earlier flume studies that investigated channel response to upstream (van den Berg van Saparoea and Postma, 2008) and downstream (Begin et al., 1981; Frankel et al., 2007) perturbations, as well as with numerical models that predict the evolution of longitudinal profiles following variations in $Q_{s,in}$ or $Q_w$ (Blom et al., 2017; Simpson and Castelltort, 2012; Wickert and Schildgen, 2018). In addition to slope changes, channels can also adjust to external forcing by changing their width (Fig. 5; Buffington, 2012; Church, 1995; Curtis et al., 2010; Dade et al., 2011). In all experiments, an increase in channel width occurred during aggradation (reduced $Q_w$, increased $Q_{s,in}$), and an instantaneous decrease in channel width occurred at the start of incision (increased $Q_w$, reduced $Q_{s,in}$, *BLF*; Fig. 5).No terraces were formed during the two control experiments after the 'spin-up' time. However, this finding does not imply that autogenic terraces do not exist in natural systems, as meander bend cut-off (Erkens et al., 2009; Gonzalez, 2001; Limaye and Lamb, 2016; Womack and Schumm, 1977) could not be tested with our experimental setup. We observed internal variability in sediment storage and release, for example in the form of bank collapse due to lateral channel migration during the experiments. However, local lateral sediment input through bank collapse did not trigger terrace formation in our experiments. Our experimental set-up also precluded terrace formation in response to internal feedbacks between the main stem and tributaries (Schumm, 1979, 1973, Gardener 1983, Schumm and Parker 1973, Slingerland and Snow 1988).

In order to link drivers and response, we turn to the work of Wickert and Schildgen (2018), who coupled equations for flow, sediment transport, and channel morphodynamics to solve for long-profile changes in transport-limited rivers. From this work, in which channel width is allowed to self-adjust following Parker (1978), we distill the following relationships between channel width (*w*), slope (*S*) and either $Q_{s,in}$ or $Q_w$:

$$Q_w \propto \frac{w}{S^{7/6}} \qquad\qquad (1)$$

and

$$Q_{s,in} \propto w \qquad\qquad (2)$$

Eq. 2 predicts the observed reduction in channel width after a decrease in $Q_{s,in}$ (Fig. 5, eq. 2). Eq. 1 predicts that slope should decrease as water discharges increases, which is consistent with the observed decrease in slope from about 0.072 to 0.043 (Fig. 5B) in the *IQ$_w$* experiment, in which water discharge doubled. However, this amount of slope decrease should be matched by an 8% increase in channel width, which runs contrary to the observed instantaneous reduction in channel width by ~57% followed by gradual widening. This response is transient, whereas Wickert and Schildgen (2018) assume an equilibrium

width; the relationship between time-evolving slope, width, and basal shear stress is the most likely cause of this discrepancy. The equilibrium-width solution used by Wickert and Schildgen (2018) assumes a constant ratio between the basal shear stress at bankfull discharge ($\tau_b$) and the critical shear stress for the initiation of sediment motion ($\tau_c$), which can be described by (Parker, 1978):

$$\tau_b^* = (1 + \epsilon)\tau_c^*$$

(3)

Parker (1978) suggested that the fraction of excess shear stress at bankfull flow ($\epsilon$) is about 0.2 for self-formed gravel-bed rivers with equilibrium widths. Empirical measurements have confirmed an epsilon of 0.2 in a large number of rivers across the US (Phillips and Jerolmack, 2016), but Pfeiffer et al. (2017) illustrated that $\epsilon$ increases in tectonically active regions. It could be that rapid uplift is analogous to incision in our experiment during its transient-response phase, causing the channel to narrow and $\tau_b$ to increase, which further accelerates incision. Our experimental results demonstrate that accurately simulating long-profile evolution may require an improved understanding of the transient response of channel width.

**5.2 Preservation of channel profiles**

A common application of fluvial terrace mapping is to reconstruct paleo-longitudinal channel profiles from terrace remnants (e.g., Faulkner et al., 2016; Hanson et al., 2006; Pederson et al., 2006; Poisson and Avouac, 2004). Reconstructed longitudinal profiles from terrace remnants are thought to be representative of the former channel profiles, ideally of conditions immediately prior to perturbations. However, morphological adjustments of a channel to external perturbations require time, such that the geomorphological response can lag behind the changes in environmental parameters (e.g., Blum and Tornqvist, 2000; Tebbens et al., 2000; Vandenberghe, 2003, 1995). The lag time between external perturbations and the onset of terrace cutting determines the degree of reworking of terrace material. Consequently, the shorter the lag time, the better the preservation potential of environmental conditions that existed prior to the time of perturbation.

In our experiments, the terrace surfaces preserve the former channel elevation profiles in the two increased $Q_w$ experiments and in the *BLF* experiment (Fig. 6A, B and E). In contrast, in the decreased $Q_{s,in}$ experiments, terrace-elevation profiles are lower than the river channel immediately preceding the perturbation and, in case of the $DQ_{s,in}$ run, the terraces are also unpaired (Fig. 6C, D). Focusing on the upstream-perturbation-related terrace surfaces (fill-top and fill-cut terraces) following an increase in $Q_w$ had experiments first, we observed short lag-times ($\leq 6$ min; between perturbations and terrace cutting in all $Q_w$-related experiments (Fig. 3A, B and 6B5B, C) and preserved), which ensured good preservation of the channel elevation profiles profile prior to perturbation well (Fig. 7A6A, B). Similarly, terrace cutting in the $IQ_{s,in}$_$DQ_{s,in}$ experiment was characterized by short ($T_R T_B$) or no ($T_L T_A$) lag-times (Fig. 6E5E). The small discrepancy between terrace slopes and initial channel slopes in this experiment (Fig. 7D) is a result of slope variations between the center of the channel belt (where initial and final channel profiles were measured), and the sides of the channel belt, where the terrace slopes were measured.

In contrast, terrace cutting in the $DQ_{s,in}$ experiment occurred with a delay of several hours and the terraces were also cut successively (Fig. 3C, 6D). The difference in lag -times between the $T_L$ and $T_R$ terrace of about two and a half

515 hours resulted in different terrace elevations on both sides of the channel, with elevation profiles several cm below the channel profile prior to perturbation (Fig. 7C). These results illustrate how short lag times are critical to enable accurate reconstructions of the pre-perturbation channel-profile, which can potentially be used to reconstruct paleo-environmental conditions. But what determines the duration of the lag-time?

The length of the lag time between the perturbation and the abandonment of a terrace surface is expected to depend
520 on the ratio of vertical incision versus lateral erosion. Bufe et al. (2018) and Malatesta et al. (2017) demonstrated that the rate of lateral channel migration scales inversely with the height of valley walls (elevation difference between a terrace surface and the active channel). As such, higher incision rates after a perturbation lead to faster wall-height growth and greater reductions in lateral mobility. Accordingly, fast incision should result in short lag times between the onset of the perturbation and terrace cutting, guaranteeing good preservation of the channel profile that existed prior to the perturbation. In contrast, slow river
525 incision and enhanced lateral channel movement can lead to long lag times, with terrace profiles that reflect a channel profile at some (unknown) phase of adjustment. The incision rate, on the other hand, is thought to be a function of the excess sediment transport capacity, and sediment transport capacity should be directly proportional to $Q_w$ (Wickert and Schildgen, 2019): doubling $Q_w$ should correspondingly double the excess sediment transport capacity, whereas halving $Q_{s,in}$ should increase the excess sediment transport capacity by a factor of 0.5. Therefore, increases in $Q_w$ should, in theory, cause more rapid incision,
530 shorter lag-times, and a higher preservation potential for the pre-perturbation channel profile than a proportionately equal reduction in $Q_{s,in}$. However, while one of the two experiments with a reduction in $Q_{s,in}$ ($DQ_{s,in}$) is consistent with this theory (Fig. 6D), in the other one ($IQ_{s,in}$_$DQ_{s,in}$), we observed relatively short lag-times (Fig. 6E). These unexpectedly short lag times might be related to how the incision phase was preceded by an aggradation phase (due to an increase in $Q_{s,in}$). Possibly, the system rapidly settled back to the initial conditions because it had not completely adjusted to the preceding increase in $Q_{s,in}$.

535 ~~The length of the lag time between the perturbation and the abandonment of a terrace surface depends on how effectively vertical incision outcompetes lateral erosion. Bufe et al. (2018) have shown that the rate of lateral channel migration scales inversely with the height of valley walls (elevation difference between a terrace surface and the active channel). As such, the higher the incision rate after perturbation, the faster wall-heights grow and the more lateral mobility is reduced. Due to this positive feedback, rapid incision after a perturbation should result in short lag times between the onset of the
540 perturbation and terrace cutting and a good preservation of the channel profile that existed prior to perturbation. In contrast, if the river incises more slowly, terraces may be cut long after incision initiates, and the terrace profile will not directly reflect the channel profile prior to perturbation.~~

The lag time between the onset of base-level fall and the cutting of terraces in the upstream reach of the channel is about ~115 min (Fig. 6I), which was the time required for the knickzone to propagate upstream.
545 As such, for  terraces related to base-level fall, the temporal lag between the onset of the perturbation and terrace cutting increases with  upstream distance. Hence

17

In contrast, terrace cutting in the $DQ_{s,in}$ experiment occurred with a delay of several hours and the terraces were also cut successively (Fig. 3C, 6D). The difference in lag -times between the $T_L$ and $T_R$ terrace of about two and a half

515 hours resulted in different terrace elevations on both sides of the channel, with elevation profiles several cm below the channel profile prior to perturbation (Fig. 7C). These results illustrate how short lag times are critical to enable accurate reconstructions of the pre-perturbation channel-profile, which can potentially be used to reconstruct paleo-environmental conditions. But what determines the duration of the lag-time?

The length of the lag time between the perturbation and the abandonment of a terrace surface is expected to depend
520 on the ratio of vertical incision versus lateral erosion. Bufe et al. (2018) and Malatesta et al. (2017) demonstrated that the rate of lateral channel migration scales inversely with the height of valley walls (elevation difference between a terrace surface and the active channel). As such, higher incision rates after a perturbation lead to faster wall-height growth and greater reductions in lateral mobility. Accordingly, fast incision should result in short lag times between the onset of the perturbation and terrace cutting, guaranteeing good preservation of the channel profile that existed prior to the perturbation. In contrast, slow river
525 incision and enhanced lateral channel movement can lead to long lag times, with terrace profiles that reflect a channel profile at some (unknown) phase of adjustment. The incision rate, on the other hand, is thought to be a function of the excess sediment transport capacity, and sediment transport capacity should be directly proportional to $Q_w$ (Wickert and Schildgen, 2019): doubling $Q_w$ should correspondingly double the excess sediment transport capacity, whereas halving $Q_{s,in}$ should increase the excess sediment transport capacity by a factor of 0.5. Therefore, increases in $Q_w$ should, in theory, cause more rapid incision,
530 shorter lag-times, and a higher preservation potential for the pre-perturbation channel profile than a proportionately equal reduction in $Q_{s,in}$. However, while one of the two experiments with a reduction in $Q_{s,in}$ ($DQ_{s,in}$) is consistent with this theory (Fig. 6D), in the other one ($IQ_{s,in}$_$DQ_{s,in}$), we observed relatively short lag-times (Fig. 6E). These unexpectedly short lag times might be related to how the incision phase was preceded by an aggradation phase (due to an increase in $Q_{s,in}$). Possibly, the system rapidly settled back to the initial conditions because it had not completely adjusted to the preceding increase in $Q_{s,in}$.

535 ~~The length of the lag time between the perturbation and the abandonment of a terrace surface depends on how effectively vertical incision outcompetes lateral erosion. Bufe et al. (2018) have shown that the rate of lateral channel migration scales inversely with the height of valley walls (elevation difference between a terrace surface and the active channel). As such, the higher the incision rate after perturbation, the faster wall-heights grow and the more lateral mobility is reduced. Due to this positive feedback, rapid incision after a perturbation should result in short lag times between the onset of the
540 perturbation and terrace cutting and a good preservation of the channel profile that existed prior to perturbation. In contrast, if the river incises more slowly, terraces may be cut long after incision initiates, and the terrace profile will not directly reflect the channel profile prior to perturbation.~~

The lag time between the onset of base-level fall and the cutting of terraces in the upstream reach of the channel is about ~115 min (Fig. 6I), which was the time required for the knickzone to propagate upstream.
545 As such, for  terraces related to base-level fall, the temporal lag between the onset of the perturbation and terrace cutting increases with  upstream distance. Hence

17

other words, terrace surfaces created through upstream knickpoint migration are diachronous, become progressively younger upstream despite being a physically a continuous unit. Faulkner et al. (2016) found decreasing OSL ages with upstream distance in a fill terrace along the Chippewa River, USA that formed in response to base-level fall. Similar results have been reported from field studies (Faulkner et al., 2016; Pazzaglia, 2013).conclusions were also reached by Pazzaglia (2013). In comparison, incision was initiated near-synchronously along the entire experimental channelreach when incision was triggered by a change in upstream boundary conditions ($IQ_w$, $DQ_{s,in}$; Fig. 5C4C, D). In summary, lag-times between the onset of athe perturbation and terrace cutting depend on the combination of local incision rates after the perturbation and the trigger for incision (base-level fall vs. a change in upstream conditions).

Lag-times between the perturbation and the onset of terrace cutting can be important when dating the surfaces of fluvial fill terraces in the field. Common methods to date the onset of river incision include the dating of terrace surface material with cosmogenic exposure dating (e.g., Schildgen et al., 2016; Tofelde et al., 2017), dating sand or silt lenses with optically stimulated luminescence close to the terrace surface (OSL; e.g., Fuller et al., 1998; Schildgen et al., 2016; Steffen et al., 2009) or dating embedded organic material with $^{14}$C (Farabaugh and Rigsby, 2005; Scherler et al., 2015). When transferring our observations to a field scenario, the ~2h or more of channel material reworking before terraces were cut within the upstream part of the reach in the *BLF* and the *$DQ_{s,in}$* experiment would result in terrace ages that are younger than the time of perturbation. The best temporal correlations between the perturbation and the terrace surface ages are achieved by those formed by changes in *$Q_w$*, due to the fast onset of vertical incision and minimal reworking of terrace surface material. To assess the significance of this time-lag in natural systems requires more work on how to scale the experiment to larger channels.

**5.3 Differences in terrace surface slopes**

To reliably use fluvial terraces to reconstruct paleo-environmental conditions (i.e., changes in base level, *$Q_{s,in}$* or *$Q_w$*), the identification of the terrace formation mechanism is important. We found that for *$Q_{s,in}$* or *$Q_w$*-related terraces, the slopes of terrace surfaces are always steeper than the active channel (the new steady-state channel after the perturbation), whereas the slope of terraces formed due to downstream perturbations is very similar to that of the active channel (Fig. 6). Similar observations have been made in the field. Poisson and Avouac (2004) measured a reduction in channel slope between terraces due to deeper incision at the upstream end of a flight of terraces in the Tien Shan. They related the changes in longitudinal profiles (inferred from the terraces) to changes in *$Q_w$*. In contrast, Faulkner et al. (2016) measured terraces in the Chippewa River, a tributary to the Mississippi River, which were created in response to base-level fall and upstream knickpoint migration due to incision of the Mississippi channel bed after deglaciation. They observed no major slope change between the longitudinal profile reconstructed from the terrace and the modern channel. According to Wickert and Schildgen (2018), the relationship between slope *S*, *$Q_{s,in}$* and *$Q_w$*, for alluvial rivers taking self-adjusting channel width and channel roughness into account, can be described as:

$$S \propto \left(\frac{Q_{s,in}}{Q_w}\right)^{6/7} \qquad\qquad (4)$$

580 ~~According to this relationship, a decrease in $Q_{s,in}$ or an increase in $Q_w$ results in a lower channel slope. A drop in base level should, after the signal has propagated upstream, result in a slope similar to the channel before the perturbation because the $Q_{s,in}/Q_w$ ratio is unchanged. Hence, our findings suggest that slope comparisons between the terrace surfaces and the active channel could indicate whether an upstream or a downstream perturbation caused the cutting of the terraces. However, such comparisons are only informative if the active channel is still graded to the boundary conditions that initiated incision and~~

585  tectonic tilting of the terraces after cutting.

590 ~~**rates (e.g., Hu et al., 2017; Lavé and Avouac, 2000). The observed slope differences between terrace surfaces and the active channel after upstream perturbations in our experiments (Fig. 6), however, imply that slope differences observed in the field can only be used to infer tectonic deformation rates if one can either rule out (Lavé and Avouac, 2000) or quantify slope changes related to changing $Q_w$ and/or $Q_{s,in}$ (Pazzaglia, 2013). Because the slope changed in our experiments of upstream perturbations, incision rates were not uniform along the channel (Fig.** **4.1.2 Terrace geometry**
595 **as an indicator of perturbation type**

Because fluvial fill terraces result from changes in $Q_w$, changes in $Q_{s,in}$, or a drop in base level, their presence alone does not indicate which of the parameters changed over time. However, our experimental results revealed differences in terrace geometry between changes in upstream ($Q_w$, $Q_{s,in}$) versus downstream ($BLF$) conditions. For terraces related to changes in $Q_{s,in}$ or $Q_w$, the slopes of terrace surfaces are always steeper than the active channel (the new steady state channel after the
600 perturbation), whereas the slope of terraces formed due to downstream perturbations is very similar to that of the active channel (Fig. 7). These observations concur with predictions from theoretical work that suggest a positive scaling of slope and $Q_{s,in}$ and a negative scaling of slope and $Q_w$, while a drop in base level should, after the signal has propagated upstream, result in a slope similar to the channel before the perturbation because of a constant $Q_{s,in}/Q_w$ ratio (Lane, 1955; Mackin, 1948; Malatesta and Lamb, 2018; Meyer-Peter and Müller, 1948; Wickert and Schildgen, 2019; Wobus et al., 2010). Similar observations have
605 been made in the field. In the Tien Shan, Poisson and Avouac (2004) found a successive reduction in slope within a terrace sequence, which they related to changes in $Q_w$. In the Central Andes, Pepin et al. (2013) explained downstream-converging terraces (and thus a reduction in terrace-surface slopes) on a piedmont through variability in climatic drivers. In contrast, along the Chippewa River in the USA (a tributary to the Mississippi River), where terrace cutting is linked to base-level fall, Faulkner et al. (2016) found no substantial slope change between the longitudinal profile reconstructed from the terraces and the modern
610 channel.

Our findings support earlier observations that slope comparisons between the terrace surfaces and the active channel could indicate whether an upstream- or a downstream-sourced perturbation caused the cutting of the terraces (Faulkner et al., 2016; Pepin et al., 2013; Poisson and Avouac, 2004; Wobus et al., 2010). However, such comparisons are only informative if the active channel is still graded to the boundary conditions that initiated incision and terrace cutting. In addition, this approach to identifying the terrace-formation mechanism requires negligible or quantifiable tectonic tilting of the terraces after cutting.

**4.2 Sediment discharge to the deposition zone**

**4.2.1 $Q_s$-signal modification in the transfer zone**

In steady-state, the transfer zone experiences no net sediment deposition or removal of sediment. Hence, when averaged over a certain time, $Q_{s,in}$ equals $Q_{s,out}$ and the signal can be considered as faithfully transmitted to the deposition zone (e.g. Romans et al., 2016). During geometrical channel adjustments, however, when sediment is either deposited or eroded to adjust the channel slope, $Q_{s,out}$ differs from $Q_{s,in}$ (Fig. 6). This is schematically shown in figure 8 by the divergence between the solid line ($Q_{s,in}$) and circles ($Q_{s,out}$). Thus, $Q_s$-signals can be considered as modified as long as the transfer zone is in a transient state. The total time of $Q_s$-signal modification depends on two variables: (1) the time a certain transfer zone requires to reach graded conditions again, i.e. the channel response or equilibrium time (Castelltort and Van Den Driessche, 2003; Howard, 1982; Métivier and Gaudemer, 1999; Paola et al., 1992) and (2) the frequency at which boundary conditions ($Q_w$, $Q_{s,in}$ base level) change. Consequently, if the period of the forcing is shorter than the required response time of the channel reach, the $Q_s$ signal will never be faithfully transmitted (Paola et al., 1992).

**4.2.2 Observable changes in sediment export to the deposition zone ($Q_{s,out}$)**

Regardless of whether the $Q_s$-signal is faithfully transmitted or modified, we observed changes in $Q_{s,out}$ in our experiments, which would likely be reflected by changes in sedimentation rates within the deposition zone. Enhanced $Q_{s,out}$, for example, was generated both by an increase in $Q_w$ (Fig. 6B) and by a drop in base-level (Fig. 6F). Hence, from the $Q_{s,out}$ record alone, the driving mechanism cannot be identified. The temporary $Q_{s,out}$ peak in the $IQ_w$ experiment (Fig. 6B and schematically in Fig. 8C) resembles the observations of earlier numerical (Armitage et al., 2013, 2011; Tucker and Slingerland, 1997) and experimental work (Bonnet and Crave, 2003; van den Berg van Saparoea and Postma, 2008). In both this earlier work and ours, the peak in $Q_{s,out}$ was generated during the transient phase of slope adjustment. According to equation 1, an increase in $Q_w$ will decrease channel slope and, therefore, trigger river incision. Because $Q_{s,in}$ was held constant during the experiment, the additional sediment that reached the outlet was remobilized from within the channel, in particular from the upstream part (Fig. 5C, G; Castelltort and Van Den Driessche, 2003; van den Berg van Saparoea and Postma, 2008; Wickert and Schildgen, 2019). In contrast, a decrease in $Q_w$ requires a steeper channel slope, which is achieved through sediment

deposition within the channel (Fig. 5E). In our experiments, this adjustment appears as a reduction in $Q_{s,out}$ relative to the upstream sediment supply during the transient slope-adjustment phase (Fig. 6C and 8D).

A decrease in $Q_{s,in}$ should, following the achievement of a graded channel profile, reduce $Q_{s,out}$, whereas an increase in $Q_{s,in}$ should increase $Q_{s,out}$ (Allen and Densmore, 2000; Armitage et al., 2011; Bonnet and Crave, 2003). According to equation 1, reducing $Q_{s,in}$ will trigger temporary incision because a lower slope is required to transport less sediment with the same $Q_w$. Conversely, increasing $Q_{s,in}$ without changing $Q_w$ will require a steeper transport slope and thus trigger aggradation. Channel incision and slope reduction occurred in the $DQ_{s,in}$ experiments (Fig. 5D, H and 6D, E), whereas aggradation and slope increase followed an increase in $Q_{s,in}$ (Fig. 5F and 6E). However, in none of the experiments with variable $Q_{s,in}$ was a clear change in $Q_{s,out}$ recognizable during the transient phase of slope adjustment (Fig. 6D, E and 8E, F). We consider the negative feedback between $Q_{s,in}$ and the bed-elevation change during the transient channel-adjustment phase as the main reason for this lack of response (Simpson and Castelltort, 2012; van den Berg van Saparoea and Postma, 2008). The additional sediment supplied upstream is deposited within the channel, resulting in aggradation, and is therefore not detectable at the outlet. When less sediment is supplied upstream, the channel incises and complements the supplied upstream sediment with remobilized sediment from within the channel, such that once again, no change in $Q_{s,out}$ is detectable at the outlet during the adjustment phase. We did not run the experiments long enough to analyze the adjusted steady-state phase, but we would expect that once the channel has adjusted to new equilibrium conditions, $Q_{s,out}$ will eventually equal $Q_{s,in}$ (Fig. 8E, F; Allen and Densmore, 2000; Armitage et al., 2011; Bonnet and Crave, 2003; Wickert and Schildgen, 2019).

Internal channel dynamics can lead to variability in $Q_{s,out}$ even without external forcing. In the *Ctrl_1* and *Ctrl_2* experiments, scatter in the $Q_{s,out}$ signal was up to 5 times the value of $Q_{s,in}$ (Fig. 6a). This variability is due to continuous lateral movement of the channel and subsequent bank collapse, which results in stochastic contributions of additional sediment. Lateral channel mobility of a stream varies with water and sediment discharge (Bufe et al., 2018; Wickert et al., 2013). However, if the volume of sediment mobilized from valley walls due to lateral migration is much larger than the change in $Q_{s,in}$, then no clear signal in $Q_{s,out}$ might be recognizable, even after channel adjustment. The channel instead will continually adjust to the stochastic lateral input of sediment.

Regarding $Q_{s,out}$ signals, we conclude that terraces, floodplains, and the channel itself act as a temporary storage space where sediment can be deposited or remobilized when boundary conditions change (Coulthard et al., 2005; Simpson and Castelltort, 2012; van den Berg van Saparoea and Postma, 2008). The consequence of sediment deposition or remobilization during transient response times is that $Q_{s,in}$ differs from $Q_{s,out}$, such that the $Q_s$-signal can be considered as modified during transient phases of channel adjustment. Our data also support earlier findings by Simpson and Castelltort (2012) and van den Berg van Saparoea and Postma (2008), who concluded from their respective numerical model and physical experiments that $Q_w$ variability creates an amplified, substantial response in $Q_{s,out}$, whereas changes in $Q_{s,in}$ create a dampened response in $Q_{s,out}$ due to the a negative feedback between $Q_{s,in}$ and channel slope. Our experiments, illustrated schematically in Fig. 8, also suggest that $Q_w$-driven $Q_{s,out}$ changes are temporary, and that as the channel slope adjusts to the new input $Q_w$, $Q_{s,out}$ evolves

675 back to its initial steady-state value. In contrast, $Q_{s,out}$ signals driven by changes in $Q_{s,in}$ may not be observable during transient channel adjustment, but will occur and persist once the channel has adjusted to new steady-state conditions.

**4.3 Combining the records**

Both fill terraces in the transfer zone and changes in sedimentation rates in the deposition zone record changes in
680 boundary conditions. Our experiments provide an opportunity to investigate the links between terrace formation and sediment export to the deposition zone, as well as how both of these records, if available, can disambiguate changes in past $Q_{s,in}$, $Q_w$, or base level.

Sediment discharge at the outlet ($Q_{s,out}$) is a function of (1) upstream sediment supply ($Q_{s,in}$) and (2) sediment deposition or mobilization within the transfer zone. Doubling $Q_w$ while holding $Q_{s,in}$ constant triggers river incision to archive
685 a lower channel slope. This incision leaves behind terraces that are steeper than the modern, graded channel (Fig. 9). The sediment mobilized in the transfer zone during the transient incision phase produces a transient peak in $Q_{s,out}$. Reducing $Q_{s,in}$ also reduces the equilibrium transport slope, causing channel incision and terrace abandonment. However, no peak in $Q_{s,out}$ during the transient phase is visible, as the extra sediment remobilized from the transfer zone is compensated by the preceding reduction in $Q_{s,in}$ (Fig. 9). Finally, incision due to a fall in base level also generates a temporary peak in $Q_{s,out}$ due to the
690 additional sediment remobilized within the transfer zone. The terraces left behind following this base-level-driven incision, however, have slopes parallel to that of the modern, graded channel. In summary, $Q_{s,out}$ reflects a combination of $Q_{s,in}$ and the geometrical adjustment of the transfer zone, which in turn is recorded by fill terraces.

Coupling the two records, if available, provides the opportunity to unambiguously identify the forcing mechanism, which is not possible using either the fill terraces or the deposits alone (Fig. 9). For example, the presence of terraces whose
695 slopes are steeper than the present-day channel, combined with a simultaneous but transient peak in $Q_{s,out}$, points towards a change in $Q_w$ as the main driver. In turn, terraces whose slopes are steeper than the main channel in combination with a lagged reduction in $Q_{s,out}$ point towards a change in $Q_{s,in}$ as the main driver. Finally, a temporary increase in $Q_{s,out}$ in combination with channel-parallel terraces that young in the upstream direction indicates past base-level fall. Complications may arise when the forcing includes a combination of changes in $Q_{s,in}$ and $Q_w$. Nevertheless, our results point to the potential of combining terrace
700 records with sediment-export data for the reconstructions of paleo-environmental conditions.

**4.4 Limitations of experiments**

Physical experiments allow for investigations of the isolated influence of individual key parameters on landscape evolution. However, a number of limitations arise when attempting to compare the experimental results to natural settings.
705 First of all, in natural sediment-routing systems, the three distinct zones of erosion, transfer and deposition (Fig. 1) are coupled to one another (Allen, 2017). Erosion processes on the hillslopes, for example, determine the amount of sediment

provided to the transfer zone, i.e. $Q_{s,in}$ (e.g., Dixon et al., 2009; Tofelde et al., 2018). In turn, changes in channel-bed elevation in the transfer zone can affect hillslope-erosion processes (e.g., Hurst et al., 2012; Roering et al., 2007). In our experimental setup, however, we investigate the response of the transfer zone as an isolated feature and can thus not account for any hillslope-channel feedbacks that might lead to additional variations in sediment supply to the channel.

Second, we varied $Q_w$ and $Q_{s,in}$ separately, forcing them to remain independent of one another. In natural systems, however, they are commonly coupled. For example, changes in precipitation can alter both $Q_w$ and $Q_{s,in}$ – directly though changes in rainfall-driven sediment-transport rates from hillslopes to the channel (Bookhagen et al., 2006; Dey et al., 2016; Steffen et al., 2010, 2009) and indirectly through long-term changes in hillslope-stabilizing vegetation types (Garcin et al., 2017; Langbein and Schumm, 1958; Schmid et al., 2018; Torres Acosta et al., 2015; Werner et al., 2018). Those feedback mechanisms between different key parameters ($Q_w$, $Q_{s,in}$) and between sub-zones of sediment-routing systems will likely complicate the forcing-response behavior of natural systems.

Third, we have only investigated a single, braided channel. Therefore, our experimental set-up does not allow us to investigate channel-geometry adjustments related to feedbacks between the main stem and adjacent tributaries (Schumm, 1979, 1973), and related terraces forming at channel junctions (Faulkner et al., 2016; Larson et al., 2015; Schildgen et al., 2016). Furthermore, we can draw no conclusions on terraces forming in meandering rivers, such as those related to meander-bend cut-off (e.g., Erkens et al., 2009; Gonzalez, 2001; Limaye and Lamb, 2016). As such, the lack of terrace formation in the two control experiments after the 'spin-up' time does not imply that autogenic terraces do not exist in natural systems, because several potential mechanisms of autogenic or complex-response terrace formation like meander-bend cut-off (Erkens et al., 2009; Gonzalez, 2001; Limaye and Lamb, 2016; Womack and Schumm, 1977) or internal feedbacks between the main-stem and tributaries (Schumm, 1979, 1973, Gardener 1983, Schumm and Parker 1973, Slingerland and Snow 1988) could not be tested with our experimental set-up.

Fourth, apart from the step changes in input parameters, $Q_{s,in}$ and $Q_w$ were held constant through time. As the experiments exhibit geomorphically effective flow conditions at all times (intermittency equals 1), we assume that the experiments integrate over a number of large floods in natural channels. Natural rivers in turn, experience a wide range of intermittencies. This variability in natural systems complicates any attempts to scale channel response times and lag-times from experiments to real systems, but also complicates the comparison of real systems with each other.

Finally, we performed a limited number of experimental runs, with only the control experiments being repeated. Although we did not repeat the experiments that included external perturbations, we consider the last phase of the two experiments during which we performed two changes ($DQ_w\_IQ_w$ and $IQ_{s,in}\_DQ_{s,in}$) as repetition of the experiments with only one perturbation ($IQ_w$ and $DQ_{s,in}$), albeit with different absolute values of $Q_w$ and $Q_{s,in}$. Comparing those experiments reveals similar trajectories of channel evolution (longitudinal profiles, slope, width; Fig., 5 and 6). In addition, the same boundary conditions ($Q_{s,in}$, $Q_w$) persisted during the 'spin-up' phases as well as at the end of the two experiments during which we performed two changes. During those conditions, the channel slopes always evolved to a value of ~0.07. Although not being exact repetitions of the same experiments, the evolution to the same equilibrium conditions indicates that the results are

reproducible. But, we acknowledge that further repetitions would improve our ability to quantify variability that is internal to each system.

Despite the above limitations, the performance of physical experiments under controlled conditions allows us to directly link causes and their effects in fluvial systems. As the key parameters ($Q_{s,in}$, $Q_w$, base level) can be varied independently, physical experiments provide the opportunity to isolate the influence of different environmental parameters on the evolution of landscapes and to test theoretical models (Paola et al., 2001). As many processes acting in landscapes are scale-independent, experimental observations can help to decipher process behavior in natural systems (e.g., Cantelli et al., 2004).

**4.5 Implications for field studies**

Despite the restrictions when comparing the experimental work to natural systems, the general patterns that we observed do have implications for field studies. First, lag times between the perturbation and the onset of terrace cutting can be important when dating the surfaces of fluvial fill terraces in the field. Common methods to date the onset of river incision include the dating of terrace surface material with cosmogenic exposure dating (e.g., Schildgen et al., 2016; Tofelde et al., 2017), dating sand or silt lenses with optically stimulated luminescence close to the terrace surface (OSL; e.g., Fuller et al., 1998; Schildgen et al., 2016; Steffen et al., 2009) or dating embedded organic material with [14]C (Farabaugh and Rigsby, 2005; Scherler et al., 2015). When transferring our observations to a field scenario, the ~2h or more of channel material reworking before terraces were cut within the upstream part of the channel in the *BLF* and the *DQ*$_{s,in}$ experiment would result in terrace ages that are younger than the time of perturbation. The best temporal correlations between the perturbation and the terrace surface ages are achieved by those formed by changes in $Q_w$ due to the fast onset of vertical incision and minimal reworking of terrace surface material. To assess the significance of this time lag in natural systems requires more work on how to scale the experiment to larger channels.

Second, in tectonically active regions, both strath and fill terraces have been used to calculate river incision rates to infer tectonic uplift rates (e.g., Litchfield and Berryman, 2006; Maddy et al., 2001; Schildgen et al., 2012; Wegmann and 
[revised manuscript text omitted]

**Summary and Conclusion**

We performed seven physical experiments to investigate the effects of changing boundary conditions ($Q_{s,in}$, $Q_w$, base level) on channel geometry and related fill-terrace cutting as well as on sediment discharge ($Q_{s,out}$). To reliably reconstruct paleo-environmental conditions from terraces in the transfer zone or sedimentary deposits in the sedimentation zone, it is important to understand (1) how information on environmental conditions may be modified and eventually transferred into the geologic record, and (2) whether the geomorphic characteristics of terraces or the patterns of sedimentation rates are specific to the forcing mechanism., fill terrace formation and signal propagation in fluvial sediments. In particular, we recorded the evolution of channel slope and width during adjustment to new boundary conditions. Furthermore, we explored the conditions under which fill terraces form and how well they preserve the channel profile prior to perturbation based on lag times between the onset of perturbation and terrace cutting, synchronicity of incision along the length of the channel, and the relationship between terrace surface slopes and terrace formation mechanisms. In addition, we examined the implications of changing

boundary conditions on signal propagation through the sediment routing system. Our experimental findings can be summarized as follows:

Our experiments provided important insights into how sediment discharge to the deposition zone ($Q_{s,out}$) and the formation of terraces in the transfer zone are coupled. The amount of sediment discharged to the deposition zone ($Q_{s,out}$) is a combination of sediment supply to the transfer zone ($Q_{s,in}$) and its modification in the transfer zone through sediment deposition or remobilization. Deposition or remobilization of sediment within the transfer zone occurs mainly during the transient response phase after a perturbation. Hence, during transient response times, $Q_{s,out}$ does not equal $Q_{s,in}$. One consequence of this finding for field studies is that the geochemical composition of sediment sampled during transient phases does not represent the modern hillslope-conditions. For example the [10]Be concentration in detrital sediment can be greatly modified, especially during incision phases, when older sediments within the transfer zone are remobilized.

The same modifications (sediment deposition and remobilization) that alter the $Q_s$-signal during transient times also form fill terraces in the transfer zone. Increases in $Q_w$ trigger channel incision to archive a lower equilibrium slope. The resulting temporary peak in $Q_{s,out}$ coincides with the cutting of terraces whose slopes are steeper than the main channel. Reducing $Q_{s,in}$ also reduces the equilibrium transport slope, causing channel incision and terrace abandonment. However, no substantial increase in $Q_{s,out}$ occurs during the transient phase, as the extra sediment remobilized from the transfer zone is compensated by the preceding reduction in $Q_{s,in}$. Finally, a drop in base-level causes a temporary peak in $Q_{s,out}$ and the formation of terraces parallel to the modern channel. Hence, if both records are available, the combination of the two can unambiguously identify the main forcing mechanism of channel adjustment. The identification of the mechanism can be important, for example, when using the height of terraces to infer channel incision rates. As upstream perturbations cause greater incision at the upstream end than at the downstream end, incision rates inferred from terrace heights are thought to vary along the profile.

1. The cutting of terraces following an upstream perturbation ($Q_{s,in}$, $Q_w$) requires a period of time (lag-time) expected to be a function incision rate, which in turn is thought to be a function of the excess transport capacity of a channel (Wickert and Schildgen, 2019). Indeed, our experiments showed that greater excess transport capacity leads to faster incision and shorter lag-times, which ensures a better preservation potential of the channel profile that existed prior to perturbation. These lag-times can also be critical for field studies that attempt to link the ages of terraces surfaces to the timing of perturbations, as long lag-times may lead to substantial temporal mismatches. An increase in $Q_w$, a decrease in $Q_{s,in}$, or a drop in base level triggered river incision and terrace cutting, combined with an instantaneous reduction in channel width.

2. The observed reduction of channel width after an increase in $Q_w$ runs contrary to the expected channel widening under equilibrium conditions. This finding indicates that the transient response of the fluvial system – not captured in the equilibrium relationship between channel width ($w$), discharge ($Q_w$) and slope ($S$) from the coupled equations of Wickert and Schildgen (2018) – may be significant. We suggest that the

905 transient channel-width response may lead to an excess shear stress at bankfull flow ($\epsilon$) that differs from the commonly assumed and encountered value of ~$0.2\tau_c$ (Parker, 1978; Phillips and Jerolmack, 2016).

[revised manuscript text omitted]

Armitage, J.J., Dunkley Jones, T., Duller, R.A., Whittaker, A.C., Allen, P.A., 2013. Temporal buffering of climate-driven sediment flux cycles by transient catchment response. Earth Planet. Sci. Lett. 369–370, 200–210. doi:10.1016/j.epsl.2013.03.020

Baynes, E.R.C., Lague, D., Kermarrec, J., 2018. Supercritical river terraces generated by hydraulic and geomorphic interactions. Geology 46, 1–4. doi:10.1130/G40071.1

Begin, Z.B., Meyer, D.F., Schumm, S.A., 1981. Development of longitudinal profiles of alluvial channels in response to base-level lowering. Earth Surf. Process. Landforms 6, 49–68.

Bierman, P., Steig, E.J., 1996. Estimating rates of denudation using cosmogenic isotope abundances in sediment. Earth Surf. Process. Landforms 21, 125–139.

Blom, A., Arkesteijn, L., Chavarrias, V., Viparelli, E., 2017. The equilibrium alluvial river under variable flow and its channel-forming discharge. J. Geophys. Res. Earth Surf. 122, 1924–1948. doi:10.1002/2017JF004213

Blom, A., Viparelli, E., Chavarrías, V., 2016. The graded alluvial river: Profile concavity and downstream fining. Geophys. Res. Lett. 43, 6285–6293. doi:10.1002/2016GL068898

Blum, M.D., Tornqvist, T.E., 2000. Fluvial responses to climate and sea-level change: a review and look forward. Sedimentology 47, 2–48. doi:10.1046/j.1365-3091.2000.00008.x

Bonnet, S., Crave, A., 2003. Landscape response  to  climate change. : Insights from experimental modeling and implications for tectonic versus climatic uplift of topography. Geology

 31, 123–126.

Bookhagen, B., Fleitmann, D., Nishiizumi, K., Strecker, M.R., Thiede, R.C., 2006. Holocene monsoonal dynamics and fluvial terrace formation in the northwest Himalaya, India. Geology 34, 601–604. doi:10.1130/G22698.1

Bridgland, D., Westaway, R., 2008. Climatically controlled river terrace staircases: a worlwide Quaternary phenomenon. Geomorphology 98, 285–315.

Brown, E.T., Stallard, R.F., Larsen, M.C., Raisbeck, G.M., Yiou, F., 1995. Denudation rates determined from the accumulation of in situ-produced [10]Be in the Luquillo Experimental Forest, Puerto Rico. Earth Planet. Sci. Lett. 129, 193–202. doi:10.1016/0012-821X(94)00249-X

Bufe, A., Turowski, J.M., Burbank, D.W., Paola, C., Wickert, A.D., Tofelde, S., 2018. Controls on lateral channel mobility and the reworked area of active alluvial surfaces, in: EGU General Assembly Conference Abstracts. p. 13437.

Buffington, J.M., 2012. Changes in channel morphology over human time scales [Chapter 32]. Gravel-Bed Rivers Process. Tools, Environ. 435–463.

Bull, W.B., 1991. Geomorphic responses to climatic change. Oxford Univ. Press.

Bull, W.B., 1990. Stream-terrace genesis: implications for soil development. Geomorphology 3, 351–367. doi:10.1016/0169-555X(90)90011-E

Cantelli, A., Paola, C., Parker, G., 2004. Experiments on upstream-migrating erosional narrowing and widening of an incisional channel caused by dam removal. Water Resources Research, 40(3).

Castelltort, S., Van Den Driessche, J., 2003. How plausible are high-frequency sediment supply-driven cycles in the stratigraphic record? Sediment. Geol. 157, 3–13. doi:10.1016/S0037-0738(03)00066-6

Coulthard, T.J.T., Lewin, J., Macklin, M.G., 2005. Modelling differential catchment response to environmental change. Geomorphology 69, 222–241. doi:10.1016/j.geomorph.2005.01.008

Curtis, K.E., Renshaw, C.E., Magilligan, F.J., Dade, W.B., 2010. Temporal and spatial scales of geomorphic adjustments to reduced competency following flow regulation in bedload-dominated systems. Geomorphology 118, 105–117. doi:10.1016/j.geomorph.2009.12.012

Dade, W.B., Renshaw, C.E., Magilligan, F.J., 2011. Sediment transport constraints on river response to regulation. Geomorphology 126, 245–251. doi:10.1016/j.geomorph.2010.11.007

Dey, S., Thiede, R.C., Schildgen, T.F., Wittmann, H., Bookhagen, B., Scherler, D., Jain, V., Strecker, M.R., 2016. Climate-driven sediment aggradation and incision since the late Pleistocene in the NW Himalaya, India. Earth Planet. Sci. Lett. 449, 321–331. doi:10.1016/j.epsl.2016.05.050

Dixon, J.L., Heimsath, A.M., Kaste, J., Amundson, R., 2009. Climate-driven processes of hillslope weathering. Geology 37, 975–978. doi:10.1130/G30045A.1

Erkens, G., Dambeck, R., Volleberg, K.P., Bouman, M.T.I.J., Bos, J. a a,A.A., Cohen, K.M., Wallinga, J., Hoek, W.Z., 2009. Fluvial terrace formation in the northern Upper Rhine Graben during the last 20,000 years as a result of allogenic controls and autogenic evolution. Geomorphology 103, 476–495. doi:10.1016/j.geomorph.2008.07.021

Farabaugh, R.L., Rigsby, C.A., 2005. Climatic influence on sedimentology and geomorphology of the Rio Ramis valley, Peru. J. Sediment. Res. 75, 12–28.

Faulkner, D.J., Larson, P.H., Jol, H.M., Running, G.L., Loope, H.M., Goble, R.J., 2016. Autogenic incision and terrace formation resulting from abrupt late-glacial base-level fall, lower Chippewa River, Wisconsin, USA. Geomorphology 266, 75–95. doi:10.1016/j.geomorph.2016.04.016

Finnegan, N.J., Dietrich, W.E., 2011. Episodic bedrock strath terrace formation due to meander migration and cutoff. Geology 39, 143–146. doi:10.1130/G31716.1

Fisk, N.H., 1944. Geological investigationInvestigation of the alluvial valleyAlluvial Valley of the Lower Mississippi River. Mississippi River Com. Vicksbg.

Frankel, K.L., Pazzaglia, F.J., Vaughn, J.D., 2007. Knickpoint evolution in a vertically bedded substrate, upstream-dipping terraces, and Atlantic slope bedrock channels. Geol. GSASoc. Am. Bull. 119, 476–486. doi:10.1130/B25965.1

Fuller, I.C., Macklin, M.G., Lewin, J., Passmore, D.G., Wintle, A.G., 1998. River response to high-frequency climate oscillations in southern Europe over the past 200 k.y. Geology 26, 275–278. doi:10.1130/0091-7613(1998)026<0275:RRTHFC>2.3.CO;2

Garcin, Y., Schildgen, T.F., Torres Acosta, V., Melnick, D., Guillemoteau, J., Willenbring, J., Strecker, M.R., 2017. Short-lived increase in erosion during the African Humid Period: Evidence from the northern Kenya Rift. Earth Planet. Sci. Lett. 459, 58–69. doi:10.1016/j.epsl.2016.11.017

Gardner, T.W., 1983. Experimental study of knickpoint and longitudinal profile evolution in cohesive, homogeneous material. Geol. Soc. Am. Bull. 94, 664–572. doi:10.1130/0016-7606(1983)94<664:ESOKAL>2.0.CO

Gilbert, G.K., 1877. Report on the Geology of the Henry Mountains. US Gov. Print. Off.

Godard, V., Tucker, G.E., Burch Fisher, G., Burbank, D.W., Bookhagen, B., 2013. Frequency-dependent landscape response to climatic forcing. Geophys. Res. Lett. 40, 859–863. doi:10.1002/grl.50253

Gonzalez, M.A., 2001. Recent formation of arroyos in the Little Missouri Badlands of southwestern North Dakota. Geomorphology 38, 63–84.

Granger, D.E., Kirchner, J.W., Finkel, R., 1996. Spatially averaged long-term erosion rates measured from in situ-produced cosmogenic nuclides in alluvial sediment. J. Geol. 104, 249–257.

Grimaud, J. L., Paola, C., Voller, V., 2015. Experimental migration of knickpoints: influence of style of base-level fall and bed lithology. Earth Surf. Dyn. Discuss. 3, 773–805. doi:10.5194/esurfd-3-773-2015

Hancock, G.S., Anderson, R.S., 2002. Numerical modeling of fluvial strath-terrace formation in response to oscillating climate. Geol. Soc. Am. Bull. 114, 1131–1142.

Hanson, P.R., Mason, J.A., Goble, R.J., 2006. Fluvial terrace formation along Wyoming's Laramie Range as a response to increased late Pleistocene flood magnitudes. Geomorphology 76, 12–25. doi:10.1016/j.geomorph.2005.08.010

Hay, W.W., Sloan, J.L., Wold, C.N., 1988. Mass/age distribution and composition of sediments on the ocean floor and the global rate of sediment subduction. J. Geophys. Res. Solid Earth 93, 14933–14940.

Hippe, K., Kober, F., Zeilinger, G., Ivy-ochs, S., Maden, C., Wacker, L., Kubik, P.W., Wieler, R., 2012. Quantifying denudation rates and sediment storage on the eastern Altiplano, Bolivia, using cosmogenic $^{10}$Be, $^{26}$Al, and in situ $^{14}$C. Geomorphology 179, 58–70. doi:10.1016/j.geomorph.2012.07.031

Howard, A.D., 1982. Equilibrium and time scales in geomorphology: Application to sand-bed alluvial streams. Hu, X., Pan, B., Fan, Y., Wang, J., Hu, Z., Cao, B., Li, Q., Geng, H., 2017. Folded fluvial terraces in a young, actively deforming intramontane basin between the Yumu Shan and the Qilian Shan mountains, NE Tibet. Lithosphere 9, 545–560. doi:10.1130/L614.1

Earth Surf. Process. Landforms 7, 303–325.

Howard, A.D., 1959. Numerical systems of terrace nomenclature: A critique. J. Geol. 67, 239–243.

Huntington, E., 1907. Some characteristics of the glacial period in non-glaciated region. Bull. Geol. Soc. Am. 18, 351–388.

Hurst, M.D., Mudd, S.M., Walcott, R., Attal, M., Yoo, K., 2012. Using hilltop curvature to derive the spatial distribution of erosion rates. J. Geophys. Res. Earth Surf. 117, 1–19. doi:10.1029/2011JF002057

Lane, E.W., 1955. Importance of fluvial morphology in hydraulic engineering. Proceedings (American Society of Civil Engineers); v. 81, paper no. 745.

Lavé, J., Avouac, J.-P., 2000. Active folding of fluvial terraces across the Siwaliks Hills, Himalayas of central Nepal. J. Geophys. Res. Langbein, W.B., Schumm, S.A., 1958. Yield of sediment in relation to mean annual precipitation. Trans. Am. Geophys. Union 39, 1076–1084.

Larson, P.H., Dorn, R.I., Faulkner, D.J., Friend, D.A., 2015. Toe-cut terraces: A review and proposed criteria to differentiate from traditional fluvial terraces. Prog. Phys. Geogr. 39, 417–439.

105, 5735–5770.

Lewis, W. V, 1944. Stream Trough Experiments and Terrace Formation. Geol. Mag. 81, 241–253.

Limaye, A.B.S., Lamb, M.P., 2016. Numerical model predictions of autogenic fluvial terraces and comparison to climate change expectations. J. Geophys. Res. F Earth Surf. 121, 512–544. doi:10.1002/2014JF003392

Litchfield, N., Berryman, K., 2006. Relations between postglacial fluvial incision rates and uplift rates in the North Island, New Zealand. J. Geophys. Res. 111, 1–15. doi:10.1029/2005JF000374

Litty, C., Duller, R., Schlunegger, F., 2016. Paleohydraulic reconstruction of a 40 ka-old terrace sequence implies that water discharge was larger than today. Earth Surf. Process. Landforms 41, 884–898. doi:10.1002/esp.3872

Mackin, J.H., 1948. Concept of the graded river. Bull. Geol. Soc. Am. 59, 463–512.

Maddy, D., Bridgland, D., Westaway, R., 2001. Uplift-driven valley incision and climate-controlled river terrace development in the Thames Valley, UK. Quat. Int. 79, 23–36.

Malatesta, L., Avouac, J.-P., Brown, N.D., Breitenbach, S.F.M., Pan, J., Chevalier, M.-L., Rhodes, E., Saint-Carlier, D., Zhang, W., Charreau, J., Lavé, J., Blard, P.-H., 2018. Lag and mixing during sediment transfer across the Tian Shan piedmont caused by climate- driven aggradation-incision cycles. Basin Res. 30, 613–635. doi:10.1111/bre.12267

Malatesta, L.C., Avouac, J., 2018. Contrasting river incision in north and south Tian Shan piedmonts due to variable glacial imprint in mountain valleys. Geology 46, 659–662.

Malatesta, L.C., Lamb, M.P., 2018. Formation of waterfalls by intermittent burial of active faults. GSA Bull. 130, 522–536.

Malatesta, L.C., Prancevic, J.P., Avouac, J., 2017. Autogenic entrenchment patterns and terraces due to coupling with lateral erosion in incising alluvial channels. J. Geophys. Res. Earth Surf. 122, 335–355. doi:10.1002/2015JF003797.Received

McPhillips, D., Bierman, P.R., Rood, D.H., 2014. Millennial-scale record of landslides in the Andes consistent with earthquake trigger. Nat. Geosci. 7, 925–930.

Merritts, D.J., Vincent, K.R., Wohl, E.E., 1994. Long river profiles, tectonism, and eustasy: A guide to interpreting fluvial terraces. J. Geophys. Res. 99, 14031–14050.

Métivier, F., Gaudemer, Y., 1999. Stability of output fluxes of large rivers in South and East Asia during the last 2 million years : implications on floodplain processes. Basin Res. 11, 293–303.

Meyer-Peter, E., Müller, R., 1948. Formulas for Bed-Load Transport. In IAHSR 2nd meeting, Stockholm, appendix 2. IAHR.

Mizutani, T., 1998. Laboratory experiment and digital simulation of multiple fill-cut terrace formation. Geomorphology 24, 353–361. doi:10.1016/S0169-555X(98)00027-0

Norton, K.P., Schlunegger, F., Litty, C., 2015. On the potential for regolith control of fluvial terrace formation in semi-arid escarpments. Earth Surf. Dyn. Discuss. 3, 715–738. doi:10.5194/esurfd-3-715-2015

Paola, C., Heller, P.L., Angevine, C.L., 1992. The large-scale dynamics of grain-size variation in alluvial basins, 1: Theory. Basin Res. 4, 73–90. doi:10.1111/j.1365-2117.1992.tb00145.x

Parker, G., Paola, C., Mullin, J., Ellis, C., Mohrig, D.C., Swenson, J.B., Parker, G., Hickson, T., Heller, P.L., Pratson, L., Syvitski, J., 2001. Experimental stratigraphy. GSA TODAY 11, 4–9.

Parker, G., 1979. Hydraulic geometry of active gravel rivers. J. Hydraul. Div. 105, 1185–1201.

1978. Self-formed straight rivers with equilibrium banks and mobile bed. Part 2. The gravel river. J. Fluid Mech. 89, 127–146.

Patton, P.C., Schumm, S.A., 1981. Ephemeral-Stream Processes: Implications for Studies of Quaternary Valley Fills. Quat. Res. 15, 24–43.

Pazzaglia, F.J., 2013. Fluvial terraces. Treatise Geomorphol. 379–412.

Pederson, J.L., Anders, M.D., Rittenhour, T.M., Sharp, W.D., Gosse, J.C., Karlstrom, K.E., 2006. Using fill terraces to understand incision rates and evolution of the Colorado River in eastern Grand Canyon, Arizona. J. Geophys. Res. 111. doi:10.1029/2004JF000201

Penck, A., Brückner, E., 1909. Die Alpen im Eiszeitalter. Tauchnitz.

Peters, G., van Balen, R.T., 2007. Pleistocene tectonics inferred from fluvial terraces of the northern Upper Rhine Graben, Germany. Tectonophysics 430, 41–65. doi:10.1016/j.tecto.2006.10.008

Pfeiffer, A.M., Finnegan, N.J., Willenbring, J.K., 2017. Sediment supply controls equilibrium channel geometry in gravel rivers. Proc. Natl. Acad. Sci. Pepin, E., Carretier, S., Hérail, G., Regard, V., Charrier, R., Farías, M., García, V., Giambiagi, L., 2013. Pleistocene landscape entrenchment: a geomorphological mountain to foreland field case, the Las Tunas system, Argentina. Basin Res. 25, 613–637.

114, 3346–3351. doi:10.1073/pnas.1612907114

Phillips, C.B., Jerolmack, D.J., 2016. Self-organization of river channels as a critical filter on climate signals. Science, 352 (6286), 694–697.

Poisson, B., Avouac, J.P., 2004. Holocene Hydrological Changes Inferred from Alluvial Stream Entrenchment in North Tian Shan (Northwestern China). J. Geol. 112, 231–249.

Roering, J.J., Perron, J.T., Kirchner, J.W., 2007. Functional relationships between denudation and hillslope form and relief. Earth Planet. Sci. Lett. 264, 245–258. doi:10.1016/j.epsl.2007.09.035

Romans, B.W., Castelltort, S., Covault, J. A., Fildani, A., Walsh, J.P., 2016. Environmental signal propagation in sedimentary systems across timescales. Earth-Science Rev. 153, 7–29. doi:10.1016/j.earscirev.2015.07.012

Savi, S., Delunel, R., Schlunegger, F., 2015. Efficiency of frost-cracking processes through space and time: An example from the eastern Italian Alps. Geomorphology 232, 248–260. doi:10.1016/j.geomorph.2015.01.009

Savi, S., Norton, K.P., Picotti, V., Akçar, N., Delunel, R., Brardinoni, F., Kubik, P., Schlunegger, F., 2014. Quantifying sediment supply at the end of the last glaciation: Dynamic reconstruction of an alpine debris-flow fan. GSAGeol. Soc. Am. Bull. 126, 773–790. doi:10.1130/B30849.1

Schaller, M., von Blanckenburg, F., Hovius, N., Veldkamp, A., van den Berg, M.W., Kubik, P.W., 2004. Paleoerosion rates from cosmogenic $^{10}$Be in a 1.3 Ma terrace sequence: responseResponse of the riverRiver Meuse to changes in climate and rock uplift. J. Geol. 112, 127–144.

Scherler, D., Bookhagen, B., Wulf, H., Preusser, F., Strecker, M.R., 2015. Increased late Pleistocene erosion rates during fluvial aggradation in the Garhwal Himalaya, northern India. Earth Planet. Sci. Lett. 428, 255–266. doi:10.1016/j.epsl.2015.06.034

Scherler, D., Lamb, M.P., Rhodes, E.J., Avouac, J.P., 2016. Climate-change versus landslide origin of fill terraces in a rapidly eroding bedrock landscape: San Gabriel River, California. Bull. Geol. Soc. Am. Bull. 128, 1228–1248. doi:10.1130/B31356.1

Schildgen, T., Dethier, D.P., Bierman, P., Caffee, M., 2002. $^{26}$Al and $^{10}$Be dating of Late Pleistocene and Holocene fill terraces: A record of fluvial deposition and incision, Colorado Front Range. Earth Surf. Process. Landforms 27, 773–787. doi:10.1002/esp.352

Schildgen, T.F., Cosentino, D., Robinson, R.A.J., Savi, S., Phillips, W.M., Spencer, J.Q.G., Bookhagen, B., Niedermann, S., Yildirim, C., Echtler, H., Wittmann, HScherler, D., Tofelde, S., Alonso, R.N., Kubik, P.W., Binnie, S.A., Strecker, M.R., 2012. Multi-phased uplift of the southern margin of the Central Anatolian plateau, Turkey: A record of tectonic2016. Landscape response to late Pleistocene climate change in NW Argentina: Sediment flux modulated by basin geometry and connectivity. J. Geophys. Res.upper mantle processes. Earth Surf. 121, 392–414.Planet. Sci. Lett. 317–318, 85–95. doi:10.1016/j.epsl.2011.12.0031002/2015JF003607

Schildgen, T.F., Robinson, R.A.J., Savi, S., Phillips, W.M., Spencer, J.Q.G., Bookhagen, B., Scherler, D., Tofelde, S., Alonso, R.N., Kubik, P.W., Binnie, S.A., Strecker, M.R., 2016. Landscape response to late Pleistocene climate change in NW Argentina: Sediment flux modulated by basin geometry and connectivity. J. Geophys. Res, Earth Surf. 121, 392–414. doi:10.1002/2015JF003607

Schmid, M., Ehlers, T.A., Werner, C., Hickler, T., Fuentes-Espoz, J.-P., 2018. Effect of changing vegetation and precipitation on denudation--Part 2: Predicted landscape response to transient climate and vegetation cover over millennial to million-year timescales, Earth Surf. Dyn. 6.

Schumm, S.A., 1979. Geomorphic thresholds: the concept and its applications. R. Geogr. Soc. 4, 485–515.

Schumm, S.A., 1973. Geomorphic thresholds and complex response of drainage systems. Fluv. Geomorphol. 6, 69–85.

Schumm, S.A., Parker, R.S., 1973. Implication of complex response of drainage systems for Quaternary alluvial stratigraphy. Nat. Phys. Sci. 243, 99–100.

Shen, Z., Törnqvist, T.E., Autin, W.J., Mateo, Z.R.P., Straub, K.M., Mauz, B., 2012. Rapid and widespread response of the Lower Mississippi River to eustatic forcing during the last glacial-interglacial cycle. GSA Bull. 124, 690–704. doi:10.1130/B30449.1

Simpson, G., Castelltort, S., 2012. Model shows that rivers transmit high-frequency climate cycles to the sedimentary record. Geology 40, 1131–1134. doi:10.1130/G33451.1

Slingerland, R.L., Snow, R.S., 1988. Stability analysis of a rejuvenated fluvial system. Zeitschrift für Geomorphol. 67, 93–102.

Steffen, D., Schlunegger, F., Preusser, F., 2010. Late Pleistocene fans and terraces in the Majes valley, southern Peru, and their relation to climatic variations. Int. J. Earth Sci. 99, 1975–1989.

Steffen, D., Schlunegger, F., Preusser, F., 2009. Drainage basin response to climate change in the Pisco valley, Peru. Geology 37, 491–494.

Tebbens, L.A., Veldkamp, A., Dijke, J.J. Van, Schoorl, J.M., 2000. Modeling longitudinal-profile development in response to Late Quaternary tectonics, climate and sea-level changes: the River Meuse. Glob. Planet. Change 165–186.

Tofelde, S., Duesing, W., Schildgen, T.F., Wickert, A.D., Wittmann, H., Alonso, R.N., Strecker, M.R., 2018. Effects of deep-seated versus shallow hillslope processes on cosmogenic $^{10}$Be concentrations in fluvial sand and gravel. Earth Surf. Process. Landforms 43, 3086–3098, doi:10.1002/esp.4471.

Tofelde, S., Schildgen, T.F., Savi, S., Pingel, H., Wickert, A.D., Bookhagen, B., Wittmann, H., Alonso, R.N., Cottle, J., Strecker, M.R., 2017. 100 kyr fluvial cut-and-fill terrace cycles since the Middle Pleistocene in the southern Central Andes, NW Argentina. Earth Planet. Sci. Lett. 473, 141–153. doi:10.1016/j.epsl.2017.06.001

Torres Acosta, V., Schildgen, T.F., Clarke, B.A., Scherler, D., Bookhagen, B., Wittmann, H., von Blanckenburg, F., Strecker, M.R., 2015. Effect of vegetation cover on millennial-scale landscape denudation rates in East Africa. Lithosphere 7, 408–420. doi:10.1130/L402.1

Tucker, G.E., Slingerland, R., 1997. Drainage basin responses to climate change. Water Resour. Res. 33, 2031–2047.

van den Berg van Saparoea, A.-P.H., Postma, G., 2008. Control of climate change on the yield of river systems. Recent Adv. Model. Siliciclastic Shallow-Marine Stratigr. SEPM Spec. Publ. 90, 15–33.

Vandenberghe, J., 2003. Climate forcing of fluvial system development: An evolution of ideas. Quat. Sci. Rev. 22, 2053–2060. doi:10.1016/S0277-3791(03)00213-0

Vandenberghe, J., 1995. Timescales, climate and river development. Quat. Sci. Rev. 14, 631–638. doi:10.1016/0277-3791(95)00043-O

Veldkamp, A., 1992. A 3/D model of quaternary terrace development, simulations of terrace stratigraphy and valley asymmetry: A case study for the Allier terraces (Limagne, France). Earth Surf. Process. Landforms 17, 487–500.

Veldkamp, A., Tebbens, L.A., 2001. Registration of abrupt climate changes within fluvial systems: insights from numerical modelling experiments. Glob. Planet. Change 28, 129–144.

Veldkamp, A., Van Dijke, J.J., 2000. Simulating internal and external controls on fluvial terrace stratigraphy: a qualitative comparison with the Maas record. Geomorphology 33, 225–236.

Veldkamp, A., Vermeulen, S.E.J.W., 1989. River terrace formation, modelling, and 3-d graphical simulation. Earth Surf. Process. Landforms 14, 641–654.

Veldkamp, T., Van Dijke, J.J., 1998. Modelling long-term erosion and sedimentation processes in fluvial systems: a case study for the Allier/ Loire system. Paleohydrology Environ. Chang.

Whipple, K.X., Tucker, G.E., 1999. Wegmann, K.W., Pazzaglia, F.J., 2009. Late Quaternary fluvial terraces of the Romagna and Marche Apennines, Italy: Climatic, lithologic, and tectonic controls on terrace genesis in an active orogen. Quat. Sci. Rev. 28, 137–165. doi:10.1016/j.quascirev.2008.10.006

Werner, C., Schmid, M., Ehlers, T.A., Fuentes-Espoz, J.P., Steinkamp, J., Forrest, M., Liakka, J., Maldonado, A., Hickler, T., 2018. Effect of changing vegetation and precipitation on denudation-Part 1: Predicted vegetation composition and cover over the last 21 thousand years along the Coastal Cordillera of Chile. Earth Surf. Dyn. 6.

Dynamics of the stream-power river incision model: Implications for height limits of mountain ranges, landscape response timescales, and research needs. J. Geop 104, 17661–17674.

Wickert, A.D., Martin, J.M., Tal, M., Kim, W., Sheets, B., Paola, C., 2013. River channel lateral mobility: Metrics, time scales, and controls. J. Geophys. Res. Earth Surf. 118, 396–412. doi:10.1029/2012JF002386

Wickert, A.D., Schildgen, T.F., 2019. Long-profile evolution of transport-limited gravel-bed rivers. Earth Surf. Dyn. 7, 17–43. doi:10.5194/esurf-7-17-2019

1225 Wittmann, H., Malusà, M.G., Resentini, A., Garzanti, E., Niedermann, S., 2016. The cosmogenic record of mountain erosion transmitted across a foreland basin: source-to-sink analysis of in situ $^{10}$Be, $^{26}$Al and $^{21}$Ne in sediment of the Po river catchment. Earth Planet. Sci. Lett. 452, 258–271. doi:10.1016/j.epsl.2016.07.017

Wittmann, H., von Blanckenburg, F., Maurice, L., Guyot, J.L., Kubik, P.W., 2011. Recycling of Amazon floodplain sediment quantified by cosmogenic 26Al and 10Be. Geology 39, 467–470. doi:10.1130/G31829.1

1230 Wobus, C.W., Tucker, G.E., Anderson, R.S., 2010. Does climate change create distinctive patterns of landscape incision? J. Geophys. Res. Earth Surf. 115.

Wohl, E., Ikeda, H., 1997. Experimental simulation of channel incision into a cohesive substrate at varying gradients. Geology 25, 295–298.

Womack, W.R., Schumm, S. a., 1977. Terraces of Douglas Creek, northwestern Colorado: An example of episodic erosion.
1235 Geology 5, 72–76. doi:10.1130/0091-7613(1977)5<72:TODCNC>2.0.CO;2

Zhang, P., Molnar, P., Downs, W.R., 2001. Increased sedimentation rates and grain sizes 2-4 Myr ago due to the influence of climate change on erosion rates. Nature 410, 891–897. doi:10.1038/35073504

1240

**Table 1. Water and sediment inputs to the experiments.**

| Experiment | 0 – 240 min (reference conditions) | | 240 – 480 min | | 480 min until end | | Graphical description |
|---|---|---|---|---|---|---|---|
| | $Q_w$ (ml s$^{-1}$) | $Q_{s,in}$ (ml s$^{-1}$) | $Q_w$ (ml s$^{-1}$) | $Q_{s,in}$ (ml s$^{-1}$) | $Q_w$ (ml s$^{-1}$) | $Q_{s,in}$ (ml s$^{-1}$) | |
| *Ctrl_1* | 95 | 1.3 | 95 | 1.3 | 95 | 1.3 | |
| *Ctrl_2* | 95 | 1.3 | 95 | 1.3 | 95 | 1.3 | |
| *IQ$_w$* | 95 | 1.3 | 190 | 1.3 | 190 | 1.3 | |
| *DQ$_w$_IQ$_w$* | 95 | 1.3 | 47.5 | 1.3 | 95 | 1.3 | |
| *DQ$_{s,in}$* | 95 | 1.3 | 95 | 0.22 | 95 | 0.22 | |
| *IQ$_{s,in}$_DQ$_{s,in}$* | 95 | 1.3 | 95 | 2.6 | 95 | 1.3 | |
| *BLF* | 95 | 1.3 | 95 | 1.3 | 95 | 1.3 | |

Formatted Table

[Figure]

**Figure 1.** Schematic summary of a sediment-routing system and records of landscape evolution. Sediment-routing systems are typically subdivided into three zones: sediment production, sediment transfer, and sediment deposition (Allen, 2017; Castelltort and Van Den Driessche, 2003). As sediment production is thought to vary with environmental conditions, changes in those conditions might be preserved in sedimentary records in the transfer zone (e.g., fill terraces) or deposition zone (e.g., sedimentation rates). Complications arise, however, as alluvial rivers within the transfer zone continuously adjust their channel geometry to the incoming water discharge ($Q_w$) and sediment supply ($Q_{s,in}$) through sediment deposition or remobilization, and thus modify the sedimentary signal. The amount of upstream sediment supply combined with additional sediment remobilized within the transfer zone minus the amount of sediment deposited in the transfer zone determines how much sediment is discharged from the transfer zone to the deposition zone ($Q_{s,out}$). Figure modified from Castelltort and Van Den Driessche (2003).

[Figure]

Figure 2. Experimental setup, data collection, and analysis. (A) Overview of experimental setup. Sediment supply ($Q_{s,in}$) and water discharge ($Q_w$) can be regulated separately. For all but the base-level fall (*BLF*) experiment, the base level was fixed. Water and sediment fell off of an edge at the outlet. For the *BLF* experiment (shown in the picture), the base level was controlled through the water level in the surrounding basin. (B) Digital elevation model (DEM) derived from laser scans showing the final topography of the increased water ($IQ_w$) experiment. (C) Overhead photograph of the $IQ_w$ experiment taken directly before the scan shown in B. The surface was covered with a thin layer of red sand before the instant increase in $Q_w$ discharge was performed. The remnants of red sand on the terraces indicate no further reworking after the onset of increased discharge. (D) Overhead photographs were turned into binary (wet, dry) images from which the average channel width within the analyzed area (orange frame) can be calculated.

[Figure]

[Figure]

1270

**Figure 3.** Fill terraces formed during experimental runs. (A-D) Top-view of  terraces that formed due to upstream perturbations. Remnants of red sand on the terrace surfaces indicate  areas that have not been flooded after the change in boundary condition was performed (i.e. fill-top terrace). The other terrace surfaces were cut with the indicated lag-times (fill-cut terraces). (E-F) During the base-level fall (*BLF*) experiment, terraces at the downstream end were abandoned instantly after the onset of base-level fall (250 min = 10 min after onset of *BLF*).  Those terraces were destroyed shortly after they were cut. A new set of terraces  formed in the upstream part ca. 112 and 117 min after the onset of *BLF*. (G-H) Downstream view for the *BLF* (G) and *IQₘ* (H) experiments.

1275

[Figure]

1280

[Figure]

**Figure 4.** Evolution of cross -sections in the upper part of the channelreach (left panel). In each cross section, the lowest point is set equal to zero to track incision. The color scheme represents time aftersince the 'spin-up' phase (i.e.last change in boundary conditions (equivalent to either 240 min or 480 min experiment time). For better comparison, we plot a maximum of 240 minutes for all experiments, despite longer recordings for some of the runs. Exact location of cross sections are indicated by the black lines in the DEMs displaying the last scan of each experiment (right panel). Cross-sections haven been chosen at the terrace midpoints and thus vary slightly between the experiments. The times given in parentheses are the absolute experiment runtimes.

[Figure]

[Figure]

1290    **Figure 5.** Evolution of longitudinal river profiles from minute 240 (end of 'spin-up' phase) onwards.  River profiles were extracted from the laser scans. Laser scans were recorded every 30 min, and an additional two scans at 10 and 20 minutes after the initiation of the base-level fall were conducted during the *BLF* experiment. Dashed arrows indicate down-basin distance along which terraces formed. Solid arrows indicate modes of aggradation or incision. Note that the $DQ_w\_IQ_w$ and $IQ_{s,in}\_DQ_{s,in}$ were split into two panels each, with one panel representing each phase.

1295

[Figure]

[Figure]

**Figure 6.** Input parameters and evolution of channel slope,  channel width and $Q_{s,out}$ during the experiments. Input sediment ($Q_{s,in}$; orange solid line) and water ($Q_w$; blue solid line) discharge were normalized to the reference input values ($Q_{s,ref}$ = 1.3 ml/s and $Q_{w,ref}$ = 95 ml/s). Slope ($S$, grey circles) was calculated based on the bed elevation difference between the inlet and the outlet divided by the length of the system. Channel elevation measurements for slope calculations were performed manually during the runs. Black arrows indicate times when terraces in the upstream part of the sandbox started to be cut. Channel width was calculated as the mean number (solid lines) of wet pixels in each of 1200 cross section within the box indicated in Fig. 2C, D. The colored shaded areas around the curves indicate the standard deviation of the 1200 measurements. The evolution of width without any external perturbation (*Ctrl_2*) is plotted for comparison with each other experiment in which external conditions were changed (B-F). Note that no measurements are available for the *Ctrl_1* experiment due to issues with the installation of the overhead camera. Sediment discharge at the outlet ($Q_{s,out}$; grey circles) during the experimental runs is compared to input sediment ($Q_{s,in}$; orange solid line); both were normalized to reference input values ($Q_{s\_ref}$ = 1.3 ml/s). Note that no $Q_{s,out}$ measurements are available for the first 280 min of the *BLF* experiment, as no sample collection was possible during the flooding of the surrounding basin during base-level regulations (Fig. 2A). The first 240 min of each experiment were adjustment to the reference settings (grey box) and were not included in the analyses.

[Figure]

[Figure]

 **Figure 7.** Elevation profile and slope comparison of terrace surfaces and active channels. Elevation profiles are given as mean (solid lines) and minimum and maximum values (dashed lines), extracted along a 5 cm wide swaths as indicated on the right panel. Swath width was reduced in two cases of too narrow terraces to 1 cm ($DQ_w\_IQ_w$ T$_L$T$_A$ terrace) and 2 cm ($DQ_{s,in}$ T$_L$T$_A$ terrace). T$_L$T$_A$ and T$_R$T$_B$ indicate terraces on one side each and refer to labels of lag-times given in figure 6Fig.5. Slopes were calculated based on a linear fit through the mean elevation profiles. Numbers in parentheses give the RMSE between the linear fit and the measured data. For the four experiments

1320    in which upstream conditions were changed (A-D), the slopes of the terraces are steeper than those of the active channel at the end of the experiment. In contrast, in the *BLF* experiment, slopes of the terraces and the active channels are about the same. Note the different y-axis for the $IQ_w$ run that was necessary to display the deep incision.for better visibility. Colors for elevationsof elevation in the right panel are the same as those in figure 4Fig. 3.

1325

[Figure]

[Figure]

**Figure 8.** Schematic model of the evolution of $Q_{s,out}$ including inferred ages of the sediment. (A-B) Channel geometry and approximate distribution of sediment deposition ages during phases of incision and aggradation. (C-F) Evolution of $Q_{s,out}$ (circles) compared to $Q_{s,in}$ (grey solid line) during the transient response phase after perturbation (dark grey) as well as after channel adjustment (light grey). The colours of the circles indicate the age (i.e. storage times before export) of the discharged sediment according to the panels A and B.

[Figure]

1335 **Figure 9**. Combining two records of landscape evolution to overcome ambiguity. Terraces in the transfer zone and sediment discharge to the deposition zone ($Q_{s,out}$) as a proxy for sedimentation rate. The presence of terraces whose slopes differ from that of the main channel, combined with a simultaneous but transient peak in $Q_{s,out}$, points towards $Q_w$ as the main driver of long-profile evolution. Terraces with steeper slopes compared to the modern channel combined with no immediate peak but an eventual reduction in $Q_{s,out}$, points towards $Q_{s,in}$ as the main driver. A temporary increase in $Q_{s,out}$ combined with terraces that parallel the modern channel profile and become younger 1340 in the upstream direction indicate past changes in base level.

. Schematic model of the evolution of signals at the outlet stored in either sediment volume or the chemical composition of the sediment.